# Connectome-driven neural inventory of a complete visual system

Aljoscha Nern[1], Frank Loesche[1,6], Shin-ya Takemura[1,6], Laura E. Burnett[1,6], Marisa Dreher[1,6], Eyal Gruntman[2,6], Judith Hoeller[1,6], Gary B. Huang[1,6], Michał Januszewski[3,6], Nathan C. Klapoetke[1,6], Sanna Koskela[1,6], Kit D. Longden[1,6], Zhiyuan Lu[1,6], Stephan Preibisch[1,6], Wei Qiu[1,6], Edward M. Rogers[1,6], Pavithraa Seenivasan[1,6], Arthur Zhao[1,6], John Bogovic[1], Brandon S. Canino[1], Jody Clements[1], Michael Cook[1], Samantha Finley-May[1], Miriam A. Flynn[1], Imran Hameed[1], Alexandra M. C. Fragniere[4,5], Kenneth J. Hayworth[1], Gary Patrick Hopkins[1], Philip M. Hubbard[1], William T. Katz[1], Julie Kovalyak[1], Shirley A. Lauchie[1], Meghan Leonard[1], Alanna Lohff[1], Charli A. Maldonado[1], Caroline Mooney[1], Nneoma Okeoma[1], Donald J. Olbris[1], Christopher Ordish[1], Tyler Paterson[1], Emily M. Phillips[1], Tobias Pietzsch[1], Jennifer Rivas Salinas[1], Patricia K. Rivlin[1], Philipp Schlegel[4,5], Ashley L. Scott[1], Louis A. Scuderi[1], Satoko Takemura[1], Iris Talebi[1], Alexander Thomson[1], Eric T. Trautman[1], Lowell Umayam[1], Claire Walsh[1], John J. Walsh[1], C. Shan Xu[1], Emily A. Yakal[1], Tansy Yang[1], Ting Zhao[1], Jan Funke[1], Reed George[1], Harald F. Hess[1], Gregory S. X. E. Jefferis[4,5], Christopher Knecht[1], Wyatt Korff[1], Stephen M. Plaza[1], Sandro Romani[1], Stephan Saalfeld[1], Louis K. Scheffer[1✉], Stuart Berg[1✉], Gerald M. Rubin[1✉] & Michael B. Reiser[1✉]

Vision provides animals with detailed information about their surroundings and conveys diverse features such as colour, form and movement across the visual scene. Computing these parallel spatial features requires a large and diverse network of neurons. Consequently, from flies to humans, visual regions in the brain constitute half its volume. These visual regions often have marked structure–function relationships, with neurons organized along spatial maps and with shapes that directly relate to their roles in visual processing. More than a century of anatomical studies have catalogued in detail cell types in fly visual systems[1–3], and parallel behavioural and physiological experiments have examined the visual capabilities of flies. To unravel the diversity of a complex visual system, careful mapping of the neural architecture matched to tools for targeted exploration of this circuitry is essential. Here we present a connectome of the right optic lobe from a male *Drosophila melanogaster* acquired using focused ion beam milling and scanning electron microscopy. We established a comprehensive inventory of the visual neurons and developed a computational framework to quantify their anatomy. Together, these data establish a basis for interpreting how the shapes of visual neurons relate to spatial vision. By integrating this analysis with connectivity information, neurotransmitter identity and expert curation, we classified the approximately 53,000 neurons into 732 types. These types are systematically described and about half are newly named. Finally, we share an extensive collection of split-GAL4 lines matched to our neuron-type catalogue. Overall, this comprehensive set of tools and data unlocks new possibilities for systematic investigations of vision in *Drosophila* and provides a foundation for a deeper understanding of sensory processing.

A wealth of high-performance visual behaviours and genetic tools for targeting specific cell types have made *Drosophila* highly suited for the study of neural circuit implementations of visual computations[4]. The anatomical understanding of the *Drosophila* visual system has been revolutionized by connectomics, which involves the reconstruction of neuron shapes and synaptic connections from electron microscopy (EM) data[5–8]. The motion–vision pathway provides an impressive example. After years of behavioural and physiological studies, detailed EM reconstructions of a small piece of the *Drosophila* visual system transformed the field by proposing several testable hypotheses for the mechanism of directional selectivity in T4 and T5 neurons[6,9]. Connectome analyses revealed that these cells receive asymmetric synaptic inputs from distinct neuron populations at different positions along their asymmetrically shaped dendrites. This structural information and the identity of implicated cell types launched a decade of experiments testing these mechanistic proposals[10]. This development illustrates

that crucial insights into function come from describing the shapes and connections of neurons. A complete visual system connectome will bring this level of analysis and incisive experimental design to all areas of visual processing.

A complete understanding of the functions of different brain regions is an ongoing challenge. However, there is little mystery about the role of visual areas of the brain because their neurons and circuits must be involved in sight. This core knowledge guides the analysis of neurons engaged in vision. That is, these neurons sample information from across the field of view, and this information is transformed by a series of steps that extract increasingly selective signals. These signals are subsequently conveyed to higher brain areas. The mammalian retina is an iconic example of this ground plan, and insects such as *Drosophila* and even the microwasp *Megaphragma*[11] exhibit the same organization in their visual systems. Studies of the mouse retina have used anatomical and functional methods to sort neurons into a comprehensive inventory of >40 retinal ganglion cell types[12–14]. A defining property of retinal cells is their arrangement into anatomical mosaics, with cells of the same type organized with approximately uniform coverage to enable sampling of all points in the tissue without blind spots[13,14]. Together, these efforts in fly and mouse vision clearly demonstrate the power of careful cell typing for functional analyses.

With recent progress in the acquisition and analyses of *Drosophila* EM connectomes[8,15–17] and light microscopy[18,19] (LM) data, the fruit-fly increasingly has an important role in understanding vision. In *Drosophila*, these studies were built on a comprehensive foundation, including the morphology of the relevant neurons, their neurotransmitters and their synaptic connections, as well as cell-type-specific genetic tools to manipulate them[20,21]. All previous systematic EM studies of *Drosophila* visual neurons were based on female brains[6,9,22–24]. Here we present a new dataset and a detailed examination of the visual system in a male brain, thereby establishing a comprehensive survey of an insect optic lobe. We detail the cell typing of all the visual neurons and a matched collection of genetic drivers. These insights and resources are poised to facilitate a holistic understanding of vision.

## Neurons of the *Drosophila* visual system

The complete central nervous system (CNS), including the entire brain and ventral nerve cord, of a male *Drosophila* was dissected, fixed, stained and cut into 66 20-µm slabs. These slabs were then imaged using seven customized focused ion beam milling and scanning electron microscopes over approximately 1 year. Subsequently, the volume was aligned, synapses were identified (Extended Data Fig. 1a) and neuron fragments were extracted using automatic segmentation (Methods). Proofreading of the visual regions on one side (Extended Data Table 1 and Extended Data Fig. 1) established the connectome presented here (available through the neuPrint database; see the Data availability section). Proofreading of the CNS is ongoing. Here we provide a complete catalogue of the neurons in this visual system (Fig. 1a,b and Supplementary Table 1; with a quantitative inventory presented in Supplementary Fig. 1).

The fly visual system comprises several distinct anatomical regions: the lamina, the medulla, the accessory medulla, the lobula and the lobula plate. Together, they form the structure called the optic lobe. Homologous regions are found in visual systems across pancrustaceans[25]. These regions are referred to as neuropils, which are structures dense with the synaptic connections of dendrites and axons. These are shown here as meshes (Fig. 1a,b), which were drawn using visible boundaries in the EM data. Our dataset contains all optic lobe neuropils in their entirety, except for the most peripheral neuropil, the lamina. In invertebrate nervous systems, the cell bodies are generally outside the neuropils.

We first classified neurons on the basis of their relationship to these anatomical regions. Optic neuropil intrinsic neurons (ONINs) have synapses confined to a single neuropil, whereas optic neuropil connecting neurons (ONCNs) connect two or more neuropils. A large and diverse set of neuron types connect the optic lobe neuropils with the central brain. The visual projection neurons (VPNs) primarily convey signals to the central brain, whereas visual centrifugal neurons (VCNs) primarily convey central brain signals to the optic lobes. Figure 1c shows a representative cell type from each group, visualized on a slice of the principal optic lobe neuropils. Several important features of visual neurons are immediately apparent, including that many types form sets of repeating neurons that collectively cover large swaths of neuropils with distinguishing innervation patterns. Nearly all connections between visual neurons are in the neuropils. However, we also found connections outside the neuropils, most frequently in the chiasmata that connect them (Extended Data Fig. 2c–e and Supplementary Table 2). For example, such connections were found in the outer chiasm involving nearly all lamina-associated neurons, a finding that extends a recent observation made for R7 and R8 photoreceptors[23].

We classified around 53,000 visual system neurons into 732 cell types (Supplementary Video 1). The number of neurons that constitute a type varied widely, with 60 cell types accounting for >50% of the synaptic connections and about 75% of the neurons. Figure 1d highlights the 160 cell types with the largest contributions to total connectivity in this connectome, including the following well-studied neurons of the directionally selective motion pathway[9,26–29]: L1, L2, L3, L5, Mi1, Mi4, Mi9, Tm1, Tm2, Tm3, Tm4, Tm9 and all 8 T4 and T5 types. It may be that the need for fast, high-resolution motion signals requires many cells and synapses. Moreover, this unbiased connectivity survey revealed numerous cells with substantial contributions to the network that have so far received minimal attention, including many Pm and TmY cell types. The inventory is summarized in Fig. 1e,f, which details the cell-type distribution by group and neuropil. We include 32 cell types not placed in our main groups. These are primarily central brain cells with limited connectivity in optic lobe neuropils.

The connectome contains about 49 million connections in the optic lobe neuropils. Figure 1g and Extended Data Fig. 2a,b summarize the input–output connectivity of all cell types, colour-coded by their group. The number of connected cells in the network spans five orders of magnitude across the visual cell types. Individual R1–R6 photoreceptors and lamina monopolar cells (for example, L1) are connected to only a few other cells, whereas on the other extreme, several giant individual inhibitory interneurons, such as Am1 and CT1, are connected to >10,000 other cells.

Our naming scheme, explained in the Methods, aims to be systematic while respecting well-established historical names. We introduce new, short names for all newly described cell types. For example, for VPNs and VCNs, the names are based on the principal neuropil, with an appended VP or VC and a unique number. For cases in which new evidence convinced us that a previously defined cell type should be divided into several distinct types, we append a letter.

## Cell typing visual neurons

We adopted a practical cell-type definition that follows the *Caenorhabditis elegans* connectome[30], the mammalian retina[13,14] and decades of work on fly visual neurons[3,5,6,9,22,31]. We grouped neurons that share similar morphology and connectivity patterns into types, with the aim that all neurons assigned to a type are more similar to each other than to any neurons assigned to different types. We generated type annotations in an iterative process that combined visual inspection of the morphology of each cell with computational methods for grouping cells by connectivity. Several team members reviewed all neurons in the dataset.

We started by naming easily recognizable neurons, building on LM studies[3,18,19], previous EM work[6] and our unpublished LM analyses. These neurons included the 15 columnar cell types in Fig. 2a, which are present in nearly every retinotopic medulla column (Supplementary Video 2). Each of these cell types could be identified by morphology. Indeed, even cells such as Mi4 and Mi9 that appear similar are distinguishable

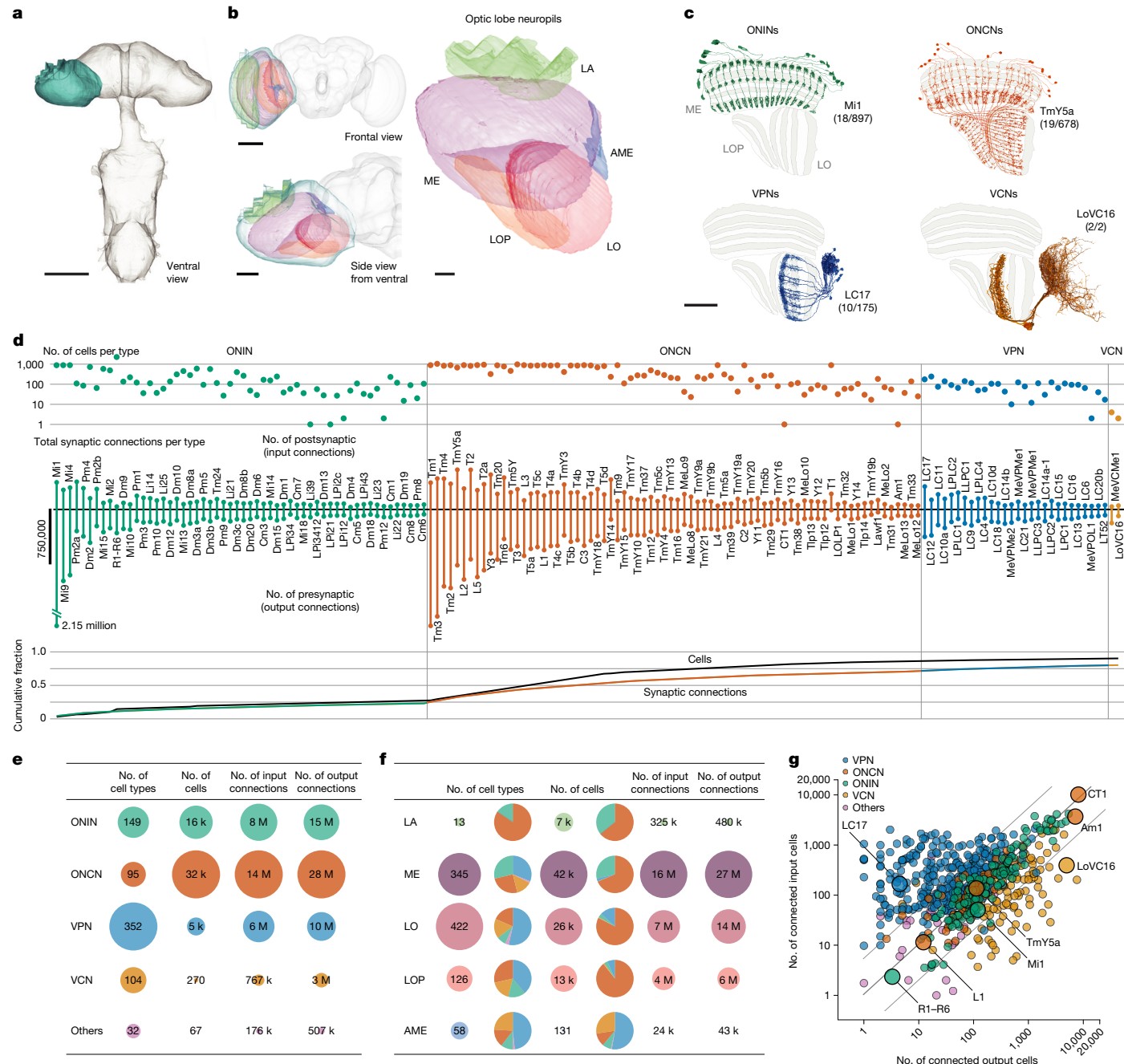

**Fig. 1 | The neurons of a male *Drosophila* visual system. a**, Overview of the male *Drosophila* CNS volume dataset, with the ventral nerve cord (VNC) attached. Scale bar, 200 μm. **b**, This study describes the complete connectome and neuron inventory of the right optic lobe (blue). The optic lobe comprises five neuropils: the lamina (LA), the medulla (ME), the accessory medulla (AME), the lobula (LO) and the lobula plate (LOP). Scale bars, 100 μm (frontal view), 50 μm (side view) or 20 μm (optic lobe neuropils). **c**, The four main groups of cell types, with an example provided for each (values in parentheses indicate the number of cells out of the total number of the cell type). Scale bar, 50 μm. **d**, The number of cells and input–output synaptic connections for the 160 cell types with the largest contributions to total connectivity in the visual system connectome (all types are presented in Supplementary Table 1). Fewer than 100 cell types account for most cells and connections. **e**, Summary of the inventory, with the number of cell types, cells and connections aggregated by cell-type

groups. The inventory includes a small group of 'other' cell types with minimal connectivity. k, thousand; M, million. **f**, Summary as in **e**, but grouped by the five optic lobe neuropils. Counts include cell types and cells with >2% of their synapses contained in a neuropil; many cells contribute to multiple neuropils. Contributions from the neurons in each cell-type group are shown as pie charts. These counts summarize the connectome in the optic-lobe:v1.1 neuPrint database and reflect the asymmetry between the completion percentage for presynapses and postsynapses (Extended Data Table 1). A few cell types are undercounted (estimated 2,777 cells from the lamina and 459 R7 and R8 photoreceptors, see Methods for details). **g**, Input–output connectivity in the optic lobe for all cell types in the inventory. VPNs generally have more input cells than output cells, whereas VCNs show the opposite connectivity pattern (excluding central brain connectivity). The example cell types from **c** and others with unusually high and low connectivity are highlighted.

at LM and, even more clearly, at EM resolution (Fig. 2b). Notably, Mi4 and Mi9 neurons could be independently classified by connectivity, even when only considering a few strongly connected input and output

types (Fig. 2c). Such connectivity differences are best assessed once connections were combined for all neurons of a cell type, because two putative type A neurons may have synaptic connections with cells of

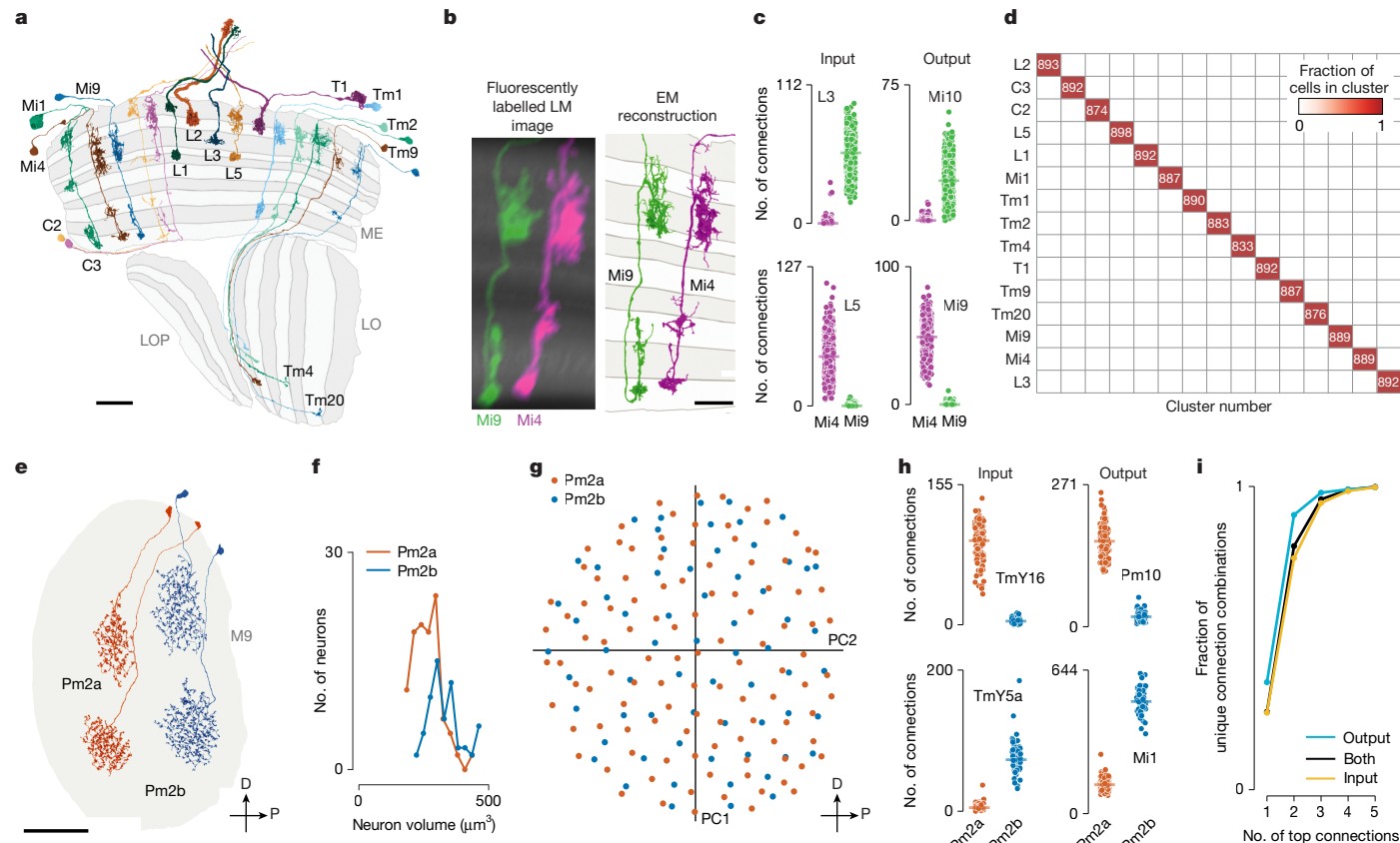

**Fig. 2 | Sorting neurons into types on the basis of morphology and connectivity. a**, Examples of 15 cell types that occur once per column in nearly every column of the visual system, shown across a slice of the optic lobe. Scale bar, 20 μm. **b**, Mi9 and Mi4, shown as LM images (Methods) and EM reconstructions, are an example of neurons that appear similar but can be distinguished by morphology in nearly all cases. Scale bar, 10 μm. **c**, Mi4 and Mi9 can also be distinguished by connectivity, as shown for selected input and output cell types. **d**, After cell typing all neurons, connectivity clustering sorted the cells assigned to the 15 columnar cell types shown in **a** into clearly distinct (note the lack of colour gradient) groups of approximately equal cell counts (numbers in the red squares), thereby confirming their assignments. **e**, Example of sorting cells with similar shapes. Two representatives of each of Pm2a and Pm2b neurons, medulla amacrine cells with highly similar morphology. D, dorsal; P, posterior. Scale bar, 50 μm. **f**, The distribution of cell volume is

similar for Pm2a and Pm2b neurons, which makes sorting these cells on the basis of this metric unreliable. **g**, Pm2a and Pm2b are sorted into two types by connectivity clustering (Extended Data Figs. 3 and 4), which reveals two overlapping mosaics. The first two principal components (PC1 and PC2) of the centres of mass of their synapse locations are plotted. This visualization preserves the spatial relationships of the cells and aligns with major anatomical axes (in this case, the medulla). **h**, Selected distinguishing input and output connections for Pm2a and Pm2b cells. Individual points represent connections of single Pm2a and Pm2b neurons; horizontal line represents the median for each group. The combination of consistent connectivity differences with overlapping cell distributions supports the split into two types. **i**, Most cell types can be distinguished by their strong connections. Shown is the proportion of unique combinations of connections across all types for the indicated number of top-ranked connections.

type B, but not the same individual type B cells. Therefore, we used an iterative process in which neurons were first preliminarily typed for use in connectivity (by type) analyses, which then confirmed or refuted the initial type assignment. In practice, connectivity-based clustering is already useful when only a minority of neurons are named, and accuracy increases as more cells are typed. Indeed, with our final cell typing, all individual cells of the 15 columnar neurons were precisely sorted into distinct connectivity-defined clusters (Fig. 2d).

When applying connectivity-based clustering with a preselected number of clusters to larger groups of cell types (for example, medulla intrinsic cells; Extended Data Fig. 3), we encountered cases of (putative) cell types that clustered with different cell types or were split into multiple types. Such cases were resolved using additional criteria such as cell morphology or, when available, comparisons to expression data from split-GAL4 lines. For example, Dm6 and Dm19 cells have similar connectivity but different arbour sizes and soma locations (attributes often indicative of their development[32]; Extended Data Fig. 3a,b) and are labelled by genetic driver lines with expression restricted to only one type. Our collection of cell-type-specific driver lines provided important clues for typing dozens of cell types.

Similar to the cell types of the mouse retina, the populations of repeating neurons in the fly visual system are also organized as mosaics to achieve uniform coverage. To illustrate this biologically and methodologically important property, we considered two related amacrine-like cells of medulla layer M9: Pm2a and Pm2b. Although Pm2a and Pm2b have similar morphologies (Fig. 2e,f), we found evidence for two populations that each separately cover visual space and have clear connectivity differences (Fig. 2g,h). We used similar analyses of repeating neurons to decide whether to divide groups into finer type categories (Extended Data Fig. 4).

Some challenging cases could only be resolved by using multiple types of information: connectivity, morphology, size, spatial distribution and cell-body position. For example, in a previous analysis of R7 and R8 photoreceptor targets in the colour pathways, individual neuron morphologies were idiosyncratic, such that we could not confidently sort all Tm5 neurons into types[23]. The completeness of the new connectome data overcame these issues. For example, we identified independent sets of connected neurons that enabled us to separately cluster Tm5a, Tm5b and Tm29, as well as Dm8a and Dm8b, into overlapping mosaics (Extended Data Fig. 5).

Our cell-clustering results demonstrate how connectivity places tight constraints on cell typing. We found that >99% of cell types can be distinguished by their top five connections (Fig. 2i). Although this method does not robustly classify individual neurons, it captures essential properties of cell types and is included in our cell-type summaries (Supplementary Fig. 1).

## Visual system architecture from connectivity

The regions of the fly visual system, like the mammalian neocortex, have been further divided into layers where connections between distinct neuron populations occur. Taken together, these layers and the approximately orthogonal columns that convey retinotopy provide a representation of visual space and a framework for describing visual neurons. Indeed, the distributions of arbours of many fly visual neurons provide strong clues about their function[6,9,33,34]. We therefore developed a computational infrastructure using principal neurons as scaffolds to delineate the layers and columns of the visual system.

We first built a coordinate system in the medulla by assigning each of the 15 columnar neurons (Fig. 2a) to hexagonal coordinates (following the approach developed for the full adult fly brain (FAFB) dataset[35]). The coordinates correspond one-for-one to lenses on the compound eye (except for some lenses on the edge). We also identified coordinates along the equator of the eye, a global anatomical reference (Fig. 3a). A complete set of the columnar neurons was assigned to most coordinates, whereas some incomplete sets were found mainly at the boundary of the medulla (Fig. 3a and Supplementary Table 3).

We then used the synapses of all neurons assigned to each medulla coordinate to create volumetric columns for the medulla and lobula (Fig. 3b and Extended Data Fig. 6a). Lobula-plate columns were built by assigning sets of T4 neurons to each Mi1 neuron (Extended Data Fig. 6b,c and Supplementary Table 4). Together, these sets of corresponding neurons were used to extend the same coordinate system to all three major neuropils. Volumetric layers were constructed by bounding depth positions along columns, calibrated using benchmark cell types (visualized alongside LM examples; Fig. 3c). These volumes established an 'addressable' visual system in the accompanying neuPrint database. They can be visualized together with the greyscale EM data (Fig. 3d) or with reconstructed neurons (Extended Data Fig. 6c,d). These tools enable database queries that simplify otherwise complex analyses, such as the spatial distribution of synaptic connections across optic lobe neuropils or as a function of depth (Fig. 3e and Extended Data Fig. 7). Notably, the columns and layers facilitate quantification of connectivity and morphology, which form the core of our comparative dataset (Figs. 4 and 5).

## Neurotransmitters in the visual system

The functional sign of each synaptic connection, whether excitatory or inhibitory, depends primarily on presynaptic neurotransmitter expression and on the receptors expressed by postsynaptic cells. The identification of neurotransmitters expressed by each neuron has been challenging and relies on genetic techniques such as reporter expression or antibody labelling. Moreover, we do not have robust methods for the detection of modulation by neuropeptides.

Recently, more reliable and scalable methods have been developed for assigning neurotransmitters. The measurement of neurotransmitter-synthesizing enzyme expression in cells using fluorescence in situ hybridization (FISH)[36,37] or RNA sequencing[38–40] has enabled the identification of neurotransmitters for dozens of cell types in the fly visual system. Although EM data do not give direct information about neurotransmitters, recent progress in the application of machine learning to classify neurotransmitters from small image volumes around presynapses (Fig. 4a) has shown high accuracy[41]. We applied these methods to the visual system EM dataset, in which the

many repeating cell types and substantial previous data on neurotransmitter expression provided an extensive scale of training data and basis for evaluating prediction accuracy.

We trained a neural network[41] to classify synapses to one of seven neurotransmitters (acetylcholine, glutamate, GABA, histamine, dopamine, octopamine and serotonin) using known neurotransmitters of 59 cell types, contributing nearly 2 million synapses as training data (Fig. 4b). The per-synapse accuracy of the trained network showed a modest improvement compared with predictions for hemibrain and FAFB datasets[41]. To further validate the predictions, we collected new data on neurotransmitter expression for 66 additional cell types, using FISH and expansion-assisted iterative fluorescence in situ hybridization (EASI-FISH)[36].

As an example, we present a complete anatomical group of neurons: the TmY cells (Fig. 4c). Three out of the 16 cell types (indicated by asterisks in the figure) were included in the training data, and the neurotransmitters expressed by 6 types were identified with new experimental data (labels in bold in Fig. 4c, experimental data in Fig. 4d). The fraction of the top neurotransmitter prediction was high, whereby 14 out of 16 TmY cell types had >85% synapses classified as a single neurotransmitter. The consistency and accuracy of predictions were comparable between the cell types included and those not used in training (Fig. 4c). Our experimental expression data were consistent in all six cases with the top predictions.

The synapse-level prediction accuracy is shown for our validation dataset of 78 cell types not included in training the neural network (Fig. 4e and Supplementary Table 5). Accuracy was further improved when these predictions were aggregated across all cells of each type. Despite this notable performance, there are limitations to the reliability of these predictions. Many cell types, including >100 VPN types, have limited presynapses in our data volume. Another approximately 50 cell types, all with low cell counts, had predictions for octopamine, dopamine or serotonin, which far exceeded the number of optic-lobe-associated cell types estimated to express these neurotransmitters[37]. Precisely because of the scarcity of neurons confirmed to express these neurotransmitters, they are not well represented in our training data; accordingly, these predictions are less reliable. The consensus neurotransmitter is reported as 'unclear' in the neuPrint database for cell types with low-confidence predictions unless independently confirmed by experimental data.

Integration of the consensus neurotransmitters with the connectivity data was informative. Approximately half of the cell types are cholinergic and are probably excitatory, and half express GABA or glutamate and are probably inhibitory. Excitatory neurons make up the majority of the neuropil connecting ONCN and VPN types, whereas the ONIN and VCN types are heavily skewed towards glutamatergic, GABAergic and aminergic cells (Fig. 4f). There are also differences in neurotransmitters between the layers in each neuropil (Fig. 4g). Furthermore, the largest neurons tend to be glutamatergic or GABAergic and therefore probably inhibitory (Fig. 4h). We also examined the synaptic 'fan-out' of visual-system neurons; that is, the average ratio of output connections to presynapses. There was a significant difference across the neurotransmitters, whereby the average fan-out of GABAergic neurons was lower than glutamatergic or cholinergic neurons, participating in about 13% fewer connections (analysis of covariance (ANCOVA): acetylcholine versus GABA, $P < 0.001$; glutamate versus GABA, $P < 0.001$; acetylcholine versus glutamate $P \approx 0.83$; Fig. 4i). This difference may reflect a more targeted role for inhibition in visual circuits or a higher efficacy of GABAergic synapses.

Most neurons seemed to release the same neurotransmitter (or neurotransmitters) across their various terminals, and molecular profiling indicated that most neurons signal with a single dominant neurotransmitter. However, previous experiments[38] have established two clear examples of co-transmission in visual neurons: Mi15 neurons express markers for both dopamine and acetylcholine, and R8 express markers for histamine and acetylcholine (Supplementary Table 5).

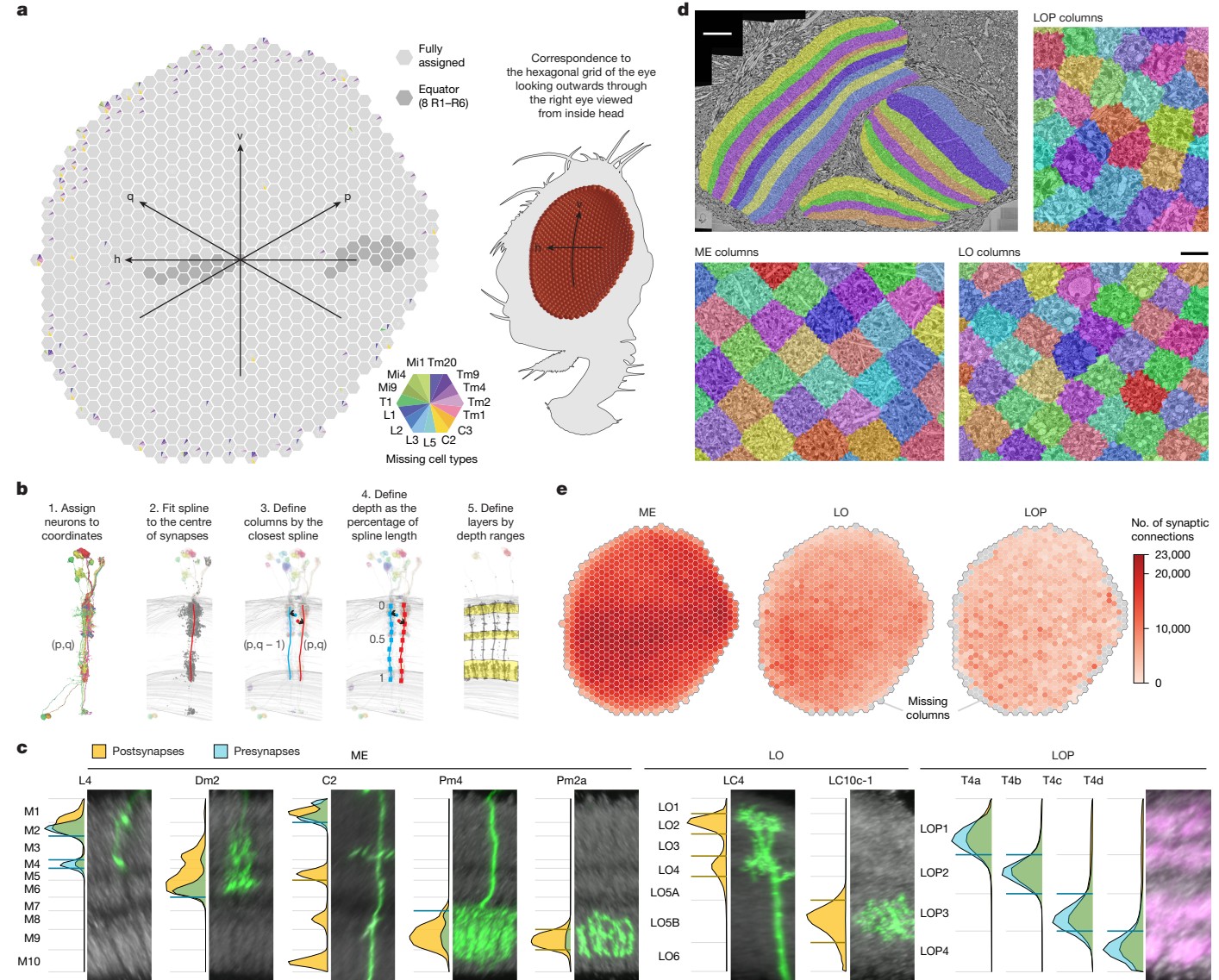

**Fig. 3 | Capturing the architecture of the visual system by analysing the connectivity of key cell types. a**, The lenses of the fly eye form a hexagonal grid and are mapped onto a hexagonal coordinate system in the medulla. p and q denote hexagonal coordinates; h and v indicate the horizontal and vertical axes of the eye, respectively. The darker hexagons correspond to locations along the equator of the eye (determined by counting photoreceptors in the corresponding lamina cartridges). The 15 columnar neuron types in Fig. 2a were assigned to each coordinate. Completely grey hexagons indicate a medulla location that has been assigned a set of all 15 neurons (Methods and Supplementary Table 3). The colour-coded wedges indicate cells of a type missing at that medulla location, most of which are along the edge. **b**, Schematic of the process used for creating columns and layers. Lobula-plate columns were based on sets of T4 neurons assigned to each Mi1 (Extended Data Fig. 6). **c**, Layer boundaries defined on the basis of the synapse distributions of marker cell types, as established from LM images. For each type, we show the distribution of presynapses and postsynapses across depths together with LM images of single-neuron clones (in green; rotated and rescaled to match the top and bottom of the neuropil in light grey). The lobula-plate image shows the neuropil (grey, nc82-antibody) and the axon terminals of the T4 neurons (magenta). The horizontal blue and orange lines indicate the distribution cut-off around a peak that defines each boundary. The collection of these parameters defines all layer boundaries, shown in grey. **d**, Layers and columns shown as volumes superimposed on greyscale EM data. Scale bars, 20 μm (top left) and 5 μm (bottom right). **e**, The spatial distribution of postsynapses by neuropil and column (Extended Data Fig. 7).

We identified a prediction of acetylcholine for Mi15 and histamine for R8, with limited predictions for other transmitters. Several visual neurons are peptide-releasing cells, such as l-LNv cells[42], and EM-based prediction methods have not yet been developed for peptidergic transmission or co-transmission.

## Quantified anatomy and connectivity

The representation of optic lobe neuropils as a coordinate system of layers and columns (Fig. 3) enabled a compact, quantitative summary of the innervation patterns and connections of all visual neurons. Like a fingerprint, these summaries identified most cell types and were broadly applicable for comparisons between cells and cell types in our dataset and with neurons imaged by LM and reconstructed in other EM volumes. These comparisons can be extended to other insect species owing to the deep conservation of the optic lobe ground plan.

We illustrate these summaries with a group of connected neurons that includes the three Dm3 types (Fig. 5a,b). Each Dm3 type forms a pattern of stripes with a distinct orientation across the medulla (Fig. 5c). Each type provides substantial input to a distinct TmY type, which itself shows oriented medulla branches that are approximately orthogonal to

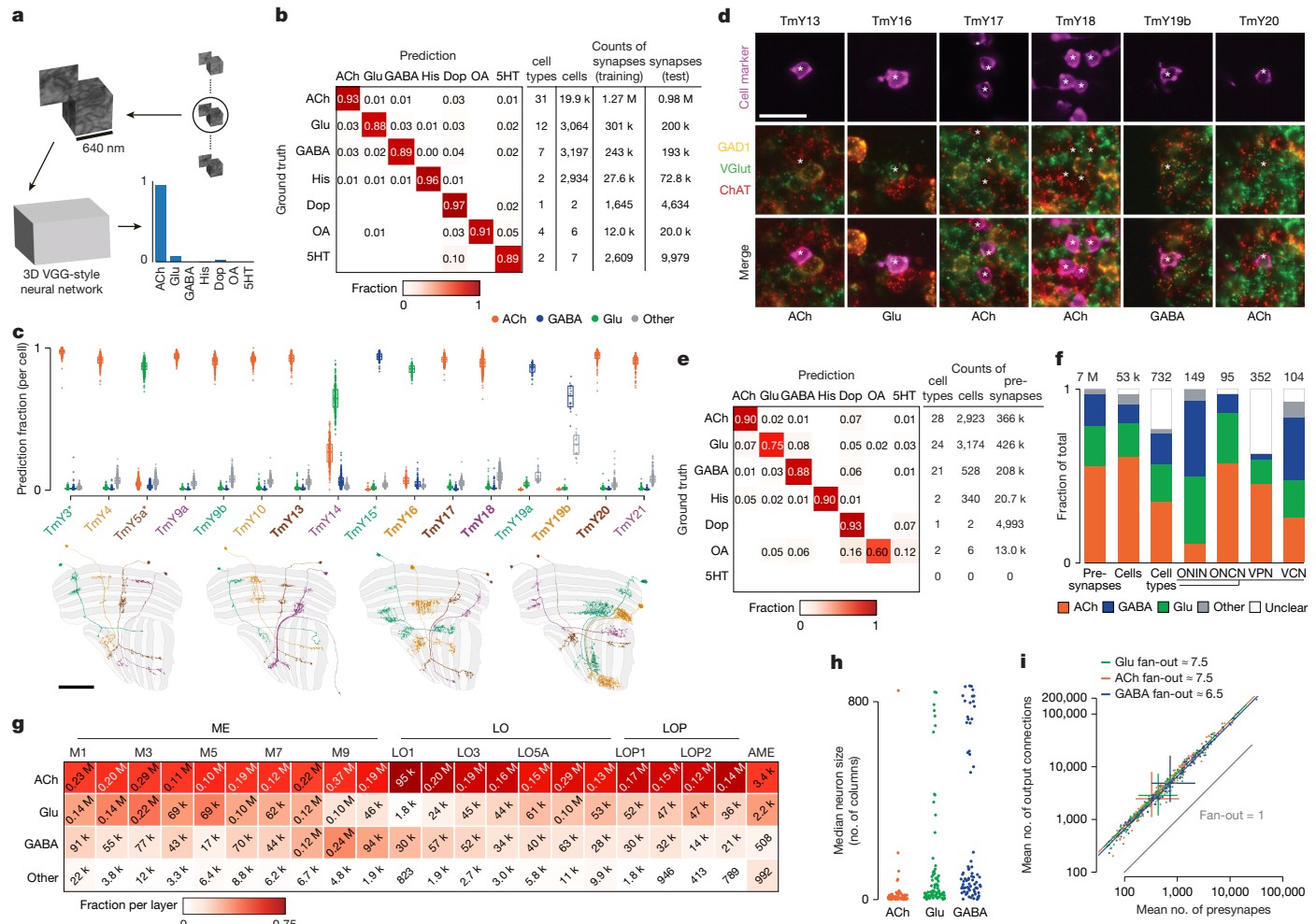

**Fig. 4 | Diversity of neurotransmitter signalling in the visual system.**
**a**, Schematic of the process for predicting neurotransmitters. Using previous neurotransmitter expression measurements in 59 cell types as ground-truth data, a visual geometry group (VGG)-style deep neural network[41] was trained to classify the following presynaptic neurotransmitters: acetylcholine (ACh), glutamate (Glu), GABA, histamine (His), dopamine (Dop), octopamine (OA) and serotonin (5HT). **b**, Performance of the per-synapse predictions evaluated on held-out synapses. Cell types, cells and synapses used for training and testing are tabulated. **c**, Synapse predictions were aggregated for cell-type-level consensus neurotransmitter assignments. Individual points represented the predicted fraction per cell; boxes indicate the median and quartiles. Here 16 TmY cell types are shown. Example morphologies are below. Three cell types (indicated by asterisks) were included in the training data, and predictions for six types (names in bold) were confirmed using new validation data. Scale bar, 50 μm. **d**, Images of driver lines for the six TmY cell types that were assayed for neurotransmitter marker gene expression[36,37], showing the cell marker for

GAL4 (top), markers for ChAT, VGlut and GAD1 (middle), and merged images (bottom), and the assigned neurotransmitter below. Asterisks mark specific cell bodies as reference points to facilitate comparisons across the three rows. Scale bar, 20 μm. **e**, Performance of neurotransmitter predictions for 78 cell types evaluated using experimental data (Supplementary Table 5). **f**, Neurotransmitter types counted for synapses, cells, cell types and cell-type groups. In these and subsequent summaries, synapse-level predictions are inherited from consensus (cell-type-level) neurotransmitters. Cells with limited predictions were scored as 'unclear'. **g**, Spatial distribution of presynapses with neurotransmitter classifications. The colour scale is normalized for each column. **h**, Median size for ONIN and ONCN cell types. **i**, Linear regression estimates of synaptic fan-out, the cell-type averaged ratio of output connections to presynapses, grouped by neurotransmitter. The whiskers indicate the median and quartiles for each transmitter. The fan-out values are significantly different between ACh and GABA ($P < 0.001$) and Glu and GABA ($P < 0.001$), but not between ACh and Glu ($P ≈ 0.83$), ANCOVA.

those of the upstream Dm3 type (Fig. 5c, bottom). Although previous LM studies have described Dm3 cells, also known as line amacrines, and their oriented arbours[2,3,19], only a comprehensive EM reconstruction and annotation fully revealed this network. Although we focused here on presenting our cell-type inventory of the visual system, we note that the intriguing structure of the Dm3–TmY network has strong implications for function, as recently noted[43].

To simplify the exploration of the circuits in our visual-system inventory, we created compact summaries of defining features of each cell type. In the example Dm3–TmY network shown in Fig. 5b, the summaries include the number of cells of that type, consensus neurotransmitter prediction, mean distribution of presynapses and postsynapses (left), the top five connected input and output cell types (middle), and the

neuron size measured as the mean number of innervated columns (right). The top input and output connections of the example neurons capture key features of this circuit. For example, each Dm3 type receives excitatory input from Tm1 neurons and provides glutamatergic (probably inhibitory) output to the other Dm3 types and one of the TmY cells. Notably, the three TmY types are the top inputs to LC15, a VPN selective for moving long, thin bars[44]. By contrast, LC11, which prefers small objects[45,46], is targeted by a large glutamatergic interneuron (Li16), another major target of TmY cells. This result suggests that there is a potential opponent processing step that tunes the selectivity of these feature-detecting LC neurons.

We quantified the detailed morphology of neurons as the synaptic distribution and column innervation, by depth, in the optic lobe

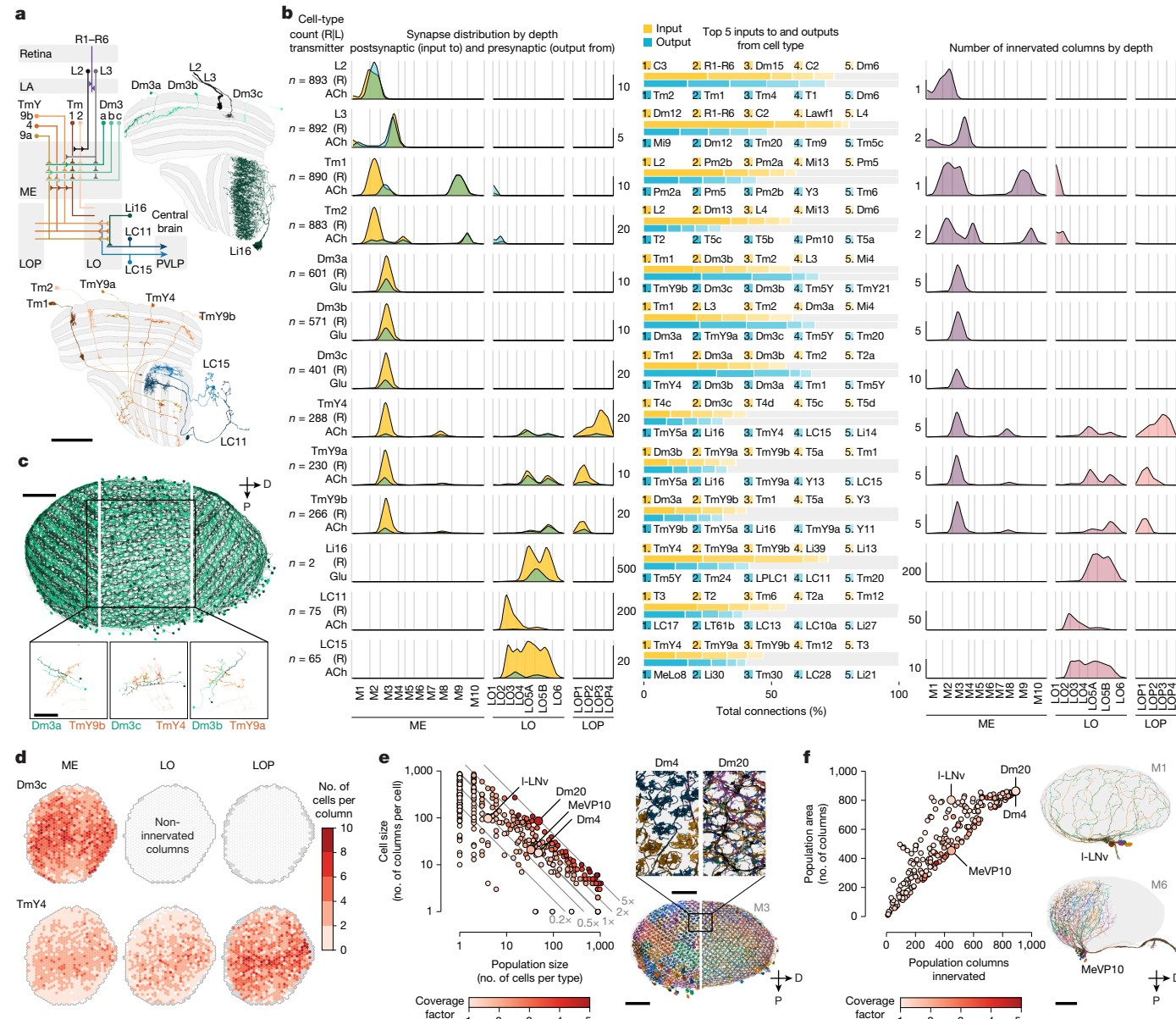

**Fig. 5 | Quantitative summary of anatomy and connectivity of visual system neurons. a**, Examples of 13 cell types associated with the Dm3 line amacrine neurons, including their major inputs and outputs (summarized in a circuit diagram). Scale bar, 50 µm. PVLP, posterior ventrolateral protocerebrum. **b**, Each row presents the quantified summary for each cell type. Far left, number and location (right or left (R|L) hemisphere) of neurons and consensus (see Methods) neurotransmitter prediction. Left, distribution of presynapses and postsynapses across neuropil layers. Right, the top five connected input and output cell types by contributed connectivity (colour shade indicates rank). Far right, cell size measured by depth (mean number of innervated columns). Data are scaled by row, indicated with vertical scale bars. The visual cell types are summarized in the Cell Type Catalogue (Supplementary Fig. 1). **c**, The stripe-like patterns of Dm3 types each cover the entire M3 layer, shown here in separate panels and neurons coloured by their dominant column coordinate. Each inset shows two individual Dm3 cells of one type from adjacent coordinates and selected connected TmY cells. Scale bars, 40 µm. **d**, The number of Dm3c and

TmY4 cells innervating each column as a spatial distribution by neuropil. Similar plots for all visual cell types are in the Cell Type Explorer web resource. **e**, Left, the relationship between the number of cells (population size) and the average number of columns innervated by each cell (cell size) in the medulla (for types with >50 synapses and >5% of total connectivity therein). Colour coded by the coverage factor (per-type average number of cells per column). The 1× line indicates onefold coverage, and cell types above or below cover the whole medulla with more or fewer neurons per column, respectively. Right, selected cell types show coverage factors for different tiling arrangements: Dm4 (about 1) and Dm20 (about 5). Scale bars, 10 µm (top) and 50 µm (bottom). **f**, The per-cell-type density of medulla coverage, comparing the column innervation of the population to its convex area. Types close to the diagonal (for example, MeVP10) densely cover the medulla, whereas types above the diagonal (for example, l-LNv) feature sparser coverage. Medulla layers are shown face-on. Scale bar, 50 µm.

neuropils. The classical layer boundaries are shown as a reference (vertical lines in Fig. 5b), but the synaptic distribution and column innervation are presented with higher depth resolution because these established neuroanatomical features do not fully capture the

organization of the neurons. An essential test of whether these measurements accurately profile the distinctive morphology of each cell type is their usefulness in sorting cells. The synapse-by-depth measurements, compiled into per-neuron feature vectors, was able to sort most

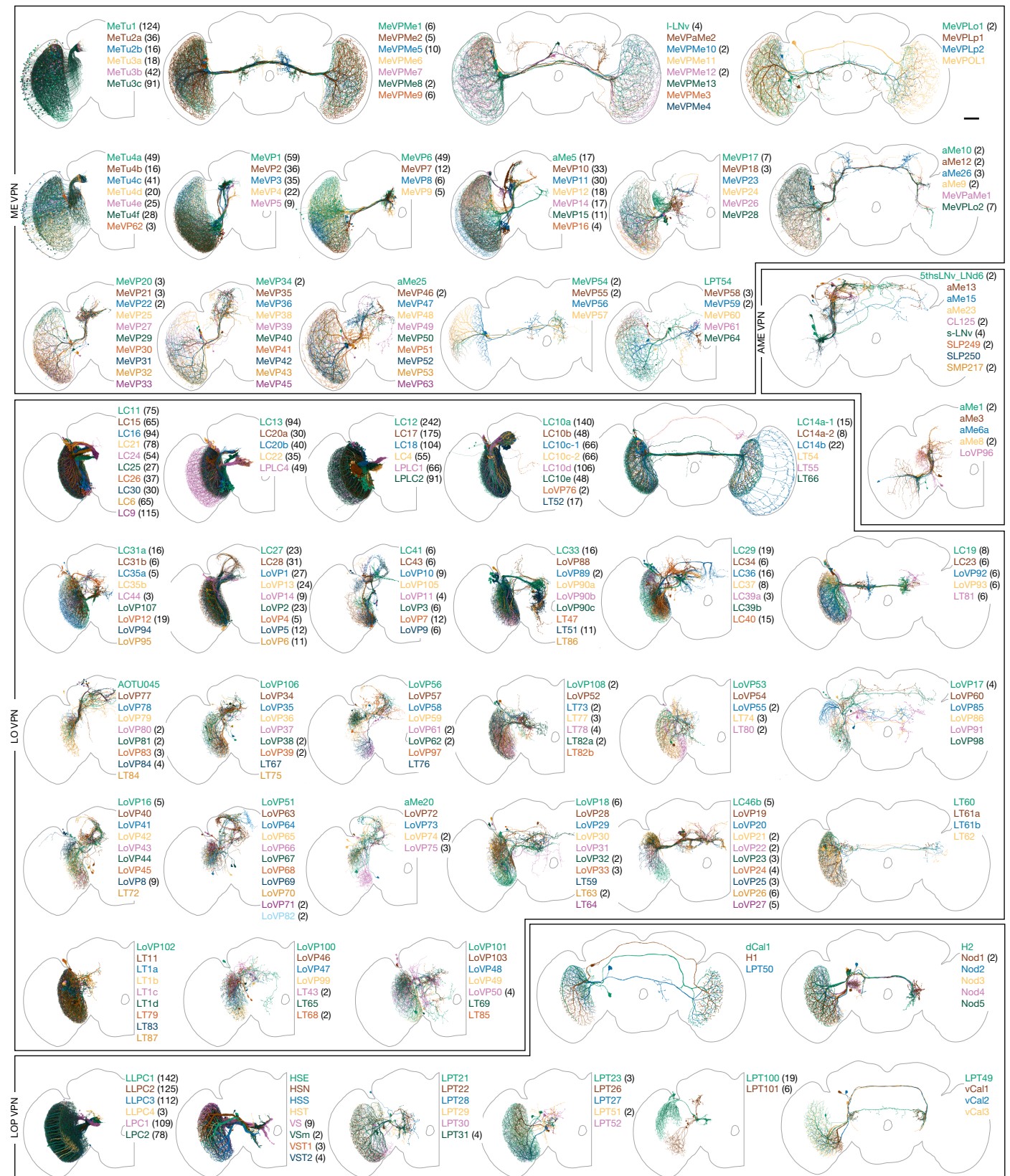

**Fig. 6 | Visual projection neurons.** All VPNs that connect the right optic lobe with the central brain or the contralateral optic lobe. We first divided the approximately 4,500 VPNs into the 352 types shown here, and then placed them into 51 groups of morphologically similar types. These groups are based on the main neuropil they receive their optic lobe inputs in, whether they project to the ipsilateral or contralateral central brain or the contralateral optic lobe, and other aspects of their morphology. Unilateral cells, neurons without projections that cross the midline, are mainly shown in half-brain panels, and most cells with arbours in both brain hemispheres are shown in full-brain views. Each panel includes the names of the cell types (colour-matched to the rendered neurons) and the number of individual cells of each type (in parentheses). Types without a number are present once per brain hemisphere. Detailed morphology of the optic lobe neuropils can be found in the Cell Type Catalogue (Supplementary Fig. 1). Scale bar, 50 μm.

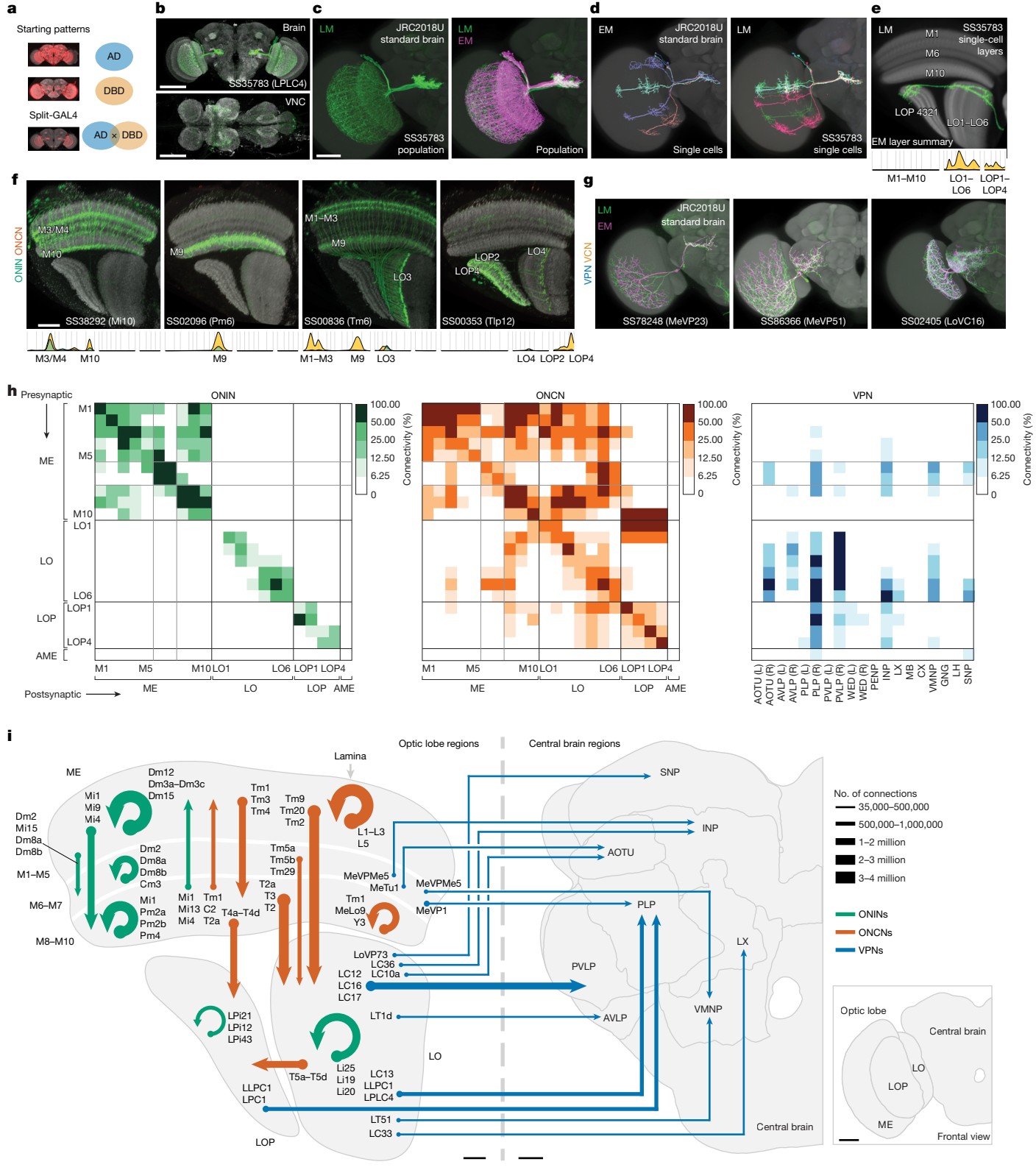

**Fig. 7** | See next page for caption.

medulla neurons into their respective types (Extended Data Fig. 8), only slightly underperforming connectivity-based clustering (cluster completeness 0.87 versus 0.9; Extended Data Fig. 3). Conveniently, this method does not require the pre-assignment of cell types beyond

those used to define columns. Cell-type summaries, along with a gallery of example neurons for all 732 types, are provided in Supplementary Fig. 1. We also developed interactive web pages that summarize all the connectivity of visual neurons that serve as a browsable companion to

**Fig. 7 | Genetic driver lines for targeting visual-system neuron types and summary of connectivity. a**, The genetic intersection of the expression patterns of two hemidrivers (AD and DBD) produces a split-GAL4 driver line selective for cells of interest. We report 582 split-GAL4 lines matched to >300 EM-defined cell types (Supplementary Table 6). **b**–**e**, LM features used for LM–EM matching. **b**, Pattern of a split-GAL4 line expressed in LPLC4 VPNs, with some off-target VNC expression. Scale bars, 100 μm. **c**, LM image of LPLC4 neurons (driver as in **b**) registered to a standard brain (JRC2018U, grey) (left) or overlaid with registered skeletons of all EM-reconstructed LPLC4 neurons (right). Scale bar, 50 μm. **d**, EM (left) and LM (MCFO[19], right) images of LPLC4 cells. **e**, Segmented, registered LM image of an MCFO-labelled LPLC4 neuron with a slice of the template brain (showing a different view than **b**–**d** to emphasize the layer patterns) above a summary (Fig. 5b) of the innervation of LPLC4. **f**, Selected split-GAL4 lines for ONINs and ONCNs. Layer patterns of genetically labelled cell populations with a neuropil marker (anti-Brp). Corresponding layers are indicated on the LM images and EM summary figures (see **e**). Scale bar, 20 μm. **g**, Selected split-GAL4 labelled VPNs and VCNs (overlays with registered EM skeletons). **h**, Connectivity between regions, including optic lobe layers, quantified as a weighted sum (Methods) for the ONIN, ONCN and VPN groups. The colour code reflects contributions to total connectivity. Inter-region connections were ranked, binned and coloured by their contribution to the cumulative sum. Standard brain region names are abbreviated and grouped (see Methods and ref. 61). **i**, Schematic of the main conduits for visual information flow in and between brain regions based on **h**. Arrows are scaled with the number of connections between regions and represent entries in the highest 50% (darkest colours), plus a few prominent connections below this threshold. Major contributing cell types are indicated for each arrow, and optic lobe layers and several central brain regions are grouped in this summary. Scale bars, 10 μm (optic lobe), 20 μm (central brain) or 50 μm (inset).

the neuPrint web interface. The *Drosophila* Visual System Cell Type Explorer is previewed in Extended Data Fig. 9.

To survey the distribution of the visual neurons and their connections, we measured spatial coverage across optic lobe neuropils (summed across layers; Fig. 5d). The example spatial coverage maps for Dm3c and TmY4 in the medulla showed a common trend of increased neuron density near the equator of the eye. Dm3c is a medulla intrinsic cell without processes in other neuropils, whereas the coverage map for TmY4 showed a notable trend of increasing overlap in the lobula plate (the Cell Type Explorer web resource contains maps for all cell types). This spatial analysis links the coverage of each cell type for retinotopic brain regions to their sampling of visual space. Figure 5e relates the cell number per type, the size in columns and coverage factor (the average number of cells per column) for the medulla (lobula and lobula plate plots are shown in Extended Data Fig. 10; interactive plots are available from the Cell Type Explorer web resource). As expected, cell numbers were inversely related to size, a finding consistent with uniform visual coverage. However, there are notable differences from this trend, such that cell types above the 1× line had higher coverage factors, with more cells than required to cover the neuropil. For instance, Dm4 and Dm20 both have 48 cells, but Dm20 cells are much larger and overlap nearly 10 cells per column, whereas Dm4 neurons neatly tile medulla layer M3 without overlap. Neurons below the 1× line exhibited partial coverage of the medulla, either as dense, regional coverage, illustrated by MeVP10 cells, or broad but patchy coverage, illustrated by l-LNv cells (Fig. 5f and Supplementary Video 3). These summary data, along with the inventory in Supplementary Fig. 1 and the Cell Type Explorer web resource, provide a comprehensive starting point for exploring circuits and for designing experiments.

## Specialized cell types

Our complete survey highlighted specialized sets of visual neurons defined by their anatomy and expected functional roles. The dorsal rim area (DRA) is a zone of the eye where photoreceptors are specialized for detecting polarized light[23]. Specialized medulla neurons integrate these R7d and R8d photoreceptors (Extended Data Fig. 11). Notably, we did not find DRA-specific ONCNs and therefore no specialized DRA-recipient cells in the lobula.

The accessory medulla is a small neuropil at the anterior–medial edge of the medulla, with a well-established role in clock circuitry and circadian behaviours[42,47]. Unlike the other visual brain areas, the accessory medulla does not exhibit obvious retinotopy, which provides support for the view that photoentrainment of circadian rhythms should not require detailed visual–spatial information. Nevertheless, the diversity of neurons with substantial connectivity in the accessory medulla (Extended Data Fig. 12) indicates an underappreciated complexity and broader roles of this region in sensory integration and behaviour.

## Communication with the central brain

VPNs represent a remarkable compression of information. Signals from nearly 50,000 local neurons that interpret the visual world are conveyed by about 4,500 neurons of 352 cell types to the central brain (Fig. 1e and Supplementary Table 1). Many VPN types, including the lobula plate tangential (LPT)[33], the lobula columnar (LC)[45] and the medulla-tubercle (MeTu)[48] neurons, have been described in detail. Nevertheless, we identified new types and further refined previous classifications in these well-studied groups. Among the VPNs, the small field-projection neurons of the lobula and lobula plate—LCs, LPCs, LLPCs and LPLCs—comprise around 3,000 cells, whereas MeTu neurons contribute an additional 500 or more. The remaining cells were morphologically varied and target many brain regions. Figure 6 shows all VPN types, illustrating the diversity of pathways that relay vision to central behavioural control areas. The neurons are grouped by anatomical similarities, primarily on the basis of their input optic lobe neuropil. Detailed connectivity and morphology summaries are provided in Supplementary Fig. 1.

We identified 104 VCN types across about 270 cells (Extended Data Fig. 13). Most types consisted of a single neuron. However, several populations of smaller-field inhibitory neurons targeted the lobula plate. Larger VCNs included several octopaminergic cells[49] that are probably important for modulating visual neurons during movement[50]. Many VCNs targeted specific layers of the medulla (mostly called MeVC neurons), the lobula (LoVC and some LT neurons) and the lobula plate (called LPT neurons for historical reasons). One of these, LoVC1 (also known as IB112)[8], gates the flow of specific channels of visual information during social behaviours[51]. Although most VCNs are presumed inhibitory (glutamatergic or GABAergic), it is noteworthy that most MeVC neurons are predicted to be cholinergic. Extended Data Fig. 13 presents the complete set of VCNs to support systematic study of these understudied cells and their roles in the central modulation of visual processing.

## Cell-type-specific genetic driver lines

Analyses and simulations of connectomes can implicate functions for many of its constituent cell types[33,43,52]. Testing these predictions requires genetic tools to manipulate or mark specific cell types for functional imaging or electrophysiology. Using the split-GAL4 intersectional method[53,54], we developed genetic driver lines that target cell types of the *Drosophila* visual system. By analysing the morphology of neurons expressed in first-generation driver lines, we developed several collections that match groups of known cell types in the lamina[21], the medulla[26] and the lobula[45]. These driver lines enabled targeted transgene expression for functional experiments, anatomical analyses[19] and high-resolution transcriptome measurements[38]. Matching driver lines to well-defined cell types can be complex, but LM images of single

neurons and populations, combined with classical Golgi surveys[1,3], provided a basis for matching most high-count ONCN and ONIN types. For example, we did not find new Dm types in the connectome compared with a previous LM study[19], except for a new division of Dm3 cells and the dorsal rim types (Fig. 5c and Extended Data Fig. 11). However, the full visual system inventory (Supplementary Fig. 1) is the ideal reference for matching a broader set of driver lines, particularly the less numerous cell types. Here we document a collection of 582 split-GAL4 lines (Fig. 7a, Extended Data Figs. 14 and 15 and Supplementary Table 6) matched to >300 EM-defined cell types across all inventory groups.

We found matches for nearly all visual-system cell types labelled by split-GAL4 lines, a result that highlights the completeness of our EM inventory. This finding is notable, as most LM images are from female flies, whereas the EM reconstruction here is of a male fly. The high degree of consensus suggests that most optic lobe cell types are not sexually dimorphic. We matched >98% cell types in our inventory (accounting for 99.8% of cells) to cell types in the female FlyWire FAFB dataset[15–17,24] (Extended Data Fig. 16 and Supplementary Table 7), thereby providing strong evidence for the completeness of the inventory and the absence of major cell-type-level sexual dimorphism. Unmatched types are candidates for sexually dimorphic neurons, and Tm26 is one such example (Extended Data Fig. 16c). A second illustrated example seems to be a rare case of a missing cell type: LoVP109 (also known as LTe12; FlyWire), is absent from the right optic lobe, but LM analysis suggests that it is not dimorphic (Extended Data Fig. 15g).

Figure 7b–e shows an example of morphological matching of the expression pattern of a split-GAL4 driver, which selectively labels a single population of visual neurons, to EM-reconstructed LPLC4 cells. We transformed the EM-reconstructed neurons into the same reference brain volume[55] as registered LM data, which enabled us to compare single-neuron morphology and population patterns (Fig. 7c,d). In most cases, the quantified optic lobe neuropil innervation (Supplementary Fig. 1) provided a simple comparison that was sufficient for confident matching (Fig. 7e for LPLC4), particularly for ONIN and ONCN lines (Fig. 7f and Extended Data Fig. 14a). We matched many VPN and VCN driver lines to EM-defined cell types through their central brain arborization patterns (visualized on a co-registered brain; Fig. 7g and Extended Data Fig. 14b). When neurons labelled by a driver line seemed to match multiple EM-defined cell types, we used finer differences in the layer innervations and features such as cell-body distribution, regional pattern coverage and arbour size and shape (Extended Data Fig. 15a–e) to refine the match. Our LM images often confirmed the accuracy of the EM-reconstructed morphologies, even for atypical neuron morphologies that seemed to represent real but rare variants (Extended Data Fig. 15f). By linking genetic driver lines to EM-defined cell types, this collection establishes a valuable toolkit for analysing visual system circuits.

## Inter-region connectivity

Understanding the flow of visual information throughout the fly brain is a substantial undertaking, but the infrastructure we developed here to catalogue neurons of the visual system provides an approach for initial analyses. Through examination of the major cell type groups, we asked which specific connectivity patterns—from particular layers to others across the optic lobe neuropils—are most prevalent. The first matrix in Fig. 7h shows the inter-region connectivity of the ONINs. As expected, we found a block-diagonal structure with prominent within-layer connections indicated by the high connectivity along the diagonal. The organization of the connections of the medulla supported the anatomical division into three units of more tightly interacting layers and the separation of interneurons into Dm (distal, M1–M5), Cm (central, M6–M7) and Pm (proximal, M8–M10) types. The summarized connectivity matrices highlighted major pathways and indicated representative cell types that substantially contribute to the strongest connection (Fig. 7i). ONCNs have a denser connectivity matrix than the ONINs,

with nearly every layer connected to every other layer. However, there were more substantial connection patterns typified by prominent neuropil-connecting cell types (highlighted in Fig. 7i). Notably, connectivity above the diagonal was much higher than below, a finding that indicated feedforward flow, for example, from the medulla to the lobula and the lobula plate. Finally, the connectivity of VPNs to the central brain reflected the broad classes of projection neurons (Fig. 6), with prominent connections from multiple optic lobe neuropils often targeting the same central brain regions (Fig. 7h,i).

Notably, many central brain regions, such as the central complex, do not receive prominent, direct visual projections. This result is partly due to the data presented in Fig. 7h,i only showing the top 93.75% of connections. For example, specific Kenyon cell types in the mushroom body that respond to visual information receive direct VPN connections[56,57], but are not shown here. Visual signals also arrive through central brain interneurons, as found for the mushroom bodies[56], thereby further expanding the footprint of vision in the central brain. Aggregation of visual neurons into this 'projectome' highlights the most prevalent connections to major brain regions. Furthermore, all connections are dissectible down to the individual cell types (Fig. 7i), which can be quickly queried using the neuPrint database. Together with the extensive collection of driver lines (Fig. 7a–g and Supplementary Table 6), we now have a comprehensive toolkit for exploring this entire visual system and discovering how visual features are detected and where vision is integrated in the central brain.

## Concluding remarks

Our survey has been exhaustively curated and proofread from the new connectome of an optic lobe of a male brain. In time, our knowledge of VPNs and VCNs will substantially increase as details of their inputs and outputs are established in the connected CNS. Together with the connectome of a female fly brain[15–17], these datasets provide a valuable resource for understanding how vision is processed in central brain circuits and, ultimately, how it guides behaviour. By mapping neural connections at an extensive scale and resolution, connectomics overcomes the traditional trade-offs between completeness and accuracy in cataloguing cell-type diversity[14]. This approach has been particularly transformative in visual systems, including the mammalian retina and the fly, facilitating detailed investigations of circuit function. Often, analyses of these data directly lead to functional hypotheses. Although scepticism persists about the interpretability of vast datasets or the translatability of structural maps to functional insights, these methods are revolutionizing neuroscience on two levels: the completeness of these surveys enables advanced analytical and modelling studies of integrative brain function, and more straightforwardly, they outline countless experimental roadmaps for investigating circuit function. Pathway analyses directly generate functional hypotheses. For instance, neurons downstream of R7 are strong candidates for mediating ultraviolet-related visual responses[58], whereas those downstream of L3 probably use absolute luminance information[59]. Moreover, neurons downstream of T4 and T5 almost certainly receive directionally selective input[7]. Extending the spatial analysis of neuronal inputs—which was essential for uncovering the role of T4 in motion detection[6]—to larger cells in deeper circuits will be important for advancing our mechanistic understanding of visual transformations, such as those implemented by LC neurons[44]. This complete connectome, paired with our new tools for incorporating visual–spatial coordinates, enables a detailed evaluation of connectivity patterns, which may reveal heterogeneity[29], subtype specificity, as in the targets of pale and yellow photoreceptors[23] (Extended Data Fig. 5), or spatial variations (Extended Data Fig. 4). An exciting avenue for discovery will come from spatially mapping molecular information—such as the identity of neurotransmitter receptors—onto connectomes[38,60]. Together, connectomic surveys, detailed analyses and genetic tools for experimental manipulation

provide the elements necessary for bridging the gap between the structural complexity and the functional understanding of neuronal circuits the field has long sought.

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

[1]Janelia Research Campus, Howard Hughes Medical Institute, Ashburn, VA, USA. [2]University of Toronto, Scarborough, Ontario, Canada. [3]Google Research, Google, Zurich, Switzerland. [4]MRC Laboratory of Molecular Biology, Cambridge, UK. [5]Department of Zoology, Cambridge University, Cambridge, UK. [6]These authors contributed equally: Frank Loesche, Shin-ya Takemura, Laura E. Burnett, Marisa Dreher, Eyal Gruntman, Judith Hoeller, Gary B. Huang, Michał Januszewski, Nathan C. Klapoetke, Sanna Koskela, Kit D. Longden, Zhiyuan Lu, Stephan Preibisch, Wei Qiu, Edward M. Rogers, Pavithraa Seenivasan, Arthur Zhao. ✉e-mail: bergs@janelia.hhmi.org; rubing@janelia.hhmi.org; reiserm@janelia.hhmi.org

## Methods

### EM sample preparation

Sample preparation was performed using previously described methods[8,62], which are similar to those described in a study of the male adult nerve cord (MANC) VNC EM volume[63].

Five-day-old adult *Drosophila melanogaster* males from a cross between Canton S strain G1 × w[1118] were raised on a 12-h day–night cycle and dissected 1.5 h after lights on. The main difference from previous work occurred during this dissection step. We dissected the entire CNS as one unit, including the brain, VNC and neck. The main difficulty, which required many attempts and extreme care, was to dissect a sample without damaging the relatively fragile neck connective. An undamaged neck connective is necessary to reconstruct the entire CNS. The optic lobe reported here is a subset of this complete CNS reconstruction. Isolated CNS samples were fixed in 2.5% formaldehyde and 2.5% glutaraldehyde in 0.06 M phosphate buffer at pH 7.4 for 2 h at 22 °C. After washing, the tissues were post-fixed in 0.5% osmium tetroxide in 0.05 M sodium cacodylate buffer for 40 min and then treated with 1% potassium ferricyanide in buffer for 2 h at 4 °C. After washing, the tissues were incubated with 0.5% aqueous uranyl acetate for 30 min at 4 °C and then post-fixed (second time) in 1% osmium tetroxide in ddH$_2$O for 30 min. This was followed by lead aspartate en bloc staining overnight at 4 °C. A progressive lowering-temperature dehydration procedure started from 1 °C when the tissues were transferred to 10% ethanol. The temperature was progressively decreased to −25 °C while the ethanol concentration was gradually increased to 97%. The tissues were incubated in 0.3% uranyl acetate in ethanol for 32 h at −25 °C. After the progressive lowering-temperature process and low-temperature incubation, the temperature was increased to 22 °C and the tissues were rinsed in pure ethanol followed by propylene oxide, then infiltrated and embedded in Epon (Poly/Bed 812; Luft formulations). Aldehyde-fixed 20% BSA cut into around 1 mm cubes was processed with the fly samples. BSA cubes were then chopped into smaller pieces (about 0.1 mm) after final infiltration. The fly CNS was embedded into 1:1 chopped BSA and pure resin in embedding moulds. This chopped BSA was used as an embedding filler covering the CNS sample to improve the quality of hot-knife slices. These methods optimize morphological preservation, enable full-brain preparation without distortion and increase staining intensity, which facilitates faster focused ion beam milling and scanning electron microscopy (FIB-SEM) imaging. Each completed sample was examined by X-ray CT imaging (Zeiss Xradia Versa 510) to check for dissection damage and other potential flaws and to assess the quality of the staining. The selected sample documented in this study was given the internal name 'Z0720-07m'.

### Hot-knife cutting

The *Drosophila* CNS is roughly 700 μm wide and more than 1,000 μm long, which makes it too large to image by FIB-SEM without milling artefacts. We therefore subdivided the brain right to left (vertical cuts in the orientation shown in Fig. 1a) and the VNC lengthwise (horizontal cuts in the orientation of Fig. 1a), both into 20-μm-thick slabs using an approach developed during the hemibrain project[8]. This enabled imaging of individual slabs in parallel across multiple FIB-SEM imaging machines. During embedding, the CNS was oriented with the long axis of the VNC perpendicular to the block face, with its caudal tip closest to the block face. The block was then trimmed into a fin shape[64] (600 μm wide and several millimetres long in the cutting direction with a sloped trailing edge) to a depth encompassing all of the VNC and half of the neck, leaving the brain in the untrimmed part of the block. This VNC and half-neck were then sectioned into a total of 31 slabs, each 20 μm thick, using our previously described hot-knife ultrathick sectioning procedure[64] (oil-lubricated diamond knife (Diatome 25° angle Cryo-Dry), 90 °C knife set point temperature, 0.05 mm s⁻¹). The remaining part of the block, containing the brain and upper half of the neck, was then reoriented and re-embedded in Epon so that the long axis of the brain was perpendicular to the new block face. This block was trimmed into a fin shape and hot-knife sectioned (cutting speed of 0.025 mm s⁻¹) into 35 slabs, each 20 μm thick. Each slab was imaged by LM for quality control, then flat-embedded against a sturdy backing film, glued onto a metal FIB-SEM mounting tab and laser-trimmed using previously described methods[64]. Each mounted slab was then X-ray CT imaged (0.67 μm pixel size) to check preparation quality, to provide a reference guide for FIB-SEM milling and to establish a z axis scale factor for subsequent volume alignment. All 66 slabs were FIB-SEM imaged separately, and the resulting volume datasets were computationally stitched together (as discussed in the section 'EM volume alignment') and used in the ongoing reconstruction of the entire CNS. The right optic lobe was contained in 13 of these slabs.

### EM volume imaging

We used imaging methods from a study describing the MANC EM volume[63], and were reproduced with minimal modifications to ensure the consistency and accuracy of the reported methods.

The 35 brain and 31 VNC slabs were imaged using seven customized FIB-SEM systems in parallel over almost a year. Unlike the FIB-SEM machines used for the hemibrain project[8], this new platform replaced the FEI Magnum FIB column with the Zeiss Capella FIB column to improve the precision and stability of FIB milling control[65]. FIB milling was carried out by a 15 nA 30 kV Ga ion beam with a nominal 8-nm step size. SEM images were acquired at 8 nm *xy* pixel size at 3 MHz using a 3 nA beam with 1.2 kV landing energy. Specimens were grounded at 0 V to enable the collection of both secondary and backscattered electrons.

### EM volume alignment

The EM volumes were aligned with an updated version of the pipeline used for the hemibrain[8], which was also used for the MANC VNC volume[63]. The 'render' web services were used for serial section alignment of the 35 brain slabs, followed by automatic adjustment for milling thickness variation[66]. The surfaces of the slabs were automatically identified using a combination of a hand-engineered[67] and machine-learning-based cost estimation[63] before graph-cut computation[63,67], followed by manual refinements using a custom tool based on BigDataViewer[68] to correct remaining issues interactively. As with the MANC and hemibrain samples, the series of flattened slabs were then stitched using a custom, distributed method for large-scale deformable registration to account for deformations introduced during hot-knife sectioning. A custom BigDataViewer-based tool was developed to interactively help the automatic alignment process in challenging regions. The code used for the render web services can be found at GitHub (https://github.com/saalfeldlab/render), as is the code used for surface finding and hot-knife image stitching (https://github.com/saalfeldlab/hot-knife).

### EM volume segmentation

Segmentation was carried out as described for the MANC VNC volume[63], but with some differences. Flood-filling network (FFN) inference was only applied in areas that were likely to contain segmentable tissue. To detect these, we used the following heuristic: the CLAHE-normalized images were downsampled to $32 \times 32 \times 32$ nm$^3$ voxel size and then filtered in-plane (*xy*) with a high-pass filter $I_{hp} = I + \min 3 \times 3(255 - I)$, where *I* is the pixel intensity and min $3 \times 3$ is the minimum function, applied convolutionally with a kernel size of $3 \times 3$ pixels. We then computed 2D (in-plane) connected components of the filtered images thresholded at a value of 220 and used the largest connected component in every section to exclude the corresponding parts of the EM images from segmentation.

We trained a semantic segmentation model for the volume by manually annotating selected segments from the FFN segmentation into eight disjoint classes: do-not-merge, glia, trachea, soma,

non-glia–nuclei, neuropil, muscle and glia–nuclei. We used these to train a convolutional neural network model to classify every voxel of the dataset at $16 \times 16 \times 16$ nm$^3$ resolution into one of these classes. For every segment, we then computed the corresponding class using a majority voting scheme and only included soma, non-glia–nuclei and neuropil segments in the agglomeration graph. The neural network used for this process had the same architecture as the FFN (residual convstack) but with all convolutional layers applied in the 'valid' mode.

Instead of manually annotating all nuclei in the dataset, we relied on the predictions of the semantic segmentation model. We computed 3D connected components of voxels classified as glial and non-glial–nuclei and post-processed them twice according to the following procedure: apply morphological erosion with a radius of 5 voxels, recompute 3D connected components and then remove components smaller than 10,000 voxels. We took the centroids of the remaining objects to be nuclei centres and disallowed distinct nuclei to be merged in the agglomeration graph. We applied a heuristic procedure for merging nuclei segments and surrounding soma segments. We applied morphological dilation with a radius of 5 voxels to every nucleus generated in the previous step. We next computed the number of overlapping voxels between these expanded nuclei and FFN segments of at least 10,000 voxels at $16 \times 16 \times 16$ nm$^3$ with a majority class of soma or non-glia–nuclei. All segments matched with a specific nucleus were then merged.

### EM volume synapse identification
We performed synapse prediction using the methods described in the MANC VNC reconstruction[63]. We obtained ground-truth synapse data from the larger CNS sample and used it to train the networks for presynaptic T-bar detection and postsynaptic partner detection[63]. The network weights for T-bar detection were initialized using the previously trained detector[63] and then fine-tuned using the available CNS training data. To quantify the performance of synapse identification in the optic lobe, 14 cubes of $300 \times 300 \times 300$ voxels were randomly selected in the optic lobe volume, and synapses in each subvolume were densely annotated. The subvolumes contained 287 annotated T-bars and 2,425 annotated postsynaptic partners. The overall precision-recall plots for T-bars alone and synapses as units (both components) are shown in Extended Data Fig. 1a. The performance was substantially better than the accuracy achieved in the hemibrain[8] and similar to the performance on the MANC reconstruction[63].

### EM volume proofreading
Proofreading followed similar methods to those documented for the hemibrain[8] and MANC[63] connectomes and used a software toolchain for connectome reconstruction, including NeuTu[69], Neu3 (ref. 70) and DVID[71]. The proofreading process is documented below in greater detail to explain the various stages and the extensive quality control measures we implemented to efficiently produce a high-quality connectome. One difference for this volume is that we automatically generated point annotations for cell bodies (as discussed in the section 'EM volume segmentation'), then added point annotations to neck fibres and nerve bundles at the early stages of the proofreading process. These annotations were fed back into the agglomeration stage, and used to forbid merges between the corresponding segments. This step considerably reduced the number of falsely merged bodies, which are difficult and time-consuming to separate. An important difference in the optic lobe connectome was the role that cell typing played in quality control and setting some proofreading priorities (see the section 'Overview of cell typing'). As we were typing cells from the early proofreading stages, we frequently compared the morphology and connectivity of cells assigned the same types. This step served as a robust check for incomplete shapes or reconstruction errors. In the optic lobe dataset, most cells and synapses belong to repeating cell types, so this parallel cell-typing and proofreading effort was particularly helpful in arriving at a high-quality connectome.

**Goals and strategy.** Our final connectome comprises the set of synaptic connections in which both the presynaptic point and postsynaptic point belong to neurons deemed traced; that is, are acceptably complete. Although capturing 100% of connections would be desirable, this is impossible to achieve in practice owing to time and labour constraints and limitations and artefacts in the data. Our goal was to produce a connectome that captures the largest practical fraction of synaptic connections in the sample while avoiding biases in synapse capture across brain regions and cell types. Furthermore, we aimed to reconstruct the major features of the morphology for each neuron (for example, major arbours and the cell body) to facilitate matching to LM imagery and/or to homologous neurons in the same sample (for example, across hemispheres or visual columns). We used several proofreading protocols to efficiently improve segmentation across multiple dimensions.

**Major phases of proofreading.** Proofreading proceeded in a series of phases, in which the majority of the proofreading team typically worked in the same phase, transitioning together from one phase to the next. Breaking the proofreading process into phases has multiple benefits. Individual proofreader efficiency is enhanced when performing similar tasks in large batches. As most proofreaders work on similar tasks, they can also assist one another and discuss difficult decisions. The reconstruction is easier to manage, coordinate and track. Moreover, subsequent proofreading phases benefit from consistent quality throughout the dataset rather than a more scattershot approach. Another benefit of this phased approach is that no neuron in our dataset is the product of a single proofreader. Different proofreaders inspected all neurons multiple times (and finally by at least one expert neuroanatomist). This means that errors in the segmentation can be identified in multiple phases. As tasks are randomly assigned to proofreaders, errors that survive the proofreading process are not biased against brain regions or cell types. In the final phases of proofreading, a proofreader's tasks may be more correlated with cell types, but in those phases relatively few edits are typically made.

Our main proofreading protocols can be categorized as bottom-up or top-down. Bottom-up protocols (cell-body fibre linking, backbone tracing, focused merging and orphan linking) apply many small improvements throughout the entire sample (or a chosen subregion) without regard to individual neuron identities. In bottom-up protocols, fine-grained tasks are preselected for the proofreader in priority order to for maximum impact on the connectome (for example, capturing synapses). Top-down protocols (cleaving and neuron approval) are targeted at whole neurons. In top-down protocols, a proofreader uses their judgment to guide their attention and address quality issues.

Although our CNS sample contains an entire brain and nerve cord, in the early phases of this project, it had not yet been completely imaged and aligned. We started with just half of the brain, containing the right optic lobe and (most of) the right central brain. The first two phases of proofreading, described below, were performed on the half-brain portion, whereas the later phases were performed after we had access to the complete brain.

**Cell-body fibre linking.** To assist with cell typing and to detect false merges, we began by attaching orphaned cell bodies (somata) to their main axons. This is generally easier than searching for cell-body fibres on neurons lacking their somata. Missing somata are surprisingly common in the initial segmentation. The finer cell-body fibres often run alongside the main axon before merging into it, which presents challenges for the automated segmentation step.

**Cleaving.** Our automated segmentation step is deliberately tuned to tolerate a modest number of false merges that require repairs during proofreading. It is tuned this way because most proofreading effort is expended towards merging falsely split neurites. An automated

segmentation that is tuned to be more aggressive in fixing oversegmentation errors will save work in the long run if the (relatively few) false merges can be repaired with efficient tools. However, false merges in the segmentation impede other phases of proofreading and review; therefore, the cleaving phase aims to eliminate all major false merges in the dataset. We inspected the largest 50,000 segments (by synapse count) and eliminated apparent false merges using Neu3 (ref. 70), a primarily 3D-oriented tool. While inspecting the segments, proofreaders had the opportunity to place 'to do' landmarks on the dataset in places where a merge was likely to be required, without interrupting their cleaving work to perform any merges. As we were able to quickly identify many of the cells that belong to repeated cell types, we grouped some of the cleaving tasks by cell type. In those cases, proofreaders were able to quickly become familiar with the expected morphology of the neurons under review and spot issues more efficiently and reliably. After the largest 50,000 segments in the optic lobe were inspected and cleaved, we addressed the largest 50,000 segments in the remainder of the volume, which (at the time) was half of the brain.

**Backbone tracing.** Having eliminated most false merges in the dataset, we could then efficiently assemble the largest fragments into accurate cell shapes. Segments were assigned in priority order according to synapse count. Obvious false splits (usually on large branches) were identified and traced across, ideally resulting in complete neuron backbones.

**Focused merging.** Our next goal was to increase the rates of synapse capture across the dataset by merging orphan fragments into their parent segment. We prioritized orphans with higher synapse counts to maximize impact on the connectome. As described in the next section on orphan linking, one method for performing such merges is to task proofreaders with careful inspection of each orphan of interest and locating the path leading to a sizable neuron. However, in many cases, there was a better option: if the orphan happened to be directly adjacent to one or more neurons, which seemed like a possible merge target, then we could present a simple yes-or-no decision to the proofreader through the Neu3 'focused merging' protocol. We did not present all of the adjacent neurons of each orphan to the proofreader. Instead, we selected merge candidates using the agglomeration scores originally generated by the automated segmentation algorithm, essentially using a manual review of merges that the automated procedure declined to accept. Focused merging can be much more efficient than orphan linking. The Neu3-focused merging protocol navigates to the precise location of the putative merge and presents the user with multiple views of the site, at which point the proofreader accepts or rejects the merge with the click of a button.

**Orphan linking.** Unfortunately, many orphan fragments could not be directly merged to their target neurons through focused merging. We prioritized the remaining orphans by synapse count and assigned them to proofreaders for manual tracing. The objective of the orphan-linking protocol was to find the target neuron for each orphan to which it should be merged (without tracing further branches of the target neuron). We note that orphan linking would be unacceptably wasteful if we did not intend to proofread the entire volume. Orphan linking is a purely bottom-up protocol because one does not know which target neuron will be found. If we had only been interested in a subset of neurons in the volume, then top-down tracing would have been the only option for improving synapse capture, although it is less efficient on a per-captured-synapse basis than bottom-up methods.

**Neuron approval by proofreaders.** Once we had captured all orphans with 50 synapses or more (or abandoned those for which no merge could be found), we entered the final top-down phases of proofreading. Proofreaders holistically assessed neurons, inspecting their

morphologies for inconsistencies. For cells belonging to columnar types, multiple cells were presented together for review to allow morphologies to be compared. Once a neuron or batch of neurons was deemed complete and error-free, the proofreader noted their approval in the database.

**Final neuron approval by expert neuroanatomists.** Our expert neuroanatomists (S.-y.T. and A.N.) gave final approval for all neurons in the dataset, relying on morphology, synapse counts and connectivity analyses to assess quality.

**Proofreader training.** For this project, our proofreading team consisted almost entirely of connectomics annotators with experience from earlier projects. All newly hired proofreaders receive at least 6 weeks of training, practising in a test environment on thousands of tasks. Feedback was provided during the training process using a mixture of automated and manual review. In addition to new hire training, all proofreaders occasionally received refresher training before transitioning to protocols they had not recently worked on.

**Additional quality controls.** Focused merging is particularly amenable to automated quality controls. In every batch of tasks, we selected a random subset of tasks for quality control and duplicated them across several proofreaders' task sets, randomly intermixed with other tasks. Approximately 10% of each proofreader's task load consisted of quality-control tasks. Each proofreader's rate of agreement with the majority decision was calculated, and those with notable disagreement rates were flagged for discussion and training. The median disagreement on true 'merge' decisions was 0.8%, and the median disagreement rate on true 'don't merge' decisions was 0.4%. For context, a disagreement rate of 0.4% corresponds to 1 disagreement for every 250 agreements.

**Proofreading time.** As this project was completed in the context of a larger reconstruction (an entire CNS), we made a few approximations to estimate the effort devoted specifically to the optic lobe. From an analysis of our task tables and segmentation edit logs, we estimate that a total of 2,584 proofreader or neuroanatomist days were expended on this dataset, approximately 10 proofreader-years. Extended Data Fig. 1b shows how that time was apportioned among the phases of proofreading described above.

**Quality metrics.** An important metric for evaluating the completeness of a connectome, or a region of a connectome, is the percentage of all synapses for which both the presynaptic and postsynaptic partners belong to 'traced' neurons. We provide this completion percentage for the optic lobe neuropils in Extended Data Table 1, and these summary metrics show that this new connectome is one of the highest quality reconstructions so far. The completion rate is high and relatively uniform across neuropils. Across the entire dataset, the connection completeness is 52.7% (54.2% when the partial lamina, discussed below, is excluded). This value is considerably higher than that achieved for the hemibrain, for which the corresponding metric is 37.5%[8].

However, it is possible for the overall completeness to be high while hiding more fine-grained problems. A useful neuron-centric metric is the 'downstream capture' of each neuron, defined as the fraction of a traced neuron's directly downstream postsynapses belonging to traced neurons. Upstream neurons with a particularly low downstream-capture fraction will have poorly reconstructed connections, even if the upstream neuron has been perfectly proofread. The distribution of the downstream-capture fraction, excluding the lamina, is shown in Extended Data Fig. 1c.

Another revealing quality metric is the number and size of unmerged orphan fragments in the dataset. Such orphans remain in the dataset owing to issues like data artefacts or especially difficult morphologies.

Our target was to merge all orphan fragments with greater than 50 synapses (upstream and downstream partners). The final distribution (Extended Data Fig. 1d) shows that there are 1,095 orphan fragments above this target cutoff. There are 9.6 million remaining unmerged fragments, most of them containing very few synapses.

**Limitations of the lamina data quality.** Completion metrics for the lamina (Extended Data Table 1) were considerably worse than the other optic lobe neuropils. The lamina is the most peripheral neuropil of the optic lobe where the axons of outer photoreceptors contact their targets. In this volume, the lamina was particularly difficult to reconstruct for two reasons: (1) it is incomplete, with a noteworthy stair-step profile owing to the trimmed edges of the 20 μm slabs, and (2) the severed axons of photoreceptors were generally heavily stained and were therefore much more challenging to segment than other neurons in the volume. Although it was not our original intention to analyse the lamina, the more complete segments of it are useful, especially for estimating the location of the equator of the eye[35], as shown in the eye map of Fig. 3a. For this reason, and despite the limitations of the sample, we invested considerable effort in the reconstruction of the lamina, but the completeness metrics reflect the lower quality and completeness of the data for this neuropil. The heavily stained inner photoreceptors (R7 and R8) that target the medulla also present considerable segmentation and proofreading challenges. Nevertheless, we proofread most of these cells, except those in one patch of the medulla, where these terminals were too fragmented to assemble into neurons. The distribution of these photoreceptors is detailed in Extended Data Figs. 1g,h and 5a, which shows how these data limitations do not extend to other cell types, even those with nearby connections. We estimated the counts of the missing cells in the section 'Estimating the count of photoreceptors and lamina intrinsic neurons'.

## LM–EM volume correspondence

To compare neurons imaged by LM and those reconstructed from EM data, we transformed the 3D coordinates of the central brain EM volume, including the optic lobes, into reference coordinates for a standard reference brain used for LM data. The spatial transformation was obtained by registering the synapse cloud of the male CNS EM volume onto the JRC2018M *Drosophila* male template[55], following an approach similar to that described for the hemibrain volume[8]. The synapse predictions were rendered at a resolution close to the JRC2018M template (512 nm isotropic) to produce an image volume. Next, we performed a two-step automatic image registration procedure using Elastix[72], with the JRC2018M template as a fixed image and the rendered synapse cloud as the moving image. The first step estimates an affine transformation, which was used to initialize the second step, during which a nonlinear (B-spline) transformation was estimated. We then manually fine-tuned this transformation using BigWarp[73] to place 91 landmarks across the datasets to define a thin-plate spline transformation to correct visible misalignments. This defined an initial transformation we used to warp neuron skeletons traced from the hemibrain dataset[8] into the coordinate system of this new (male CNS) EM volume. We identified putative neuron correspondences between hemibrain neurons and preliminary neuron traces from this volume and used these to identify remaining misalignments. We corrected these in BigWarp by placing 82 additional landmarks, bringing the final total of landmarks to 173. The composition of the transformations from these steps (affine, B-spline and thin-plate-spline) is the spatial transformation from the JRC2018M template to the EM space. We also estimated the inverse of the total transformation using code from GitHub (https://github.com/saalfeldlab/template-building). As there are transformations already defined between JRC2018M, JRC2018F and JRC2018U, establishing the correspondence for one of the three automatically does it for the other two. Figure 7c,d,g and Extended Data Figs. 14b and 15b–f show several image registration applications to compare neurons from the optic lobe dataset to similar cells in LM and EM (hemibrain) images using the JRC2018U template.

## Defining anatomical regions in the EM volume

Before segmentation, the neuropils were defined using synapse point clouds. The neuropil boundaries were initialized on the basis of the JRC2018M *Drosophila* male template[55] after registration with our EM volume, as described in the section 'LM–EM volume correspondence'. Then, the boundaries of the right optic lobe neuropils were refined with hand-drawn corrections based on the greyscale data. Although some boundaries were clear, others, such as those surrounding the inner chiasm of the optic lobe, were not sharp. The regions of interest (ROIs) for each neuropil were drawn to enclose the maximum number of synapses in the corresponding neuropil (without overlapping with adjacent neuropils). We named the regions following the standard nomenclature for the optic lobe[3] and central brain[61]. We then used an iterative process for the optic lobe to establish column and layer references, first reported in classic neuroanatomical studies[1–3]. We matched all of the classically defined layers except for the fifth layer of the lobula, which we divided into LO5A and LO5B following our previous definition from LM data[45]. This process is described in the section 'Defining column centre lines and ROIs' and the section 'Layer ROIs'. The resulting ROIs are available in neuPrint and are systematically named. Column ROIs were named by concatenating the neuropil, brain hemisphere and hexagonal coordinates (for example, LOP R col 15 21), and layer ROIs were named by their neuropil, brain hemisphere and layer number (for example, ME R layer 07).

## Connectome data access overview

The primary data source for our analysis was the optic-lobe:v1.1 dataset in neuPrint[74], a Neo4J[75] graph database. We accessed three levels of nodes in the database: (1) Segment; (2) SynapseSet, which are collections of synaptic sites representing T-bars and postsynaptic densities; and (3) Synapse, which refers to the individual synaptic site. These nodes have a hierarchical relationship, whereby the Segment nodes contain SynapseSet nodes and the SynapseSet nodes contain Synapse nodes. Both the Segment and SynapseSet nodes connect to other Segment and SynapseSet nodes, respectively. These connections are calculated on the basis of the relationship between individual Synapse nodes. All Synapse nodes derive from an Element type and inherit their properties, such as the 3D ($x,y,z$) coordinate in the EM volume. The majority of Segment nodes correspond to small, unproofread and unnamed fragments with few synapses and are not relevant to most analyses. To enable faster queries, the subset of Segment nodes that are relevant for typical analyses are also labelled as Neuron nodes. To qualify as a Neuron, a Segment must have at least 100 synaptic connections, a defined 'type', 'instance' or known soma location. In most data analyses presented here, we primarily work with named Neurons; that is, Neuron nodes with defined type and instance properties. We ignored non-Neuron Segments and excluded Neurons for which types are suffixed with '_unclear' and have a short list of reconstructed Neurons to ignore.

For full clarity, in the graph database syntax[74], the nodes are ':Neuron' ':Segment' and have a ':Contains' relationship to ':SynapseSet's, which have a ':Contains' relationship with ':Synapse's. ':Neuron's and ':SynapseSet's have ':ConnectsTo' relationships to other ':Neuron's and ':SynapseSet's, respectively. These relationships are calculated on the basis of the ':SynapsesTo' relationship between individual ':Synapse' nodes. Each ':Neuron' and ':Segment' have the property 'bodyId' that serves as a unique identifier. All nodes that inherit from ':Element', such as ':Segment,' ':Neuron,' and ':Synapse' contain spatial properties and can therefore take advantage of the spatial functions of Neo4J.

In neuPrint data loaders, ROIs are defined as connected volumes; that is, spatially contiguous collections of voxels. Consequently, each element with a spatial location can be assigned to any number of ROIs,

such as ROIs we defined for layers, columns and neuropils. This is a simple assignment for point objects such as postsynapses, presynapses and column pins (defined below). Because synaptic connections are typically based on a single presynapse and several postsynaptic densities, neuPrint uses the convention of assigning the connection to the ROIs of the postsynaptic site.

Although our study focused on the visual system, several cell types have processes in the central brain. Although proofreading and naming efforts are ongoing and not as complete as the optic lobe, the reconstructed arbours are included in the dataset as skeletons (and meshes) but without synaptic connections. In the database, the aggregated summaries of synaptic sites and their innervated brain regions representing the current state of the reconstruction effort are included as properties of neurons. These aggregations rely on the confidence threshold of 0.5 set for the optic-lobe:v1.1 dataset. As a result, the summaries are consistent with the late 2024 snapshot of the reconstruction (optic-lobe:v1.1), but the synapse properties of ':Neuron's differ from the number of ':Synapse's they contain (as central brain connections are incomplete, only half the connection is present).

In the Python environment, neuPrint-python[76] provides access to the neuPrint servers and translates function calls to Neo4J Cypher queries (https://connectome-neuprint.github.io/neuprint-python/docs/). We relied on the carefully selected, reasonable default values of the neuPrint database (the most relevant is the synapse confidence threshold of 0.5). To simplify analyses, we aggregated data as close to the source as possible either in pandas DataFrames or using the 'fetch_custom' function of neuPrint-python with optimized Neo4j Cypher queries. The code we share in the repository (see the Code availability section) uses a combination of these methods for data access. For estimating the spatial location of an element, such as a synapse, in relation to columns and layers, we added the data type ':ColumnPin' to neuPrint, which represents points positioned along the centre lines of each column (of the medulla, the lobula and the lobula plate) and with properties representing the identity of and depth in a column.

In addition to the neuPrint Neo4J database, we provide neuron morphologies and boundaries of ROIs as meshes through Google Storage buckets. Meshes were generated from ROIs with marching cubes, followed by Laplacian smoothing and decimation by quadrics (https://github.com/sp4cerat/Fast-Quadric-Mesh-Simplification). Meshes for brain regions, columns and layers are available in three different spatial resolutions between 256 nm and 1,024 nm (per side of voxels), whereas meshes of most individual neurons were generated from 16 nm voxels.

Data specific to our analyses and retrieved by other methods are stored inside the code repository in the '/params' directory, such as the following:

- The 'Primary_cell_type_table' groups the named neuron types into the four main categories: ONIN, ONCN, VPN, VCN and 'other' (see the section 'Cell-type groups').
- The identification of example cells (and their corresponding bodyId) for each neuron type, which we call 'star neurons' inside the 'all_stars' spreadsheet (see the section 'Gallery plots').
- In a second spreadsheet, we provide some heuristically defined 'Rendering_parameters' (see the section 'Gallery plots').
- Additional files contain parameters used to generate pins and layers: 'pin_creation_parameters' (see the section 'Defining column centre lines and ROIs' and the section 'Layer ROIs').

The data inside the neuPrint database, precomputed meshes stored at the Google Cloud and the data inside the '/params' directory are sufficient for replicating our data analyses and figures; for example, using the source code we provide in our repository (see the Code availability section).

We provide our code in notebooks (literate programming[77]) to explain analytical steps or command-line scripts for long-running processes. Both rely on functions for prototypical implementations or object-oriented components shared between several of our applications.

## Visualization of reconstructed EM neurons

The neurons in our dataset are all in the right optic lobe; that is, on the right side of the fly brain. However, we represent this optic lobe on the left side of our images (Fig. 1a), as this is the view most familiar to anatomists (that is, looking down at a sample in a microscope) and is directly comparable to LM data. To provide further intuition for these perspective changes, we show the coordinate system of the correspondence of the medulla to the retina (and lenses) of the right eye when viewed from inside the brain and looking outwards (Fig. 3a).

Throughout the article and associated materials, EM-reconstructed neurons were rendered using several methods introduced here (detailed in later sections). Neurons are generally shown as orthographic projections of a 3D view; most are programmatically rendered in Blender[78] and described in the section 'Pipeline for rendering neurons'.

Many visual system neurons are shown on an optic lobe slice view, an anatomical section containing neurons of interest and relevant optic lobe neuropils. Parts of the neurons may lie outside the sliced volume; therefore, we used a pruning step to show layer innervations (which was required owing to curvature of the optic lobe neuropils).

- The section 'Pipeline for rendering neurons' describes the method used to produce the images in Figs. 1c, 2a,b,e, 4c and 5a,e,f and Extended Data Figs. 5e, 6c,d, 8c and 15a, and the Cell Type Catalogue (Supplementary Fig. 1).
- For full-brain views of rendered neurons, we used methods described in the section 'Pipeline for rendering neurons' to produce the images in Fig. 6 and Extended Data Figs. 11–13.
- We used some manually produced, custom Blender visualizations for the images in Figs. 1b, 5c and 7i and Extended Data Fig. 8a.
- The web-based viewer neuroglancer[79] was used to produce the images in Figs. 1a and 3b,d and Extended Data Figs. 2d,e and 16b,c.
- The LM images and the comparisons with EM-reconstructed neurons registered to a standard reference brain were produced by methods described in the section 'Split-GAL4 lines that are expressed in visual-system neurons' and shown in Figs. 2b, 3c and 7 and Extended Data Figs. 3b, 14 and 15.

## Pipeline for rendering neurons

Despite the complex, extended 3D shapes of most neurons, 2D visual representations of neurons have served as highly detailed descriptions of cell types across generations of neuroscientists, from hand drawings of Golgi-stained neurons[1–3] to computer-generated ray traces of reconstructed EM volumes[80]. Following open-science principles, we developed a rendering pipeline based on the state-of-the-art open-source software Blender[78], which enabled us to produce static printable images following the visual style developed over more than a century of communicating microscopic structures and videos for accessible science communication. The Cell Type Explorer web resource contains interactive figures[81] to facilitate individual exploration of the morphology of a neuron.

To render detailed, high-resolution printable images and videos, we developed a software pipeline and integrated neuVid[82], a pipeline for making videos, into our workflow. In both cases, we began with the bodyId for individual neurons (for which we most commonly use 'star neurons' selected from a curated list of representative neurons for each cell type, listed in the 'all_stars' spreadsheet in the code repository '/params' directory). We then queried neuPrint for additional information about this neuron, such as the layer and column innervation, cell type names, synapses and neighbouring cells (of the same type). Based on five templates, we combined this information with material properties and virtual camera location ('Rendering parameters' spreadsheet in the code repository '/params' directory) into JSON description files. Next, we generated the images using our image pipeline or neuVid for

videos. Based on the image and video JSON descriptions, we downloaded the required meshes for neurons and brain regions from a Google Cloud storage bucket (see the Data availability section) and temporarily stored them inside the '/cache' directory of the local file system.

The rendering was done using Blender for both videos and images. For the videos, we relied on the neuVid package[82] to move the camera and to assemble individual frames of high-resolution images. Supplementary Videos 2 and 3, and neuron-specific videos that are linked from the Cell Type Explorer web resource, were produced using this pipeline.

For rendered, static images, we developed the pipeline to allow free positioning of the camera. Most neuron images are rendered from two main directions, similar to the frontal and side views of the optic lobe in Fig. 1b. For the full-brain view (for example, Fig. 6), we placed the virtual camera anterior to the animal with a posterior viewing direction of the orthographic projection. For the optic lobe slice view, we positioned the orthographic camera laterally and within a few degrees of the dorsoventral axis (similar to the side view of Fig. 1b). In our standard renderings with the full-brain view, we produced whole-brain images for neurons with midline-crossing projections and produced images of slightly more than half a brain for all other neurons. To render representative examples of all optic lobe neurons in an optic lobe slice view, we established three virtual camera positions: an equatorial view used for most neurons, and ventral and dorsal views better suited to visualize a subset of neurons and types. In Supplementary Fig. 1, the slice position of each panel is indicated with a letter (E, D or V, respectively).

The whole-brain view benefits from outlines of the different brain regions, specifically the two optic lobes and the central brain. To produce these outlines, the 3D meshes were moved along the optical axis away from the virtual camera. The mesh material was modified inside Blender to not reflect light and to give the illusion of a featureless backdrop. The neurons were then plotted and rendered in their original coordinates.

The optic lobe slice views (for example, Figs. 1c and 2a, and all the panels of the gallery in the Cell Type Catalogue, Supplementary Fig. 1) required substantial fine-tuning to establish representative visualizations of individual neurons that capture their layer-specific arborization patterns while visualizing the layers and capturing the axons, cell-body fibres, soma locations and, where relevant, central brain arborizations. We obtained a visual representation of the layers of the optic lobe neuropils by intersecting the medulla, the lobula, the lobula plate and accessory medulla meshes with a rectangular cuboid of around 2.5 nm thickness near the ventral, equatorial or dorsal part of the neuropil. We used a similar approach for the videos with 100-nm-thick cuboids. The intersections between layer meshes and cuboids were then represented as a wireframe around the edges of the newly created mesh. We used the wireframe to emphasize the boundary of the slice with a darker colour. The neurons themselves were sliced with a thicker rectangular cuboid. Depending on the cell and innervation pattern, we used between 750-nm-thick and 3-µm-thick cuboids to constrain the visible parts of a neuron. For most neurons, we used slices only inside the layered optic lobe neuropils (medulla, lobula and lobula plate), although we show all of the reconstructed arbours outside these three neuropils. This enabled us to capture the layer-specific arborization patterns (owing to the curvature of the optic lobe neuropils, projections through thick slices obscure most of these patterns). For about 100 neurons, we extended the slice outside the layered optic lobe neuropils. For seven cell types (Cm-DRA, Cm27, Li37, LPi14, LPi4a, LPT30 and Mi20), we manually pruned arbours to achieve the visualization goals described in the section 'Gallery plots'. The parameters for the visualizations of each type are stored in the 'all_stars' table in the '/params' directory.

To show innervation patterns in specific medulla layers, such as in Figs. 2e and 5e,f, our optic lobe slice view was modified so that the camera was turned approximately 90° for a face-on view of the medulla layers. In these images, only the relevant medulla layer was rendered.

For renderings that included synapses (Extended Data Figs. 8c and 15a), we only show synapses to/from the indicated neurons, and only the synapses inside the bounding rectangular cuboid were rendered. The synaptic sites are represented by a sphere and were assigned a visually pleasing diameter.

The generated raster graphics were stored on the local file system. In the final step of assembling individual panels or galleries of images, such as in Fig. 6, Extended Data Fig. 13 and Supplementary Fig. 1, we used values from the same JSON descriptions to add text and to combine several images into a single page using the pyMuPDF[83] library.

We defined workflows for long-running and multistep computations with Snakemake[84]. Although we prototyped and could produce most images on a personal computer running Blender, with more than 700 cell types and several views of each type to render, we found Snakemake helpful in distributing the workload across the nodes of an LSF compute cluster.

## Cell-type groups

To facilitate further analyses and data presentation, we assigned each cell type to one of five main groups: ONIN, ONCN, VPN, VCN and other. Assignments to groups were done at the cell-type level, not the cell level. In general, ONINs and ONCNs were defined as cell types that have nearly all their synapses (>98% of both input and output connections) in the optic lobe, with ONINs restricted to a single neuropil of the right optic lobe: lamina, LA(R); medulla, ME(R); lobula, LO(R); lobula plate, LOP(R); and accessory medulla, AME(R) (the abbreviated label of each is the name of the corresponding ROIs in neuPrint). ONCNs are the cell types that connect two or more of these neuropils. VPNs receive substantial inputs in the optic lobe and project to the central brain. VCNs also connect the optic lobe and central brain but with the opposite polarity. 'Other' was used primarily for cell types with central brain and optic lobe synapses, with a comparatively small proportion of their connections in the right optic lobe (and therefore did not fit well into the VPN or VCN groups).

Photoreceptors (R1–R6, R7 and R8) were included with ONIN types, as this matched the distribution of their synapses. However, they could alternatively be considered ONCN cells (given their projection patterns). Ascending and descending neurons were placed in the 'other' group. We note that some of these have substantial optic lobe synapses and, therefore, could alternatively be classified as VPN (for example, DNp11) or VCN (for example, DNc01) types.

Although most cell types were unambiguously placed into one of these groups on the basis of their synapse distributions and overall morphology, more marginal cases required further criteria, particularly for deciding between VPN, VCN and 'other' groups. For most cells, the assignment was based on the relative numbers of synaptic connections. To classify a cell as a VCN (VPN) instead of other, we required that >10% of the total output (input) connections were located in the optic lobe (both optic lobes combined for neurons that also have synapses in the left optic lobe). We applied this threshold to the averaged synapses across all cells of a type; in the very few cases (for example, DN1a and SMP217) in which one or more (but not all) of the cells of a type passed the 10%, threshold but the cell type average did not, we required that at least half of the cells met the VPN or VCN criteria. The AME was treated in the same way as other optic lobe neuropils for this purpose (placing, for example, the s-LNv clock neurons in the VPN group). A few cell types placed in the 'other' group (in particular, PVLP046) have a substantial number of synapses near the optic lobe but outside the primary ROIs ('NotPrimary' in neuPrint); treating these synapses as optic lobe synapses would change their classification.

Additional criteria used when classifying cells as VPNs or VCNs included the proportion of downstream synaptic connections relative to total synaptic connections in the optic lobe and central brain, the grouping of related cell types and details of arbour structure (for example, for cells that primarily receive input in one part of the optic lobe

and have major output connections in both another optic lobe neuropil and the central brain). When grouping the cell types, we aimed to keep the number of cell types in the 'other' category low; we therefore, chose a relatively low minimum proportion of optic lobe synapses (10%, see above) for VPNs and VCNs and (tentatively) classified a few ambiguous types as VPN or VCN based on the balance of the available evidence.

## Estimating the count of photoreceptors and lamina intrinsic neurons

To provide a nearly complete count of cells in the optic lobe (related to Fig. 1 plus the estimate listed in the figure legend), we accounted for two technical limitations of the dataset. First, because the lamina is incompletely contained in our volume, the number of reconstructed Lai and R1–R6 cells is an underestimate of the total count of these cells. Second, owing to poor segmentation of some photoreceptor cells, not all R7 and R8 photoreceptors in the medulla could be reliably identified, which resulted in another undercount (Extended Data Fig. 5a). For the analyses, we used the Corrections.xlsx data table in the '/params' directory. An approximate number of R1–R6 cells was obtained as the total number of non-edge columns times six (again rounded to the nearest ten). We counted labelled Lai cells (labelled using split-GAL4 driver SS00808) in three optic lobes and used the rounded average (210) as our estimate for Lai. For an estimate of R7 and R8 numbers, see the section 'Assigning R7 and R8 photoreceptors and medulla columns to different types of ommatidia'.

## Overview of cell typing

We used both morphology and synaptic connectivity to group cells into types (Figs. 1 and 2). Morphology-based typing was generally done by direct visual inspection of reconstructed EM bodies. The combination of features such as cell-body positions and the number, size and layer position of their arbours gave cells of many optic lobe cell types a distinct appearance, which, with experience, was often directly recognizable. As the initial typing was carried out in parallel to proofreading, our review of cell morphologies also identified apparent reconstruction errors and directed targeted proofreading efforts. Morphology-based cell typing was facilitated by the considerable amount of available previous information on optic lobe cell types[3,6,19,45,85,86] both from published descriptions and unpublished LM analyses (see the section 'Split-GAL4 lines that are expressed in visual-system neurons'). Cell-type annotations were performed iteratively. As the proportion of typed (or preliminarily typed) neurons increased, we increasingly relied on synaptic connectivity (see the section 'Connectivity-based cell clustering') to confirm or refine annotations, to identify outliers and to place cells into preliminary groups for further review and annotation based on morphology.

Over several cycles of this procedure, and because of the high level of completeness of our dataset, we assigned types to nearly all reconstructed EM bodies above a minimum size (about 100 combined input and output synapses). All EM bodies (except many in the lamina) that our expert proofreaders assessed to be individual neurons have been annotated with a type and instance. We used all available evidence to label some atypical cells (for example, near the margins) and gave about 750 remaining bodies names, all including '_unclear'. These unclear types provide some information (for example, R7_unclear) but indicate the currently unresolved typing status (about 600 of these unclear bodies are R7 or R8 photoreceptor neurons that we considered either too incomplete or are from columns we could not assign as pale or yellow (see below)). Most of the remaining unnamed bodies in the dataset above the 100-connection threshold were smaller fragments that were primarily severed parts of annotated neurons and bodies in the lamina for which annotation and proofreading were less complete than in the other parts of the optic lobe. To support proofreading efforts and connectivity-based clustering, we also assigned candidate types to some smaller fragments; most of these were eventually merged with

other EM bodies, but about 700 of such fragment annotations are still available in the 'instance' field of neuPrint. Additional aspects of cell typing are addressed in the sections 'Cell-type nomenclature', 'Cell-type groups' and 'Connectivity-based cell clustering'.

## Cell-type nomenclature

There can be no perfect nomenclature for this diverse set of neurons that respects historical usage while systematically describing the details of each cell type. In addition to following the historical precedent, we aimed to use short names that are typically easier to remember and, in most cases, give some indication that (and often how) a cell type is associated with the visual system (Figs. 1 and 2). Most of the cell-type names used in the visual system dataset, both the existing names and new names introduced here, consist of a short base name, for example, Tm, Dm or LoVP, that provides a broad classification of a neuron, followed by a number that distinguishes the individual types in this group.

Many individual cell-type names for ONINs and ONCNs are based on the systematic names provided in previous studies of the Golgi[3], which offered many of the base names we extended for naming new cell types. In addition to this resource, the following sources of names (and descriptions) for previously described neurons were included: several ONINs and ONCNs[6,7,9,38,87,88], Dm and some Pm cells[19], Tm5a–Tm5c[89], DRA neurons[23,90], LCs[45,85,86], LPCs and LLPCs[91], MeTus[48], LPTs[33], OA neurons[49], and many VCNs and VPNs and some Li cells[8]. For new names, we introduced additional cell-type base names, which were mainly part of a new systematic naming scheme for VPNs and VCNs.

Most new names for VPNs or VCNs follow a format that starts with Me, Lo or Lp, which indicates the main neuropil a cell type has input synapses (for VPNs) or output synapses (for VCNs) in, followed by VP or VC and a number. For VCNs with prominent outputs in two or more neuropils, we used names starting with OL (for optic lobe) instead. For bilateral cells, an additional indicator for the neuropils in the second (left) optic lobe was included after VP or VC (for example, MeVPMe1). In addition to these systematic names, a few new names were based on existing naming schemes, extending the base names used for groups of similar cell types (for example, LC10e and LLPC4).

We kept many hemibrain names for VPN and VCN neurons[8], but in cases for which previous names omit any indication of prominent visual-system connections, we introduced new synonyms in place of names based on central brain regions (for example, LoVC16 for PVLP132). For the few cell types with central brain and optic lobe synapses that were not classified as VPN or VCN but as 'other' (described in the section 'Cell-type groups'), we used their hemibrain names. In three cases, we applied a placeholder name, adding '_TBD', because we anticipate that future central brain data will provide a more appropriate designation.

We also introduced two new cell type base names: Cm and MeLo. The Cm cells are central medulla ONINs, named to complement the existing Dm (distal medulla) and Pm (proximal medulla) base names. This base name is applied to medulla intrinsic neurons with their main arbours in layers M6 and M7. The MeLo base name is a simple contraction of the first two letters of medulla and lobula and is applied to ONCN cell types that connect the medulla and lobula but do not project through the inner chiasm (in contrast to Tm cells). Cm and MeLo replace the Mti and ML labels that we had used for a small number of cell types from these groups in previous work[23].

Some established names for visual system cell types include letters to indicate further divisions beyond a shared base name and number; for example, LC10 types are split into LC10a, LC10b, and so on. In most cases, we continued to use these existing names (and introduced a few names of this type, for example, to subdivide an existing type). The use of letters and numbers, at least in the visual system, has not been standardized. That is, LC10a and LC10b are not necessarily more closely related than, for example, LC6 and LC16. We also did not make a formal distinction (as done in the hemibrain dataset[8] for central brain

neurons) between types identified through morphology versus connectivity. Many cell types most readily distinguished by connectivity still show other anatomical differences, at least at the population level. In two cases, we named subdivisions of cell types for which the existing names already include a letter (LC10c-1 and LC10c-2 and LC14a-1 and LC14a-2) but do not advocate for more widespread use of this practice.

The same name was applied to all cells of the type with morphological variability in features suggested by base names. For example, several Tlp13 cells have branches into the medulla that make them resemble Y neurons, but as this is not a feature of the entire type, we kept the Tlp base name.

There are many gaps in the numbers attached to base names. For example, our list of TmY neurons starts with TmY3 and TmY4, skipping TmY1 and TmY2. We did this to minimize confusion with previous usage. Some cell types illustrated and previously named in ref. 3 (for a long time, the main, if not only, source of cell-type information for neurons in the *Drosophila* optic lobe) seem to represent variants of other types. For example, the cell types labelled Tm6 and Tm14 from a previous study[3] both match our Tm6, whereas Mi8 is probably an atypical Mi1. Moreover, the cell types labelled Tm15 and Tm20 from that study[3] both resemble our Tm20. Second, to avoid incorrect matches with earlier studies (some cell types are difficult to match to the limited information available from single-view Golgi drawings), several recent efforts by us and others have started with nonoverlapping sets of numbers when naming additional types (a practice continued here for many cell types, for example, the Tm cells).

### Summarized inventory of visual neurons and connectivity
Figure 1 shows a summarized overview of the visual neurons in the optic-lobe:v1.1 dataset. Supplementary Table 1 lists the 732 unique cell types and 778 unique cell instances.

For Fig. 1d, we considered all the cell types that belong to the four main neuron groups defined in the section 'Cell-type groups'—ONINs, ONCNs, VPNs and VCNs—and report the number of input and output connections for each cell type. There are different ways to count connections in the dataset, so it is essential to clarify our conventions. A synaptic connection (or simply connection) is a paired presynapse on one body and a matched postsynapse on another. Autapses (self-connections) have been expunged from the dataset, and most of our analysis is restricted to named neurons (see the section 'Overview of cell typing'). Because connections are directed and between two segments, each side of the connection can be counted as an input connection or an output connection. The counts in Fig. 1 report these input and output connections for the 160 cell types listed in Fig. 1d (pooled across all neurons of each type) or for all cells in each group (Fig. 1e) or by neuropil and group (Fig. 1f). In the remaining analyses and materials (except analyses of synapse completeness), we report connections limited to only those between identified (named) cells. Therefore, the counts found on, for example, the Cell Type Explorer web resource, are expectedly lower (average connection completion percentage for the dataset is about 53%). Presynapses is used for counts of T-bars (presynaptic active zones). Postsynapses is used to refer to counts of postsynaptic densities opposing presynapses. Because synaptic connections are asymmetric, with one presynapse typically opposed to multiple postsynapses, the count of postsynapses in a neuron is identical to the number of input connections, but the count of presynapses is not the same as the number of output connections. This difference is related to the 'fan-out' of neurons, which is examined in Fig. 4i. There was also an asymmetry in completion percentage between presynapses and postsynapses (Extended Data Table 1), which explains why the output connection numbers are higher than input connections when querying the database for neuron connections (as in Fig. 1). Nearly all (about 97%) outputs (through presynapses) have been assigned to a named neuron, whereas about 55% of postsynapses are assigned to named neurons. This connection asymmetry is related to the physical structure

of *Drosophila* synapses, in which a single, large presynaptic area of one neuron typically contacts multiple postsynaptic sites of often multiple different neurons. These postsynaptic sites are typically located on finer, more difficult to reconstruct processes than presynapses.

In Fig. 1f, a neuron was assigned to one of the five optic lobe neuropils if >2% of the summed presynapse and postsynapses of a neuron were contained in the corresponding neuropil ROI. Similarly, a neuron type was assigned to one of the optic lobe neuropils if >2% of all the summed presynapse and postsynapses of all the neurons belonging to that cell type were in the corresponding neuropil ROI. All five lamina tangential (Lat) types were confirmed to have arbours in the lamina (LM data, not shown), but because the distal lamina is not in the EM volume, we captured very few of their synapses and therefore some Lat types were not counted in Fig. 1f. For Fig. 1g, we considered all the connections (>1) between the 732 cell types and report the mean number of connected input cells and output cells for each cell type.

### Connectivity-based cell clustering
To group cells into types by connectivity (Fig. 2 and Extended Data Fig. 3), we used aggregate connectivity to named cell types as the basis for clustering. For each reconstructed EM body to be clustered, we calculated the sum of its connections, further split into inputs and outputs, to all cells of each named cell type in the dataset (not including connections with cells with '_unclear' type annotations). Connections with cell types with synapses in both optic lobes were further split by instance (right and left side), because the connectivity in the right optic lobe of the left and right instances of cells of the same type can substantially differ. Small EM bodies annotated as 'fragments' of a cell type (for example, the instance MeLo9_fragment_R in neuPrint) were included with the corresponding cell types when calculating the summed connection weights. Excluding 'unclear' cells and adding 'fragments' was crucial during the active proofreading stage of the reconstruction, when many EM bodies still had obvious reconstruction errors or were clearly incomplete. Finally, as the dataset reported here does not include detailed connectivity in the central brain, we only used connections in the right optic lobe, which for this purpose was the sum of connections in the LA(R), ME(R), LO(R), LOP(R) and AME(R) ROIs.

The resulting connectivity table was used as input for hierarchical clustering. Clustering was performed in Python using the scipy.hierarchy[92] and fastcluster[93] libraries. We used Ward's linkage method with cosine distance as the metric based on the observed excellent agreement between our expert-curator-led morphology-based cell typing (not to be confused with the independent clustering based on quantified morphology described in the section 'Morphology clustering') and connectivity clustering for well-characterized cell types (Fig. 2 and Extended Data Fig. 3). The key aspect of using cosine distance seems to be the normalization that is part of calculating this metric; L2 normalization of each row of the connectivity table followed by clustering with Ward linkage and Euclidean distances produced similar results. The clustering output was divided into a preselected number of clusters (using scipy.hierarchy.fcluster with the maxclust criterion). For morphologically distinct cell types, connectivity clustering primarily served as a method for preselecting cells for subsequent morphology-based typing. For other cell types, connectivity was the primary determinant of cell groupings, and reviewing cell morphologies was mainly used to identify outliers, such as cells with apparent reconstruction errors. However, we note that for several cell types initially separated by connectivity clustering, we identified at least some morphological differences between the two cell populations (for example, differences in arbour spread or cell-body locations).

To illustrate the clustering of larger groups of cells, we used 15 columnar cell types (Fig. 2d) and ONIN neurons of the medulla with ≥10 cells per type as examples (Extended Data Fig. 3a). For the first group, we set the number of clusters to the number of types (in this case

independently defined by morphology) and observed a one-to-one correspondence between cell type annotations and cluster assignments. For the second example, we selected a number of clusters slightly exceeding the number of types (80 for the 68 medulla ONINs with ≥10 cells per type). In this case, although most clusters only contain cells of a single type, some cell types are not separated (but can be by reclustering), and some other types (combined as one type based on shared features, spatial coverage or genetic information from driver lines) are split into multiple clusters. This example illustrates why additional criteria (as mentioned below) are required to decide which clusters should be annotated as distinct types.

Connectivity differences between two cell populations do not automatically imply that these represent different types. For this reason, we considered several factors, including cell morphology, consistency and magnitude of the connectivity differences, and, when available, genetic information in the form of split-GAL4 driver line patterns to help decide when to split cells into separate types. In addition, we relied on the spatial distribution of repeating neurons in the optic lobe, as many cell types form mosaics covering a neuropil (Fig. 2g and Extended Data Fig. 4). For this analysis, we used connectivity clustering to split the cells of existing types or combinations of similar types into two (or more) groups and plotted the positions of individual cells in each group (calculated as the centre of mass of the positions of their synapses). Positions were transformed into a 2D representation by plotting the first two principal components (PCs) (obtained for a set of preselected synapses for each neuropil to standardize the view across cell types; denoted as, for example, $PC1_{ME}$ and $PC2_{LO}$, for the first and second PCs, for viewing the synapses of candidate cell types in the medulla and lobula, respectively). In general, we considered the combination of a split into two regular, overlapping patterns (mosaics), especially when combined with consistent connectivity differences, as a basis for dividing a cell population into two types. Extended Data Fig. 4 shows examples of the outcomes from applying this method, ranging from clear cases for splitting cells with overlapping mosaics to regional patterns or patterns with no clear spatial structure along with only modest connectivity differences, which were not further subdivided.

We note that some of our cell types consisted of single cells that we named individually (often applying established names) because the available evidence suggests that they are uniquely identifiable. However, some cells with unique names belong to small groups of types that together cover part or all of the columns and which could alternatively be combined using similar coverage criteria (examples are the HS cells, typed separately as HSE, HSN and HSS, or the CH cells, typed separately as DCH and VCH).

## Assigning R7 and R8 photoreceptors and medulla columns to different types of ommatidia

R7 and R8 photoreceptors each exist in three major subtypes with different light responses primarily due to distinct rhodopsin expression[90]. In most ommatidia, subtypes are present in one of three combinations, R7d/R8d (DRA photoreceptors), R7y/R8y (yellow R7/R8) or R7p/R8p (pale R7/R8). Previous studies have shown that these subtypes differ in their synaptic connectivity and in specific morphological features of associated cell types or, in the DRA region, the photoreceptors themselves[23,94–96]. As rhodopsin expression cannot be directly detected in EM images (without additional genetic labels that were not used in this or previous EM studies), assigning candidate pale and yellow photoreceptor subtypes to EM reconstructions relied on morphological features (the presence of arbours of specific cell types in a column) that were identified by previous LM studies. As described further below, we used both such anatomical markers and synaptic connectivity to assign candidate subtypes to the reconstructed R7 and R8 photoreceptors in this dataset. We used synaptic connectivity to distinguish different groups of photoreceptors and anatomical markers established in previous studies (primarily Tm5a and aMe12 branches) to assign

subtypes to these groups. The morphological markers also enabled us to distinguish photoreceptor groups without relying on synaptic connections and to assign some columns without reconstructed R7 or R8 photoreceptors (Extended Data Fig. 1g,h) as yellow or pale. However, as further discussed below, it is unclear whether the available information is sufficient to predict subtypes for all R7 and R8 photoreceptors, and we accordingly typed a substantial subset of cells (and columns) as 'R7_unclear' or 'R8_unclear'.

We identified DRA photoreceptors, R7d and R8d, on the basis of the unusual layer pattern of R8d cells, which, in contrast to other R8 types, project to approximately the same layer (around M6) as R7d, and the distinct synaptic connectivity of R7d and R8d.

We placed non-DRA R7 cells into two groups on the basis of their relative number of synaptic connections to two pairs of cell types: Tm5a plus Dm8a and Tm5b plus Dm8b (Extended Data Fig. 5c). We chose these cell types on the basis of previous reports of selective connections of R7 subtypes to these neurons. Connectivity clustering of R7 cells using all their synaptic connections in the medulla produced a near-identical split. As Dm8a and Dm8b and Tm5a and Tm5b are crucial for sorting R7 cells by connectivity, we confirmed that the typing of these types themselves did not depend on the subtype assignments of R7 and R8 cells (Extended Data Fig. 5c,d).

Following previous studies[23,94–96], we examined distinct features of the morphology of Tm5a, Dm8a, Dm8b and aMe12 branches as candidate markers for yellow and pale columns (summarized in Supplementary Table 3). For Tm5a, we looked for prominent arbours extending distally from layer M6. These branches typically represent the distal part of a single main arbour of these cells and are primarily located in a single column, although a minority of Tm5a cells had more than one distal branch. For Dm8a and Dm8b, we identified prominent distal projections (sometimes referred to as 'home column' branches) from layer M6 to about M4. Consistent with published work[95,96], the majority of Dm8 cells had one such branch; some cells had more than one or no clear home column arbour. The identification of these home columns was primarily based on visual inspection of Dm8 cells (with candidate cells chosen by connectivity). We note that for small vertical branches, it can be ambiguous whether these are home column arbours. We did not attempt to introduce a more quantitative definition of a Dm8 home column because EM-resolution anatomical ground-truth data (which would be necessary to test the utility of such a definition for identifying pale and yellow columns) are currently not available. aMe12 neurons are large cells that, in a subset of medulla columns, have thin processes projecting along photoreceptors in a distal direction from layer M6, and we searched for columns with such vertical branches. To place these different anatomical features in specific columns, we visually matched them with adjacent R7 or L1 cells that served as column markers.

We found that nearly all columns with a candidate R7y cell suggested by synaptic connectivity included a Tm5a arbour but lacked vertical aMe12 branches, which was in near perfect agreement with expectations for R7y neurons (reported to be associated with Tm5a but not aMe12 vertical branches). Accordingly, we typed this subset of R7 cells as R7y cells. We found that these R7 cells were typically also associated with vertical branches of one Dm8 type (Dm8a), identifying a potential additional marker for these columns. Although most R7y cells were annotated on the basis of both morphology and connectivity, only one type of evidence was used for a small subset of these assignments.

We also found that the number (244) of strong R7y candidates identified as described above was considerably smaller than the expected total number of R7y cells in an eye of this size. With an expected ratio of yellow to pale ommatidia of about 1.5 (a number that may be higher in male flies[97]), we estimated that about 460 of the around 770 non-edge, non-DRA columns in the dataset house R7y cells. Even taking into account the incomplete reconstruction of R7 and R8 cells in the dataset, these numbers suggest that the second group of R7 cells

(which show preferential connectivity to Dm8b and Tm5b and, in most cases, are located in columns without a Tm5a branch or a Dm8a home column process) includes a substantial number of R7y cells. In the absence of additional markers that are known to reliably subdivide this group, we decided to only type cells in this group that were found in columns with an aMe12 branch as R7p and label the remaining cells as R7_unclear. We note that most likely, not all aMe12 branches were reconstructed or identified, thereby resulting in an undercount of not just yellow but also pale R7 cells. Our results suggest the possibility that yellow R7 (and R8) cells (as defined by rhodopsin expression), may terminate in different types of medulla columns: columns with Tm5a and Dm8a arbours (here identified as yellow columns) and other columns more similar to those with pale photoreceptor input. Further testing this hypothesis will probably require new experimental data. Although LM analyses of Dm8 cells, which are potential additional markers for pale and yellow columns, revealed two Dm8 subtypes: one (yDm8) associated with yellow and one (pDm8) with pale columns. However, the exact correspondence between yDm8 and pDm8 and the connectivity defined Dm8a and Dm8b is unclear. For example, we found similar numbers of Dm8a and Dm8b cells, whereas yDm8 have been reported to outnumber pDm8.

Subtypes of R8 cells were selected on the basis of the R7 cell in the same column. We also provide annotations of columns as pale, yellow and DRA (Supplementary Table 3). In the absence of sufficiently reconstructed or conclusively typed photoreceptor cells, column labels were based on the Tm5a and aMe12 morphological markers.

## Assigning neurons to medulla hexagonal coordinates

We assigned neurons to 892 hexagonal coordinates in the medulla in a multistep process that iteratively refined the assignments (Fig. 3). First, we manually assigned medulla coordinates to individual cells of 11 types using presynapses and postsynapses of the following connections: L1–Mi1, L2–Tm1, L3–Mi9, L4–Tm2, L5–Mi4 and L3–Tm20. Using the spatial distribution of the medulla presynapses of these 11 cell types, we calculated a straight-line central axis (a proto-column) for every medulla coordinate using the first principal component.

Second, we used these linear axes to allocate hexagonal coordinates for 13 cell types using the following paired connections in the medulla: L1 (L1–Mi1, L1–L5 and L1–C3); Mi1 (L1–Mi1, L5–Mi1 and Mi1–C2); C3 (L1–C3, L5–C3 and Mi1–C3); T1 (C3–T1); L2 (C3–L2 and L2–T1); Tm2 (L2–Tm2); Tm1 (L2–Tm1 and C3–Tm1); L5 (L1–L5, L2–L5 and L5–Mi1); Mi4 (L5–Mi4 and Mi1–Mi4); Mi9 (Mi4–Mi9, Tm2–Mi9 and C3–Mi9); Tm20 (Mi4–Tm20, Tm1–Tm20 and L2–Tm20); C2 (L1–C2, Mi1–C2 and L5–C2); and L3 (L3–Mi9, L3–Tm20 and L3–Mi1). For every paired connection, synapses were assigned to the nearest coordinate axis, and individual cells were labelled with the mode of the synapse assignments. For individual cells for which paired connections labelled different columns, we manually inspected the location of the in relation to the cells of the same type in neighbouring coordinates to assign the coordinate location. We also used manual inspection in relation to its neighbours to verify coordinates where cell types were missing or duplicated and to verify the assignment of every cell.

Third, we used the presynapses from these 13 cell types to calculate prototypes for the curved central axis for every column, which we refer to as columnar pins or column centre lines (described below). These prototype column pins were then used to automatically assign columns to the following 15 cell types: L1, L2, L3, L5, C2, C3, Mi1, Mi4, Mi9, T1, Tm1, Tm2, Tm4, Tm9 and Tm20. As before, we visually inspected the location of every cell in relation to the cells assigned to neighbouring coordinates to verify coordinates where cell types were missing or duplicated and to verify the assigned coordinate of every cell. The final coordinate assignments are summarized in Fig. 3a and Supplementary Table 3. We do not use L4 cells, which have multicolumnar processes, or Tm3 cells, which are not reliably assignable to the common coordinates of the other cell types. By using a large set of neuron types, the identification of columnar coordinates was robust to variability in cell types and reconstruction errors.

## Assigning T4 neurons to Mi1 cells to extend the coordinate system to the lobula plate

To create columns in the lobula plate and a mapping between medulla and lobula plate columns (Fig. 3 and Extended Data Fig. 6), we used the neurons of each T4 type: T4a, T4b, T4c and T4d. We used T4 cells for three reasons: (1) the axons of each type innervate one of the four layers of the lobula plate; (2) their dendrites receive strong Mi1 input; and (3) there are roughly as many T4 neurons of each type as there are medulla columns (T4a, 849; T4b, 846; T4c, 883; T4d, 860; and Mi1, 887). The major difficulties in assigning T4 neurons to medulla columns are that T4 dendrites are not strictly columnar but innervate about six medulla columns and that there are many cases in which a unique assignment of individual Mi1 to individual T4 of each type is not possible. But once all dendrites of T4 neurons were assigned to medulla coordinates, we could use the same coordinate assignment for their axons in the lobula plate, thereby linking medulla columns to lobula plate columns.

To achieve a nearly unique assignment between T4 neurons of each type and Mi1 neurons, we set up a global optimization problem (called 'maximum weight matching' in graph theory). First, for each T4 type, which we denote as T4$x$, we created a connectivity matrix $C_{ij}^x$ containing the number of connections between Mi1 neuron $i$ and T4$x$ neuron $j$. Then, to account for the bias in the number of synapses in different parts of the medulla, we normalized each connection by the total number of connections of each a Mi1 neuron to all T4$x$ neurons; that is, we defined a normalized connectivity matrix $\widetilde{C}_{ij}^x = C_{ij}^x / \sum_k C_{ik}^x$. Finally, we obtained putative Mi1–T4$x$ assignments using $\widetilde{C}_{ij}^x$ as the cost matrix in the Python linear_sum_assignment function from the scipy.optimize library[92] (setting the parameter 'maximize' to 'True').

Putative Mi1 to T4$x$ assignments were rejected if the distance between the neurons was too large (see below). We measured the distance between an Mi1 neuron $i$ and T4$x$ neuron $j$ as $\|r_i^{\text{pre,T4}x} - r_j^{\text{post}}\|$ where $\| . \|$ denotes the Euclidean norm, $r_i^{\text{pre,T4}x}$ is the mean position of presynapses from a Mi1 neuron $i$ to any T4$x$ neuron, and $r_j^{\text{post}}$ is the mean position of postsynapses of T4$x$ neuron $j$. The distribution of distances between putative Mi1–T4$x$ pairs showed a gap around 8 μm (approximately the distance between columns in medulla layer M9), which we chose as the distance threshold above which we rejected the putative Mi1–T4$x$ assignment. The results of this assignment procedure are detailed in Supplementary Table 4 and summarized in Extended Data Fig. 6b.

To create columns in the lobula plate, we defined valid assignments as groups of at least four T4 neurons of different types assigned to the same Mi1 but for which at least three T4 neurons were sufficiently close to the assigned Mi1. The mean position of presynapses on the axons of these T4 neurons provided us with at least three points to create the lobula plate columns. This procedure created 712 Mi1–T4 valid groups with all 4 types and 75 groups with three types.

## Defining column centre lines and ROIs

Column ROIs subdivide a neuropil ROI and are based on the centre lines of columns (pins) (Fig. 3). The basic idea of a column pin is that it is a smooth, slightly bent line that goes through the centre of synapses of specific neurons assigned to the same hexagonal coordinate (as described in previous sections). More formally, a pin is a list of 3D points (pin points, called ColumnPin in neuPrint) that start at the top of a neuropil ROI and end at the bottom.

Starting from a neuron to hexagonal coordinate assignment in the medulla (15 cell types, as described in the section 'Assigning neurons to medulla hexagonal coordinates'), we constructed pin $\alpha$ corresponding to hexagonal coordinates $\alpha$ according the following eight steps:
1) For all neurons assigned to the hexagonal coordinate $\alpha$, we obtained the 3D positions $r_i$ of their presynapses and postsynapses in the medulla, where $i$ runs from 1 to the number of synapses.

2) We identified the axes of biggest variation of $r_i$ using principal component analysis (PCA). We fixed the sign of the axis of biggest variation (PC1) such that PC1 points from the top to the bottom of the medulla ROI.

3) We computed the Euclidean distances $l_i$ of $r_i$ from their mean $\sum_i r_i$ in the PC2–PC3 plane, and dropped the $r_i$ for which $l_i < f_{lat}\sigma_{lat}$, where $\sigma_{lat} = \sqrt{\sum_i l_i^2}$ is the root mean squared distance of all $r_i$. The factor $f_{lat} = 2$ was chosen manually such that the $r_i$ with $l_i \geq f_{lat}\sigma f_{lat}$ looked like they were part of a different column.

4) For the remaining $r_i$, we defined their normalized projection onto PC1 as $t_i$ such that $t_i = -1$ corresponded to the top and $t_i = 1$ to the bottom of the medulla ROI. To clarify, this PC1 was computed in step (2), using all $r_i$.

5) For those $r_i$ with $t_i < 0.1$, we computed a new PC1 and found its intersection with the top of the medulla ROI. We denoted this 3D point as $r_{top}$. If there were fewer than $N_{PC}$ points with $t_i < 0.1$, then we took the points with the $N_{PC}$ smallest $t_i$ values instead to compute the new PC1. We chose the integer $N_{PC} = 800$ manually to ensure a robust estimation of $r_{top}$. In an analogous computation, using the points with $t_i > 0.2$ or the points with the $N_{PC}$ largest $t_i$ values and the intersection of their newly computed PC1 with the bottom of the medulla ROI, defined $r_{bottom}$.

6) We defined a list $L'_\alpha$, for which the first element was $r_{top}$, the last element was $r_{bottom}$ and intermediate elements were determined by rank ordering the $r_i$ by $t_i$. We defined another list $L_\alpha$ for which the first element was $r_{top}$, the last element was $r_{bottom}$ and the intermediate elements were the averages of the elements in $L'_\alpha$ with their $N_{avg}$ neighbours. The integer $N_{avg} = 260$ was determined manually such that the elements in $L_\alpha$ formed approximately a smooth line. This line was often helical and typically did not go through the centre of the $r_i$.

7) To create a line that went through the centre of the $r_i$, we found the shortest sublist $O_\alpha$ of $L_\alpha$, for which the first element was $r_{top}$, the last element was $r_{bottom}$ and the 3D Euclidean distance between neighbouring points was at most $d_{max}$. We defined $d_{max}$ as the 3D Euclidean distance between $r_{top}$ and $r_{bottom}$ divided by $N_{tang}$. We selected the integer $N_{tang} = 7$ such that $O_\alpha$ corresponded to a line that went approximately through the centre of the $r_i$. Note that the minimal length of $O_\alpha$ is $N_{tang} + 1$.

8) For each of the three spatial coordinates of points in $O_\alpha$, we used a spline interpolation (a piecewise cubic hermite interpolating polynomial (PCHIP)) to uniformly sample $O_\alpha$ with $N_{samp}$ points. This list of length $N_{samp}$ defined pin $\alpha$. We manually determined $N_{samp} = 121$ by checking whether the synapse distributions along a pin looked approximately smooth and did not change much with small changes in $N_{samp}$.

The parameters for this calculation are stored in the 'pin_creation_parameters.xlsx' file inside the '/params' directory.

To create pins in the lobula and the lobula plate, we first needed a neuron-to-hexagonal-coordinate assignment. In the lobula, this assignment was simply inherited from the medulla by choosing neurons that innervate both neuropils (namely, Tm1, Tm2, Tm4, Tm9 and Tm20). For the lobula plate, we used T4 neurons because they innervate both the medulla and the lobula plate. We assigned T4 neurons to hexagonal coordinates by assigning T4 neurons of each subtype to Mi1 neurons (from the section 'Assigning T4 neurons to Mi1 cells to extend the coordinate system to the lobula plate') and then using the hexagonal coordinate assignment for Mi1.

The steps to construct pins in the lobula and lobula plate were similar to the eight steps to construct pins in the medulla, but there were also important differences:

A. Instead of two separate PC analyses for the top and bottom of each pin (step (5) above), we performed one PCA using all the $r_i$ from step (2). With the exception of the top of the lobula, we defined $r_{top}$ and $r_{bottom}$ as the intersection of this newly computed PC1 with the top and bottom of the neuropil ROI, respectively. For the top of the lobula, $r_{top}$ was computed differently because the top of the lobula ROI contains ridges from the chiasm, which we did not want to replicate in our pins. However, they are necessarily replicated in our column ROIs as they subdivide a neuropil ROI. Instead, we defined $r_{top}$ as the extrapolation of the newly computed PC1 to the median position of the points with the 37 smallest $t_i$ values (from step (4)). Defined this way, the $r_{top}$ of different lobula columns lie on a slightly curved surface rather than on ridges.

B. In the lobula plate, PCA (in steps (2) and (A)) was not performed on the collection of all synapses of neurons but instead on the collection of spatially averaged synapses of neurons. To be clear, in the lobula plate, the number of points on which PCA was performed equalled the number of neurons (three or four). The reason is that the lobula plate ROI is not as thick at the medulla and lobula, such that the spatial variation of all synapses along the column (from top to bottom) can be comparable to that in the lateral directions. Using synapses would result in a PC1 tilted away from the columnar direction (between the top and bottom of the neuropil).

C. We chose different empirical parameters: in the lobula, we used $f_{lat} = 1.5$, $N_{avg} = 37$, $N_{tang} = 2$ and $N_{samp} = 76$, and in the lobula plate, $f_{lat} = 1$, $N_{avg} = 56$, $N_{tang} = 2$ and $N_{samp} = 51$.

D. In the lobula and the lobula plate, we imposed three criteria to make a pin (which would have been automatically satisfied in the medulla):
   (i) The number of $r_i$ after step (3) is at least $N_{avg}$.
   (ii) The 3D Euclidean distance between $r_{top}$ and the spatial average of the $r_i$ with the 5% largest $t_i$ values was at most $d_{max}$ (as defined in step (7)).
   (iii) The 3D Euclidean distance between $r_{bottom}$ and the spatial average of the $r_i$ with the 5% smallest $t_i$ values was at most $d_{max}$.

Criterion (i) was required to perform step (7). Criteria (ii–iii) were imposed because otherwise, the synapse positions $r_i$ (after step (3)) could be far from both the top and bottom neuropil ROI, for example, for the lobula if the cell type Tm20 was missing. In this case, the pin would often be tilted. We propose that a more complete neuron-to-hexagonal coordinate assignment in the lobula and the lobula plate could obviate the need for these criteria.

Based on the parameter $N_{samp}$, we produced medulla pins with 121 points along their centre line, lobula pins with 76 points and lobula plate pins with 51 points. This discretization was used to measure the position of a synapse along each pin, which we refer to as its depth. Because of the criteria in step (D), only 870 pins were created in the lobula and 783 pins in the lobula plate. These pins were less regular than those in the medulla. Therefore, for the lobula and lobula plate, we iteratively refined the pins by regularizing existing pins using neighbours and filling in missing pins. The iterative algorithm replaced pins $\alpha_{old}$ with pins $\alpha_{new}$ as follows:

- Regularization: for every created pin $\alpha_{old}$, we identified the created pins $\beta_{old}$ with hexagonal coordinates $\beta_{old}$ that differed from $\alpha_{old}$ by at most 2. For every $\beta_{old}$ and for every 3D point in the list of the pin $\beta_{old}$, we subtracted the first 3D point in the list of the pin $\beta_{old}$. The median of these lists across $\beta_{old}$ plus the first 3D point in the list of the pin $\alpha_{old}$ defined the new pin $\alpha_{new}$.

- Filling-in: for every missing pin $\alpha_{old}$, we found (i) the 4 created pins $\beta_{old}$ with hexagonal coordinates $\beta_{old}$ that differ from $\alpha_{old}$ by ±1 along either of the two hexagonal coordinates (but not both), or, if (i) did not exist, (ii) the 2 created pins $\beta_{old}$ where $\beta_{old}$ differs from $\alpha_{old}$ by ±1 along one hexagonal coordinate. If neither (i) nor (ii) exists, then there was no filling-in. Otherwise, the new pin $\alpha_{new}$ was defined as the median of the pins $\beta_{old}$ across $\beta_{old}$.

The iteration was stopped if no more pins were filled in. This took 2 iterations in the lobula, resulting in 875 pins, and 3 iterations in the lobula plate, giving 817 pins.

Given all the pins of a neuropil ROI, any 3D point in the neuropil ROI can be assigned a pin by minimal Euclidean distance; that is, it is assigned to pin $\alpha$ if the minimal Euclidean distance to pin $\alpha$ is smallest among all pins. The minimal Euclidean distance of a point to a pin was defined as the minimum Euclidean distance of the point to any pin point of that pin. We used this to show that the curved centre lines are a more accurate model of the shape of the neurons in the medulla, and therefore of the columns, compared with a straight line (quantified in Extended Data Fig. 6a, in which presynapses and postsynapses are assigned to columns through this method). The pinpoints are stored as ColumnPins in neuPrint, which can be used for efficient assignments of synapses to columns and corresponding depth. For details, see the section 'Connectome data access overview'.

The column ROIs shown in Fig. 3d were created by uniformly sampling a box surrounding the corresponding neuropil ROI at a resolution of 512 nm. Then, all sampled 3D points in the neuropil ROI were assigned a pin as described. All 3D points assigned to the same pin constitute a column ROI. The column ROIs were upsampled if an assignment at a finer resolution needed to be made. Assignments at different resolutions produced slightly different results. These column ROIs can facilitate quantification, as in Fig. 3e and Extended Data Fig. 7, as well as visualization (examples in Extended Data Fig. 6c,d).

### Layer ROIs

As a pin is an ordered list of 3D points going from the top to the bottom of a neuropil ROI in a fixed number of steps, we can view the order of the list as an indicator of layer depth (Fig. 3). Specifically, for any 3D point in a neuropil ROI, we found the order of the closest 3D pin point (in terms of Euclidean distance). We defined 'depth' as the order normalized to a number between 0 and 1 such that 0 corresponded to the first points in pins and 1 to the last points. By assigning a depth to a synapse, we computed 1D synapse distributions. For the distributions in Fig. 3c and Extended Data Fig. 7a, the synapse distributions were subsampled by a factor of 2. For the 1D synapse distribution of a cell type, we took all the synapses of all its neurons in each of the three neuropils for which pins were defined.

Layer ROIs subdivide a neuropil ROI on the basis of depth thresholds that define layer boundaries. Each depth threshold was chosen as the cut-off of a peak in the presynapse or postsynapse distribution of a cell type (except for the lobula plate, see later in this paragraph). The choice of cell type, presynapses or postsynapses, which peak and if we chose the upper or lower cut-off of a peak was guided by LM images selected to match previous notions of these layers. In Fig. 3c, we show the correspondence between EM and LM images and the specified criteria for defining layers. In the lobula plate, each depth threshold was defined as the mean of two T4 type depth thresholds. We used the mean of the thresholds of two types as there were gaps in depth between the different T4 types (therefore, defining the top of, for example, layer 3 separately from the bottom of layer 2 was not useful), as can be clearly seen in the LM image (Fig. 3c, right).

The peak cut-off values on 1D synapse distributions were determined using the following steps, which depend on one parameter frac_peaks:
1. We identified the threshold syn_thre on the 1D synapse distribution such that the fraction of the distribution above syn_thre was frac_peaks.
2. The lowest depth value at which the 1D synapse distribution surpasses syn_thre was defined as the lower cut-off value of the first peak. The next depth value, which was at least three depth points away from the lower cut-off and at which the 1D synapse distribution fell below syn_thre, was defined as the upper cut-off value of the first peak. Lower and upper cut-off values for other peaks, at larger depths, were similarly defined. Each cut-off value needed to be at least three depth points away from other cut-offs.

We chose frac_peaks for each neuropil roughly based on the peakedness of the 1D synapse distributions as follows: frac_peaks = 0.85 in the medulla, frac_peaks = 0.8 in the lobula and frac_peaks = 0.75 in the lobula plate. These parameters, the number of desired layers per neuropil ROI and a few other values are stored in the 'layer_creation_parameters.xlsx' table in the '/params' directory.

We could apply the same depth threshold to all pins to make layer ROIs. However, if done directly, we found that layers exhibited blockiness owing to the discretization of pins. We therefore applied a smoothing procedure. We defined smooth meshes for each layer as alpha shapes, which depended on two parameters: $\alpha$ and frac_ext. These were constructed as follows:
1. We identified the pin points at the lower and upper layer boundaries.
2. We identified the upper and lower edge pin points from the upper and lower pin points, respectively.
3. We identified the upper and lower edge mesh points by taking the neuropil ROI mesh points and the points closest to the upper and lower edge pin points, respectively.
4. We displaced the upper and lower edge mesh points further away from the corresponding upper and lower edge pin points by adding frac_ext multiplied by their difference.
5. We used the upper and lower pin points and the displaced upper and lower edge mesh points to define an alpha shape with parameter $\alpha$.

We used $\alpha$ = 0.0006 and frac_ext = 1 for the medulla, and $\alpha$ = 0.0004 and frac_ext = 0.5 for the lobula and lobula plate. These parameters are stored in the 'layer_mesh_creation_parameters.xlsx' table inside the '/params' directory. Note that a smaller $\alpha$ corresponds to more smoothing and smaller frac_ext to smaller layers in the lateral dimensions. The resulting smooth layer meshes could be slightly overlapping and/or missing small volumes of the neuropil ROI. These smooth layer meshes were only used to construct layer ROIs.

Similar to the column ROIs, the layer ROIs shown in Fig. 3d were created by uniformly sampling a box surrounding the corresponding neuropil ROI at a resolution of 512 nm. Then, all sampled 3D points in the neuropil ROI were assigned a layer with a two-step procedure. First, each 3D point in a neuropil ROI was assigned to a putative layer by depth, which resulted in 'blocky' layers. Second, the assignment was finalized by looping through the smooth layer meshes (from top to bottom) and assigning a layer if the point was contained in that smooth layer mesh. The putative assignment was necessary to account for the small volumes in the neuropil ROI that were not covered by any of the smooth meshes. Volumes covered by two smooth layer meshes were handled by assigning to the latter layer. All 3D points assigned to the same layer constituted a layer ROI. The layer ROIs are stored in neuPrint, which is how we use them to acquire synapses in each layer in Figs. 4g, 5b and 7h and the Cell Type Explorer web resource.

### Training and evaluation data for neurotransmitter predictions

Neurotransmitter assignments used as the ground truth for training and evaluating neurotransmitter predictions (see below) came from the literature or new experiments and are summarized in Supplementary Table 5. We primarily used data from the literature[33,38,49,87,98–103] for training data and performed new experiments to generate an expanded set of evaluation data (Fig. 4). Both published and new experiments use the expression of molecular markers, detected at the mRNA or, in a few cases, protein level, as indicators of the neurotransmitter. New data were generated using FISH. We used two adaptations of this method, FISH[37,104] and EASI-FISH[36]. Members of the Janelia Project Technical Resources and FlyLight Project teams performed the EASI-FISH and FISH experiments, respectively, using published protocols. We used Fiji to assess these results of the experiments qualitatively and to select sections for display. We routinely adjusted the brightness and contrast of the image stacks for evaluation and display. Cell types of interest were identified using split-GAL4 lines to specifically label

these cells. In some cases, multiple cell types were examined using the same driver line; this was possible when soma locations of these cell types were clearly different or if the cells of a group of neurons with overlapping cell-body locations showed signals with the same FISH probes. In most FISH and EASI-FISH experiments, we only probed for markers for cholinergic, glutamatergic and GABAergic transmission (ChAT/VAChT, VGlut and GAD1 probes, respectively). We only followed this first round with probes for octopamine, serotonin and dopamine markers in a few cases when we suspected an aminergic phenotype or if the results with the first probe set showed no clear signals in cells of interest. We acknowledge that this approach may miss some cases of co-transmission, but made many experiments far more efficient. It is also partly justified by published distributions of FISH markers for aminergic transmission[37], which show only a few cells with clear labelling in or near the optic lobes or, in the case of dopamine, include many small cell bodies in the medulla cell body rind but suggest (through co-labelling with a split-GAL4 driver) that these belong primarily, if not exclusively, to Mi15 neurons[37]. Supplementary Table 5 lists neurotransmitter data for 88 cell types that were not included in the training dataset. The 78 evaluated as 'validation with new experimental data' in Fig. 4e are based on this set, excluding the uncertain types ([Cm11] and [Pm2]; the square brackets denote potential expression in a group of cell types), those with co-transmission or unclear labelling (Mi15, CL357, l-LNv, MeVC27, OLVC4 and T1), and assigning histamine to HBeyelet, R8p and R8y and counting the latter two as one type.

## Neurotransmitter prediction

We predicted the identity of the neurotransmitter for 7,014,581 presynapses in the optic lobe dataset using an existing method[41] (Fig. 4). We first trained an image classifier network on EM volumes ($640 \times 640 \times 640$ nm$^3$) to predict each presynaptic site as one of seven possible neurotransmitter types (and their common abbreviations): acetylcholine (ACh), glutamate (Glu), GABA, histamine (His), dopamine (Dop), octopamine (OA) and serotonin (5HT). The ground-truth neurotransmitter types are described in the section 'Training and evaluation data for neurotransmitter predictions' and were based on previous experimental data, detailed in Supplementary Table 5. Entire neurons (for ACh, Glu, GABA and His) or individual synapses (owing to the low number of neurons for Dop, OA and 5HT) were partitioned into disjoint training, validation and testing sets of 70%, 10% and 20%, respectively, optimizing for similar class frequency in each partition. Models included an additional class to indicate nonsynaptic or unrecognized structures, trained by sampling random locations in the bounding box of all synaptic locations.

In neuPrint, the computed probability of each neurotransmitter type is attached to the presynapses and can be queried using the following properties: ntAcetylcholineProb, ntGlutamateProb, ntGabaProb, ntHistamineProb, ntDopamineProb, ntOctopamineProb and ntSerotoninProb. The values range from 0 (impossible) to 1.0 (certain), sum to 1, and should be interpreted as relative probabilities. The accuracy of the synapse-level neurotransmitter predictions is assessed in Fig. 4b. Although the training data were restricted to high-confidence ($\geq 0.9$) synapses, for evaluating the performance of the trained network, we produced a confusion matrix (Fig. 4b) showing classification results on the testing dataset and an additional 1,014,151 synapses with a detection confidence of <0.9.

In addition to these synapse-level predictions, we provide aggregated neurotransmitter predictions (with quality controls applied, as explained below) at the neuron and cell-type level, which are stored in neuPrint as properties predictedNt and celltypePredictedNt, respectively. First, we computed the neuron-level neurotransmitter confidence score as previously described[41] and assigned the most frequent presynaptic neurotransmitter as the prediction of the neuron if that cell has $\geq 50$ presynapses and confidence $\geq 0.5$. Similarly, we computed

the cell-type-level confidence score and assigned the most frequent presynaptic neurotransmitter as the prediction for each cell type if the type has $\geq 100$ presynapses (summing over all neurons of the type) and confidence $\geq 0.5$. Individual neurons and cell types that did not meet these criteria were labelled with 'unclear' neurotransmitters. Finally, we provide a consensus neurotransmitter type (in neuPrint, property consensusNt). Cell types with predictions for Dop, OA or 5HT that were not supported by the high-confidence experimental data listed in Supplementary Table 5 had their consensus type set to unclear, as these neurotransmitters were under-represented in the training data and independent measurements of the abundance of these transmitters did not match their prevalence in the predictions. The consensus neurotransmitters are reported in the Cell Type Explorer web resource and the Cell Type Catalogue (Supplementary Fig. 1), and form the basis for the summary data in Fig. 4.

To assess the significance of the fan-out values of cell types predicted to signal with different neurotransmitters, we conducted an ANCOVA, which combines the analysis of variance with regression analysis. We used the Python package statsmodel[105] to test whether the categorical variable, the neurotransmitter type, significantly affects the slope of the linear regression (Fig. 4i).

## Number of innervated columns of a cell type

The column pins we have built for the medulla, the lobula and the lobula plate provide a framework for quantifying the size of visual system cells by measuring the number of columns a neuron has synapses in as a function of depth (Fig. 5 and Supplementary Fig. 1). We did this in two different ways. A direct method, which is used in Fig. 5b and the Cell Type Catalogue (Supplementary Fig. 1), takes advantage of the pin points stored in neuPrint to assign synapses to depths along each pin, and this method is described in the section 'Summary of connectivity and size by depth'. We also wanted to summarize the size of neurons in each neuropil with a single number, but simply summing the column assignments of all synapses across depth led to overestimates of cell size that did not match our expectations. We therefore implemented a trimming method, explained below, that was used as the basis of the size and coverage analysis in Fig. 5d–f, Extended Data Fig. 10 and the Cell Type Explorer web resource.

The trimming procedure was implemented to solve a problem we frequently encountered: columnar neurons like L1 or Mi1 in the medulla have a clear home column (the column with the largest synapse count summed over depth), but a few synapses were often sprinkled away from their home column. As a result, when summed over depth, the direct column count would typically be larger than we intuitively expect (for example, columnar neurons should be of size about 1 in units of columns). To correct for this overestimation, we trimmed these excessive synapses using the following steps:

1. For each neuron, we ranked the columns by the number of synapses they contain (summed over depth), with the main column having rank 1 (columnar cells have a clear home column, but many cell types do not). The cumulative fraction of the number of synapses as a function of rank has a convex shape that eventually flattens out around 1, and the curve was similar for most neurons of the same cell type.
2. We then took the median of the cumulative fraction curve across neurons and used a knee/elbow finder to identify the rank (rank*) at which the curve defined by the median cumulative fraction versus rank (with the point (0,0) added) had the highest curvature. We used the knee/elbow finder algorithm kneedle[106], for which we specified that the curve is concave and increasing, and we used the default parameter $S = 1$.
3. If the median cumulative fraction at rank* was less than 0.775, we took rank* to equal the rank for which the median cumulative fraction was at least 0.995 (or, if this value was not reached, its maximum).
4. Then, for every neuron separately, we discarded all synapses in columns with synapse count less than that in the column of rank rank*.

If all columns had unique synapse counts, this is the same as discarding all synapses in columns of rank larger than rank*.

After trimming, we counted columns and then averaged these distributions across neurons of the same cell type.

### Spatial coverage of optic lobe neuropils for cell size, coverage factor and completeness

Spatial-coverage metrics were calculated per neuron type for each major optic lobe neuropil (that is, the medulla, the lobula and the lobula plate) (Fig. 5). Metrics were calculated separately for neurons from the same neuron type depending on the hemispheric location of their somata by querying the neuPrint database for their instance (for example, aMe12_L or aMe12_R). Individual synapses were assigned to columnar hexagonal coordinates (see the section 'Defining column centre lines and ROIs') using their assignment in the neuPrint database. We used the trimming procedure (see above) to summarize the cell size in column units or, conversely, the number of cells per column. In all other cases, we used the direct column occupancy data.

The spatial patterns are represented on the hexagonal eye map introduced in Fig. 3a as heatmaps of the number of cells and synapses per column. Raw, untrimmed column occupancy data were used to summarize the number of synapses per column, whereas the trimmed data were used when plotting the number of cells per column (Fig. 5d and the Cell Type Explorer web resource). The number of synapses per column reflects the sum of both presynapses and postsynapses in that column. Conversely, when calculating the number of synapses across all neuron types in a given optic lobe neuropil (Fig. 3e), only postsynapses were included to avoid double counting presynapses and postsynapses at the same connection. The maximum colour scale value was set as the 98th percentile value of the highest value across the medulla, the lobula and the lobula plate for both plots (Cell Type Explorer web resource).

We quantified the cell size (Fig. 5e, Extended Data Fig. 10 and Cell Type Explorer web resource) as the median number of columns innervated by neurons of the type in a designated optic lobe neuropil after synapses had been trimmed per neuron based on the trimming procedure described in the section above.

To examine the spatial overlap between individual neurons of the same type, we calculated the coverage factor (Fig. 5e,f, Extended Data Fig. 10 and Cell Type Explorer web resource) for each optic lobe neuropil. This metric was calculated as the mean number of neurons that contribute synapses to a single column across all occupied columns of the specified optic lobe neuropil after trimming using the above-mentioned method.

To assess the proportion of total columns innervated by a neuron type, we calculated the 'columnar completeness' per neuropil for each type to describe the proportion of total columns in each optic lobe neuropil that are occupied by synapses from all neurons of the type (for this calculation, we used raw column occupancy data without trimming). This metric is provided as a count of innervated columns in Fig. 5f and Extended Data Fig. 10 and as the proportion of total columns (0–1) in the Cell Type Explorer web resource.

The extent to which neuron types densely or sparsely innervated a particular optic lobe neuropil was investigated by comparing the total number of columns innervated by neurons of that type, related to the 'columnar completeness' described above, and the columnar area covered by all neurons of the type. For most cell types, the columnar area was found by fitting a convex hull around the hexagonal coordinates based on their assignment in neuPrint and calculating its area. For the cell types in which the convex hull was a poor approximation of the area covered, we instead counted the innervated columns. This included aMe2, aMe10, AN27X013, AN09A005, Cm-DRA, MeLo11, MeTu4f, MeVP15, MeVP20, MeVPMe8, Mi16, R7d, R8d, TmY19b and LC14b in the medulla, LC14a-1, LC14a-2, LC14b, LC31a, Li37, LoVC17, LoVP12, LoVP26, LoVP49, LoVP92, LPT31, LT80, LT81 and TmY19b in the lobula, and LPT31, LC14b and LPT100 in the lobula plate. The columnar area metric (Fig. 5f, Extended Data Fig. 10 and Cell Type Explorer web resource) enabled us to quantitatively distinguish cell types that densely covered fractional parts of a neuropil from types that sparsely covered the entire neuropil.

### Summary of connectivity and size by depth

To produce the synapse distribution by depth, featured prominently in the Cell Type Catalogue (Fig. 5 and Supplementary Fig. 1), we first identified all synapses of each target cell from the designated cell type. Using a $k$-dimensional tree, we identified the nearest neighbouring column pin by Euclidean distance for each presynapse and postsynapse. The depth property of the column pin associates each synapse with one of the 121 medulla, 76 lobula or 51 lobula plate depth bins (see the section 'Defining column centre lines and ROIs'). Across all neurons of a cell type, we calculated the mean count separately for presynapses and postsynapses per depth. We smoothed the distribution per optic lobe neuropil with a Savitzky–Golay filter (window size of 5 and first-order polynomials) before plotting on the left-side panel of the summary data figures. Separately, we provide the percentage of synapses located in the AME over the total number of synapses of a cell type, which are again separated by presynapses and postsynapses.

To quantify the size of each cell type as a function of depth, we identified the depth for all synapses. As columns are implemented as ROIs in neuPrint (see the section 'Connectome data access overview'), each synapse comes with an associated column. For each cell type, we calculated the mean count of columns with synapses (combining presynapses and postsynapses) per depth and smoothed the distribution per optic lobe neuropil with a Savitzky–Golay filter (window size of 5 and first-order polynomials) before plotting on the right-most panel of the summary data.

The centre panel of the cell-type summary contains a connectivity summary. For all neurons of a target cell instance (effectively type, but separately accounting for right and left instances), we found all synapses and their connecting partner cells. After removing unnamed segments and connections between neurons with a mean count ≤ 1.0, we ordered them by the fraction of input and output connections and show the top five.

### Morphology clustering

To test whether the 68 medulla ONIN cell types with more than 10 cells per type (11,102 neurons) mentioned in the section 'Connectivity-based cell clustering' (Extended Data Fig. 3) could be split into cell types by their morphology alone, we assigned each neuron a feature vector of length 244 (Fig. 5 and Extended Data Fig. 8). The features captured the counts of presynapses and postsynapses and their size distributions across depths of the medulla, similar to the first and third columns in Fig. 5b, but for individual neurons instead of cell types (and for the medulla only). Moreover, the number of innervated columns was split into presynapse and postsynapse innervations and was calculated directly from the pins (introduced in the section 'Defining column centre lines and ROIs' rather than the column ROIs stored in neuPrint). We also subsampled the depth by a factor of 2 (as described in the section 'Layer ROIs'). We used a confidence of 0.9 for the included synapses, based on the expectation that higher confidence synapses would produce more robust feature vectors, but we did not systematically explore this parameter.

The first 61 features are the number of presynapses across the depths of the medulla, normalized to the number of innervated columns of presynapses (with no trimming). This normalization meant that we multiplied the number of presynapses across depths of the medulla with the same factor $N_{pre,size}/N_{pre,syn}$, where $N_{pre,syn}$ equals the sum of the number of synapses across depths in the medulla, and $N_{pre,size}$ equals the sum of number of innervated columns across depths in the medulla. Although this normalization step is not biologically motivated, it follows the

common strategy of bringing different features to the same scale to compare them. The second 61 features were similar to the first 61 but for postsynapses instead of presynapses. The third 61 features were the untrimmed number of columns innervated by presynapses (without further normalization). The final 61 features were the number of columns innervated by postsynapses. Example feature vectors for three neurons are shown in Extended Data Fig. 8a.

For the clustering step, we used the same standard library described in the section 'Connectivity-based cell clustering' but used Euclidean distance as the metric for comparing feature vectors. Again, we made a flat cut in the hierarchical clustering diagram to obtain 80 clusters. The corresponding confusion matrix and the completeness and homogeneity scores[107] are shown in Extended Data Fig. 8b.

### Confusion matrices of cell-type clustering

To facilitate comparisons of the connectivity and morphology clustering process, we present the results as identically formatted confusion matrices in Extended Data Figs. 3a and 8b. The numbers in the confusion matrices are the nonzero numbers of neurons for a given cell type in each cluster.

The rows (cell-type names) are ordered lexicographically. The columns (cluster identities) are ordered by the number of neurons in a cell type. Starting with the first cell type, we picked the first cluster as the one with the largest number of neurons of that cell type; the second cluster is the cluster with the second largest number; and we continued picking clusters until the number of neurons in each remaining cluster was 5% or less for that cell type. Then we continued picking from the remaining clusters using the second cell type with an analogous procedure as for the first cell type. The colours in the confusion matrices correspond to five nonoverlapping categories:

1. Red: 1-to-1 clusters in which at least 80% of neurons of one cell type are in 1 cluster, and at least 80% of neurons in that cluster are of that cell type.
2. Blue: many-to-1 clusters in which at least 80% of neurons of 1 cell type are in 1 cluster, but less than 80% of neurons in that cluster are of that cell type.
3. Green: 1-to-many clusters in which less than 80%, but at least 10% of neurons of 1 cell type are in a cluster, but at least 80% of neurons in those clusters are of that cell type.
4. Yellow: mixed clusters in which less than 80% but at least 10% of neurons of 1 cell type are in a cluster, and less than 80% but at least 10% of neurons in those clusters are of that cell type.
5. Grey: outliers in which everything else is nonzero.

These categories are also described as probabilities in Extended Data Figs. 3a and 8b.

### Cell Type Explorer web resource analysis

We provide a set of interactive, interlinked web pages to facilitate quick browsing of connections between all the cell types in our inventory (Extended Data Fig. 9). Each cell type is detailed on a single web page, except for the cases of bilateral neurons with both left and right instances in the right optic lobe. These cell types are described on two pages, one per instance. The web page for each cell type provides the mean presynapses and postsynapses in selected ROIs. The synapses in the medulla, the lobula and the lobula plate layers were determined by querying synapses in the specified layer ROIs (described in the section 'Layer ROIs') for each neuron in the optic lobe. The lamina and accessory medulla do not have layers, so the synapses were queried for the ROI. In the central brain, queries were conducted on all primary ROIs, excluding the left and right optic lobes. These synapses are then presented as the mean values across neurons of a given type in each of these ROIs.

The coverage factor, columnar and area completeness, and cell size (in columns) were quantified for each of the medulla, the lobula and the lobula plate, as described in the section 'Spatial coverage of optic lobe neuropils for cell size, coverage factor and completeness'.

The connectivity table displayed on the web pages was generated by fetching all connecting neurons either upstream (inputs and T-bars) or downstream (outputs and postsynaptic densities) from the cell type named on the web page. It lists the total number of connections between the input and output cell type and the titular cell type (total connections), the total connections divided by the number of the cell count of the titular cell type (connections/[cell type]), and provides both the percentage and cumulative percentage of the total connection count. The table displays all connections between neurons with a mean count of >1.0. Most cells have a long tail of weak connections that are not displayed on the web pages. All connections, including the omitted weaker connections, can be found in neuPrint.

The Cell Type Explorer is available at https://reiserlab.github.io/male-drosophila-visual-system-connectome/ and for download as a set of static html pages at Zenodo (https://doi.org/10.5281/zenodo.10891950).

### Gallery plots

For the gallery plots in Supplementary Fig. 1 (and renderings of selected neurons throughout), we used the optic lobe slice views detailed in the section 'Pipeline for rendering neurons'. We established three camera orientations in Blender[78] that produced three slices oriented equatorially, dorsally and ventrally, labelled in panels E, D and V, respectively. Over multiple rounds of manual curation, we selected representative neurons that captured many features of each cell type (when there are more than one to choose from). We chose cells with a primary orientation close to one of the three slices. An additional consideration was to select cells that, when projected along these camera views, had cell bodies that could be visualized outside the optic lobe neuropils wherever possible (cell bodies are generally outside the neuropils; this is simply an artefact of projecting 3D bodies onto 2D images). The slice width used for the neuron projection varied with the morphology of the individual cell. For cells that required a thicker slice, the layer pattern of the projection was affected as the layers are curved, an effect most prominent for neurons rendered on the dorsal and ventral slices. Therefore, although every effort was taken to show neurons with characteristic layer innervation patterns, the accompanying summary of synapses by depth should be used as the characteristic layer innervation pattern of the cell type. For a small number of cells with morphology not captured by the three standard slices, we used a combination of custom slices to generate projections on the dorsal, equatorial or ventral slice views that faithfully conveyed the morphology of the cells. The JSON descriptions of these customized views are in the 'src/gallery_generation' directory. For example, we used this strategy to capture the vertical cell-body process for the Cell Type Catalogue of the Mi20 cell type. We refer to the set of selected neurons for each cell type as the 'star neurons', and the parameters used for rendering the projections of the neurons, including the slice used and slice thickness (width), are listed in the 'params/all_stars.xlsx' file in the code repository (also see the section 'Connectome data access overview').

### Split-GAL4 lines that are expressed in visual system neurons

Supplementary Table 6 summarizes our collection of split-GAL4 lines for visual-system neurons. Most driver lines are newly reported here, and we include lines from our previous publications[21,23,26,33–35,38,44,45,48,58,87,91,108,109] and another that is in the process of publication (H. Dionne et al., manuscript in preparation). Further details on the construction and annotation of these lines are described below. Supplementary Table 6 includes annotations of the primary optic lobe cell type (or types) that each line drives expression in. Images used for characterizing the lines are available at https://splitgal4.janelia.org/cgi-bin/splitgal4.cgi. Split-GAL4 lines currently maintained as fly stocks (the majority) can be requested through the same website; many are also available from

the Bloomington Stock Center. Most remaining lines could be remade using publicly available AD and DBD hemidrivers.

Newly reported split-GAL4 driver lines were generated as previously described[38,45,110]. In brief, we selected candidate hemidriver (AD and DBD) combinations based on images of GAL4 line expression patterns (overall expression patterns[109,111] or MCFO-labelled individual cells[19,112]) and tested whether these combinations could drive the expression of a UAS reporter specifically in cells of interest. Promising AD–DBD combinations were combined into stable fly stocks, and the expression patterns were retested. For detailed characterization, we examined images of overall expression patterns in the brain and VNC at low resolution (×20 objective) and, for most lines, additional high-resolution (×63 objective) images of cell populations, MCFO-labelled individual cells or both. Most split-Gal4 screening, sample preparation and imaging were performed by the FlyLight Project Team using established methods[19,45,110,113] and publicly available protocols (https://www.janelia.org/project-team/flylight/protocols). The FlyLight split-GAL4 project has been summarized in a recent publication[114].

Images shown in Fig. 7 and Extended Data Figs. 14 and 15 were displayed using VVD viewer[115]. Details of the underlying genotypes (for example, reporters used) and images (for example, objective used) for each panel are included in Supplementary Table 6 (column 'Figure details') with the following abbreviations: MCFO-1 and MCFO-7 (ref. 19), 20xChr (for 20XUAS-CsChrimson-mVenus in *attP18* (ref. 116), syt-HA (for pJFRC51-3XUAS-IVS-Syt::smHA in *su(Hw)attP1*) and myr-smFLAG (for pJFRC225-5XUAS-IVS-myr::smFLAG in *VK00005*)[19,110]. The last two transgenes were used as a combined stock, but most images only show the myr-smFLAG reporter (as indicated in the table). For overlays of EM skeletons and LM images, both were registered to a template brain (see the section 'LM–EM volume correspondence'; JRC2018U)[55] and overlayed in VVD viewer. In most cases, images show projections through the full brain in an anterior view or through a subvolume selected and oriented to display the layer pattern in the optic lobe. Some LM images are of left optic lobes, mirrored to match the location of the EM skeletons in the right optic lobe.

Assignment of LM images to EM cell types was done by visual comparison of LM and EM morphologies at multiple levels of detail. The anatomical features used for LM–EM matching are illustrated in Fig. 7 and Extended Data Figs. 14 and 15. Supplementary Table 6 lists matched cell types with the names used in the Cell Type Catalogue (Supplementary Fig. 1) and throughout this article. We cannot provide confident 1-for-1 matches between optic lobe cells expressed in every driver line and EM-defined cell types. Some LM–EM matches are to groups of cell types rather than to single types, whereas some represent more tentative candidate matches. This is usually the case for cell types that are primarily distinguished by EM connectivity and/or by relying on features that are difficult to assess with the available LM images. Such cases are identified in Supplementary Table 6 as follows: cell type names in square brackets denote expression in one or more of a group of cell types, and type names in parentheses indicate lower confidence matches in a group of annotations. For example, LC31a,(LC31b) means that we are confident a driver is expressed in LC31a and probably also LC31b. [Pm2a,(Pm2b)] describes a driver with expression in Pm2a or Pm2b, or both, but for which we consider expression in Pm2a more likely than expression in Pm2b. In a few cases, primarily for brevity, we list groups of cell types under a single name, for example, [Tm] for expression in as yet unidentified Tm cells or T4 for expression in T4a, T4b, T4c and T4d.

The completeness of the EM cell-type inventory greatly facilitated the LM–EM matching efforts. Sexually dimorphic neurons could present limitations to this completeness, but we found little evidence for them. Most images used to annotate the split-Gal4 collection are of female flies, whereas this EM reconstruction is of a male optic lobe. However, nearly every visual system cell type in the LM images of most lines could be readily matched to the EM-defined cell types, with one exception mentioned in the main text.

The cell-type annotations in Supplementary Table 6 are intended to facilitate the identification of driver lines that may be useful for experiments on specific optic lobe neurons of interest. Because driver lines with expression in some off-target cell types can still be valuable for many experiments, we included some lines with additional expression in the collection. Our annotations generally do not include these off-target cell types in the central brain, VNC and, in some cases, other (minor) visual-system expression patterns. The images used for annotating the lines are available online (see above and the Data availability section) and could be consulted if details of any additional expression may prove crucial for a planned experiment.

## Matching cell types between the male optic lobe and FlyWire datasets

To facilitate comparison between the male optic lobe dataset (denoted OL in this section) and the female FlyWire FAFB datasets[15–17,24], we matched cell types across datasets (Extended Data Fig. 16 and Supplementary Table 7). This matching was performed after the completion of cell typing in the OL dataset. The only exception to this are a small number of cell-type changes in the most recent optic lobe dataset release (optic-lobe:v1.1); in several cases, our decision to make these changes was influenced by cell-typing choices made in the FlyWire dataset (see the section 'Versions of the dataset'). The methods used for matching are summarized below. A full listing of the cell-type matches is provided in Supplementary Table 7 and summarized in Extended Data Fig. 16a. We note that this is a nearly complete, yet preliminary, matching of the cell types. We anticipate that upcoming central brain and left optic lobe connectivity data in the male brain, along with advances in neuron annotations in FlyWire, will contribute to further validation and refinement of the cell-type matches between the datasets.

Supplementary Table 7 contains the cell-type name given in the OL dataset and the name (or names) for the corresponding cell types in FlyWire. FlyWire cell types are based on previous publications[16,24], and, in some cases, two different type name annotations, one from each reference, are provided. We note that additional annotations for some FlyWire cells beyond those included in these cell-type references (for example, previously described subdivisions of MeTu neurons[48]) are not included in our comparison.

For most cell types, we matched cells between the datasets as one-to-one matches. However, around 13% of types were matched as many-to-one, one-to-many or many-to-many. One-to-many matches are cases in which a single optic lobe cell type was split into multiple types in FlyWire, whereas many-to-one indicates a correspondence between multiple optic lobe types and a single Flywire type. Examples of a one-to-one match and a two-to-one match are shown in Extended Data Fig. 16b. For ONIN and ONCN cell types for which there were both previously described types[16,24], the classification of the matches refers to the match between the OL dataset and previously described types[24]. For six VPN cell types, the matches were classified as many-to-many because these cell types could be matched as groups in each dataset, but we were not confident about the matches between individual types in these groups. We also identified several cell types, classified as 'unmatched', that seemed to be present in one dataset but absent from the other. Some of these types are strong candidates for male-specific or female-specific neurons, but others probably represent biological variability between the two flies or technical variability owing to incomplete reconstruction of some cells (examples in Extended Data Fig. 16c).

Supplementary Table 7 also includes cell-type counts for each dataset (with counts for the two sets of FlyWire type annotations listed separately). Previously described annotations of the FlyWire dataset[16] cover most of the cells on both the right and left brain hemispheres, and in these cases, we provide the counts from both sides. The table also includes neurotransmitter predictions from both datasets. The listed predictions for the optic lobe cell types are the consensus predictions (from neuPrint; provided in Supplementary Table 1 and described in the

section 'Neurotransmitter prediction'). For FlyWire, the table lists the most frequently predicted neurotransmitter[41] across the neurons of a given type (union of the previously published annotations[16,24]). FlyWire types for which the top two predicted neurotransmitters are tied are labelled 'unclear'. Because some hemibrain types were renamed in our study (to provide more systematic names to optic lobe neurons), we include hemibrain matches in Supplementary Table 7 based on the previously reported hemibrain–FlyWire matches[16].

To match cell types, we first used NBLAST to identify morphologically similar neurons between the FlyWire and the OL datasets following a previously described procedure[16] for hemibrain–FlyWire matching. For FlyWire, we used previously prepared skeletons[17] for all neurons, whereas for OL, we downloaded skeletons from neuPrint (release optic-lobe:v1.0.1). Skeletons were pruned to remove twigs smaller than 5 μm and then transformed from OL into FlyWire (FAFB14.1) space using a combination of non-rigid transforms. Once in FlyWire space, they were resampled to 0.5 nodes per μm of cable to approximate the resolution of the FlyWire L2 skeletons and then turned into dotprops. NBLAST was then run both in forward (FlyWire→OL) and reverse (OL→FlyWire) direction, and the minimum score between both was used.

From the NBLAST scores, we extracted for each FlyWire neuron a list of potential OL-matching neurons based on the top five hits. A sample of individual FlyWire neurons for each type was co-visualized with their potential OL hits in neuroglancer, and the corresponding cell-type match was recorded. Although the registration of the optic lobes of the two datasets is imperfect in places, this approach was sufficient to identify candidate matches at the cell-type level. All neurons for matched cell types were then co-visualized to confirm the match, which verified that the total number of neurons was consistent between datasets. Although this approach was sufficient to match most cell types across datasets, in a few cases, we also used the output of connectivity co-clustering (with coconatfly or cocoa, see below) to identify and confirm matches. The primary use cases were morphologically similar ONIN and ONCN types, mainly among those that had only been typed unilaterally[24] and were not present in a previously published typing[16].

Neuroglancer scenes were prepared with co-aligned OL and FlyWire meshes. The static FlyWire meshes from materialization 783 (ref. 17) were transformed to OL space (using the transformation to the male brain described in the section 'LM–EM volume correspondence' for the final step). Cell-type annotations for OL neurons were sourced from neuPrint (release optic-lobe:v1.0.1). In a small number of cases, the matches were updated to reflect changes in optic-lobe:v1.1 (see above and the section 'Versions of the dataset)'. FlyWire cell-type annotations were sourced from GitHub (https://github.com/flyconnectome/flywire_annotations)[16]. We also matched intrinsic and connecting neurons (ONIN or ONCN) to the cell types as previously described[24], as downloaded on 24 April 2024 from https://codex.flywire.ai/api/download?data_product=visual_neuron_types&data_version=783. For convenience, a neuroglancer annotation layer was prepared to previously published cell types[16,24]. At the time of writing, the authors of ref. 24 described more ONIN and ONCN cell types than another study[16] but only provided them for one optic lobe in FlyWire. These scenes are provided with a unique URL per OL cell type in Supplementary Table 7. For the ONIN and ONCN cell types with annotations from ref. 16 and ref. 24, the linked neuroglancer scenes include both FlyWire types. Subsets of the displayed cells can be selected through regular expression searches of the scenes.

Skeleton and mesh processing, spatial transforms and NBLAST were performed using the navis[81] and fafbseg-py Python packages (https://github.com/navis-org/navis). Connectivity co-clustering was performed using coconatfly (https://natverse.org/coconatfly) and cocoa (https://github.com/flyconnectome/cocoa). Code and neuroglancer links are available from GitHub (https://github.com/flyconnectome/ol_annotations). Connectivity data for FlyWire neurons can be obtained from the FlyWire CAVE annotation backend, for example, through https://fafbseg-py.readthedocs.io/en/latest.

A separate team member independently reviewed the initial set of matches obtained as described above. Similar to the initial matching, this review primarily relied on comparisons of morphology through visual inspection of the neuroglancer scenes. For a minority of types, mainly in the ONIN and ONCN groups, which are difficult to distinguish by morphology alone, we also reviewed connectivity patterns. This was done by comparing aggregate cell type to cell-type connectivity or connectivity clustering or co-clustering (using adaptations of the approach described in the section 'Connectivity-based cell clustering'). Clustering was consulted in some cases as it provides a quick overview of cell-level matches and includes untyped cells. Overall, this review resulted in minor adjustments (for example, identifying untyped candidate matches for some initially unmatched types, indicated in the 'notes' column of Supplementary Table 7) of the initial matches.

The comparison of predicted neurotransmitters for the matched cell types provides yet another independent evaluation of both the putative matches and the neurotransmitter predictions (in two independent EM volumes, with independent training data, but using the same method as previously described[41]). In general, the concordance was very high. We used a conservative policy for reporting consensus transmitter predictions in the OL dataset. Therefore, 170 out of the 732 cell types have unclear predictions (132 of which are VPNs). Of the remaining OL cell types, there were 19 with either unclear predictions in FlyWire and/or no cross-dataset matches so that no neurotransmitter predictions could be compared. Once these cell types were excluded (without double counting), there were 544 cell types for which predictions could be evaluated, of which 503 agreed (92.5%). Of the 41 mismatched predictions, 8 were OL predictions for histamine, which was not included in the FlyWire training or predictions. Of the remaining 33 mismatched predictions, 26 were OL predictions for glutamate and FlyWire predictions for GABA (notably, there are no mismatched predictions for the opposite assignments).

## Brain region connectivity

This analysis (Fig. 7) aimed to quantify the amount of connectivity among brain regions with a weighted sum of their connected neurons. The brain regions considered in Fig. 7h,i are the ROIs from the optic lobe neuropil layer (and the AME) and prominent grouped central brain regions.

If one neuron received input connections from brain region A and sent output connections to brain region C, we asked how much that neuron contributed to the A → C connectivity. We first counted the output connections in C, then computed the fraction of input connections in A. We considered the product of the count and the fraction as the contribution of this neuron to the A → C connectivity. For example, a neuron X has 10 upstream connections in region A and 20 in region B, and 30 downstream connections in region C and 15 in region D. Then contributions of neuron X to inter-region connectivity are 10 for A → C, 20 for B → C, 5 for A → D and 10 for B → D.

Using the visual-system neuron groups (defined in the section 'Cell-type groups'), we analysed the inter-region connectivity of all ONINs, ONCNs and VPNs separately. Each is represented by one connectivity matrix. For the ONIN and ONCN analyses, we examined the connectivity between layer regions: medulla layers {1, 2, 3, 4, 5, 6, 7, 8, 9 and 10}, lobula layers {1, 2, 3, 4, 5A, 5B and 6} and lobula plate layers {1, 2, 3, 4}, where curly braces indicate sets. For the VPNs, we analysed connectivity between these 21 optic lobe regions and central brain regions. Several brain regions have been grouped into super regions to simplify this summary[61]. The regions are abbreviated in Fig. 7h, and many are illustrated in Fig. 7i (projection view) with the following abbreviation: anterior optic tubercle (AOTU), anterior ventrolateral protocerebrum (AVLP), posterior lateral protocerebrum (PLP), posterior ventrolateral protocerebrum (PVLP), wedge (WED), periesophageal

neuropils (PENP), inferior neuropils (INP), lateral complex (LX), mushroom body (MB), central complex (CX), ventromedial neuropils (VMNP), gnathal ganglia (GNG), lateral horn (LH) and superior neuropils (SNP).

### Versions of the dataset

Two releases of the optic lobe dataset are available: the initial release (optic-lobe:v1.0.1) and the current version of the dataset (optic-lobe:v1.1). optic-lobe:v1.1 incorporates updates to the segmentation (mainly some merges of small, previously unconnected parts of reconstructed segments with these neurons); these changes collectively result in a small but detectable increase of reconstruction completeness (Extended Data Table 1). We also revised some cell-type definitions (5 types were split into a total of 11 new types and 2 types were merged into a single type). Several of these cell-type changes were based on observations made after comparing cell types between datasets (see the section 'Matching cell types between the male optic lobe and FlyWire datasets') and further improve the consistency of cell typing between this and other studies[16,24,33]. In addition, we changed the names of a few cell types (for example, to introduce names that better reflect the considerable similarity of some cell type pairs) and adjusted the typing of some individual cells (nearly all of these are R7 and R8 photoreceptors, see the section 'Assigning R7 and R8 photoreceptors and medulla columns to different types of ommatidia'). In the data presented in the article, all layer and column ROIs that are used in subsequent analyses are based on the optic-lobe:v1.0.1 data and were not regenerated after the minor updates to optic-lobe:v1.1. All rendered neurons shown in the figures, with the exception of those in Fig. 3b and Extended Data Figs. 8a and 16b, are based on the v.1.1 meshes.

### Fly stock availability

Fly stock availability is detailed in Supplementary Table 6. Most split-GAL4 driver lines are available at https://splitgal4.janelia.org/cgi-bin/splitgal4.cgi. Moreover, many lines have been deposited in the Bloomington *Drosophila* Stock Center (BDSC). BDSC Research Resource Identifiers for these stocks are included in the table (column 'BDSC RRID'). For most of the additional lines that are currently not maintained as combined stocks, the component AD and DBD hemidrivers are available from the BDSC.

### Reporting summary

Further information on research design is available in the Nature Portfolio Reporting Summary linked to this article.

### Data availability

The connectome data are directly accessible through the neuPrint database server (https://neuprint.janelia.org/?dataset=optic-lobe:v1.1). The Cell Type Explorer web resource is available at https://reiserlab.github.io/male-drosophila-visual-system-connectome/index.html and can also be downloaded as a zip file from Zenodo (https://doi.org/10.5281/zenodo.10891950)[117]. The SWC skeleton is available through neuPrint+ (a web interface) by clicking the bodyId in the skeleton viewer, which will provide a download button. For bulk downloads, the meshes of neurons, ROI boundaries and skeletons are provided as Google Cloud Storage buckets at gs://flyem-optic-lobe. In our shared code (see the Code availability section) we provide programmatic access to these files. For example, the skeletons and meshes of individual neurons are accessible through the OLNeuron class. The example Jupyter notebook (https://github.com/reiserlab/male-drosophila-visual-system-connectome-code/blob/main/src/python-bootcamp/access_skeleton_and_mesh.ipynb) shows how to store the skeleton as a *.swc file and the mesh as a Wavefront *.obj file. The LM–EM transformation vectors from our EM sample to the JRC2018M template brain are available from Figshare (https://doi.org/10.6084/m9.figshare.24002001.v1)[118]. In that file, the 'dfield' transformation vectors map points from EM space to LM template space, and 'invdfield' vectors map points in the opposite direction. Images of split-GAL4 driver lines are available at https://splitgal4.janelia.org/cgi-bin/splitgal4.cgi.

### Code availability

The Python source code to replicate our analyses and visualizations follows established best practices for computational analysis[119]. In the version-controlled code repository (https://github.com/reiserlab/male-drosophila-visual-system-connectome-code), we document dependencies such as Pandas[120], NumPy[121], SciPy[92], Jupyter[122], Plotly[123], Snakemake[84] and Trimesh[124]. In the code repository's 'docs' directory, we share a range of technology-specific 'getting started' guides. This includes a guide for setting up the required python-3.10 (or newer) and our software on Linux, Windows or Mac computers, running simple analysis and using the workflow management system Snakemake to replicate our figures.

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

**Acknowledgments** Janelia's FlyEM Project Team (https://www.janelia.org/project-team/flyem) acquired and reconstructed the EM dataset in collaboration with the Connectomics group at Google. We thank staff at the Janelia Research Campus for their dedicated support and FlyEM team members, in addition to the named authors who contributed, including Steering Committee members G. Card and V. Jayaraman; G. Meissner for help preparing and imaging fly lines; M. Eddison, Y. He and G. Ihrke for performing EASI-FISH and FISH experiments; staff at Janelia's Fly Core for fly care; S. Turaga for support of N.C.K.; B. Mensch for feedback on the manuscript; C. Goina, H. Otsuna, K. Rokicki, R. Svirskas and staff at the Janelia Scientific Computing Software for their work on confocal image processing, storage and public release to Janelia websites; staff at the Janelia's Open Science Software Initiative for supporting the maintenance of ImgLib2 and BigDataViewer; and the supportive Janelia Fly community and members of the 'FAFB optic lobe working group', especially members of the Behnia, Chiappe, Silies and Wernet groups for discussions. This work used VVDviewer, based on software funded by the NIH: Fluorender: An Imaging Tool for Visualization and Analysis of Confocal Data as Applied to Zebrafish Research, R01-GM098151-01. This work was supported by the Howard Hughes Medical Institute and the Wellcome Trust (220343/Z/20/Z and 221300/Z/20/Z to G.S.X.E.J. with G.M.R., G. Card, S. Waddell and M. Landgraf). This article is subject to HHMI's Open Access to Publications policy. HHMI lab heads have previously granted a nonexclusive CC BY 4.0 licence to the public and a sublicensable licence to HHMI in their research articles. Pursuant to those licences, the author-accepted manuscript of this article can be made freely available under a CC BY 4.0 licence immediately upon publication.

**Author contributions** M.B.R., A.N. and G.M.R. drafted the manuscript with input from S.B., L.E.B., E.G., J.H., G.B.H., K.J.H., M.J., N.C.K., S.K., F.L., K.D.L., S.P., S.R., G.M.R., L.K.S., P. Seenivasan, S.S. and A.Z. S.B., J.B., L.E.B., J.C., M.D., J.F., E.G., I.H., J.H., G.B.H., P.M.H., M.J., W.T.K., N.C.K., S.K., F.L., K.D.L., A.N., D.J.O., S.M.P., S.P., E.M.R., S.S., P. Seenivasan, E.T.T., L.U., A.Z. and T.Z. implemented software. K.J.H., Z.L. and P.K.R. prepared the sample. A.N. and G.M.R. produced GAL4 driver lines. H.F.H., W.Q. and C.S.X. imaged the sample. C.O., T. Pietzsch, S.P., S.S., L.K.S. and E.T.T. aligned the sample. S.B., M.J., C.O. and S.-y.T. produced the automated segmentation protocol. G.B.H., C.O. and P.K.R. identified the synapses. S.B., J.F., N.C.K., A.N., M.B.R. and A.Z. identified neurotransmitters. J.B., A.N. and S.S. performed the LM–EM mapping. S.B., J.B., L.E.B., E.G., J.H., S.K., F.L., K.D.L., A.N., C.O., M.B.R., E.M.R., P. Seenivasan, S.-y.T. and A.Z. defined the neuropil compartments. S.B., B.S.C., M.C., S.F.-M., M.A.F., G.P.H., C.K., J.K., S.A.L., M.L., A.L., C.A.M., C.M., A.N., N.O., C.O., T. Paterson, E.M.P., J.R.S., A.L.S., L.A.S., S.-y.T., S.T., I.T., A.T., J.J.W., C.W., E.A.Y. and T.Y. proofread the segmentation. S.B., L.E.B., M.D., E.G., J.H., N.C.K., S.K., F.L., K.D.L., A.N., M.B.R., E.M.R., L.K.S., P. Seenivasan and A.Z. analysed the connectome. S.B., J.B., L.E.B., M.D., E.G., J.H., P.M.H., S.K., F.L., K.D.L., A.N., M.B.R., E.M.R., S.S., P. Seenivasan and A.Z. developed visualizations. S.B., L.E.B., E.G., J.H., G.S.X.E.J., N.C.K., C.K., S.K., F.L., K.D.L., A.N., C.O., S.M.P., M.B.R., P.K.R., E.M.R., S.S., P. Seenivasan, S.-y.T., E.T.T., L.U. and A.Z. performed quality checks on the connectome. J.H., A.N., M.B.R. and S.-y.T. determined the cell types, and A.M.C.F., P. Schlegel, A.N. and G.S.X.E.J. matched cell types to FlyWire. S.B., J.F., R.G., H.F.H., G.S.X.E.J., C.K., W.K., F.L., A.N., S.M.P., S.P., M.B.R., S.R., G.M.R. and S.S. supervised internal efforts.

**Competing interests** The authors declare no competing interests.

**Additional information**
**Correspondence and requests for materials** should be addressed to Stuart Berg, Gerald M. Rubin or Michael B. Reiser.

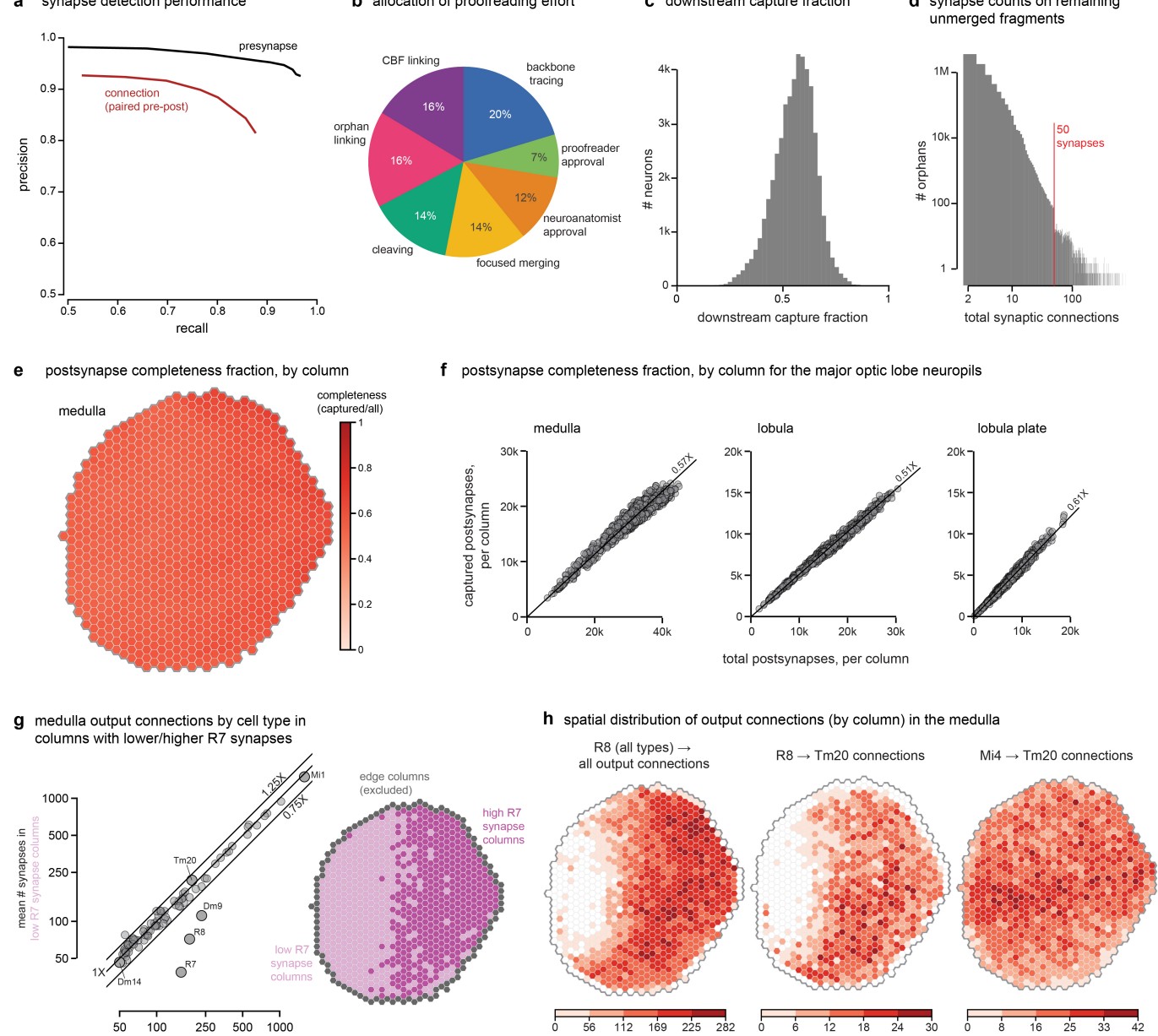

**Extended Data Fig. 1 | Proofreading summary and quality control for the optic lobe connectome.** (a) Performance of synapse detection. The precision/recall curve for presynapses (black), and connections (pre-post pairs, red). Precision is the probability that a detected feature was human-annotated, and recall is the percentage of features found by human-annotators that were also identified by our trained network (described in Methods: EM volume synapse identification). (b) The estimated proportion of time spent in the major phases of proofreading, totaling 2584 proofreader/neuroanatomist days expended on this dataset, approximately ten proofreader-years. (c) The distribution of downstream capture fraction for the neurons in the optic lobe connectome, excluding the lamina and those with <30 presynapses. (d) The distribution of synapse counts for all 9.6 M remaining unmerged fragments ('orphans') in the optic lobe, excluding the lamina. (e) The completeness fraction for the postsynapses in medulla, by column is approximately uniform, showing no spatial gradients. (f) The postsynapse completeness fraction is shown as a scatter plot for all three major optic lobe neuropils, where each point represents

the data for one column ROI (see Fig. 3). The superimposed line plots the median fraction across all columns, which are nearly identical to the completeness fraction for each neuropil (Extended Data Table 1). (g) The mean synapse counts, by cell type, for neurons belonging to cell types with downstream connections in most medulla columns (at least 30 output connections in each of at least 600 medulla columns), sorted into two groups based on R7 output connection count in each column. The map on the right shows the positions of the medulla columns making up these two groups. This summary shows that the data limitations of the R7 and R8 photoreceptors (see Methods and Extended Data Fig. 5) do not extend to other cell types, with the exception of Dm9, a cell type with extensive physical contact with R7/R8. (h) Spatial distribution of selected connections to illustrate that the missing photoreceptor connections, R8 in this example, do not extend to other cell types with connections in similar columns and layers of the medulla. Connections of all R7 and R8 types were combined for analysis in g,h.

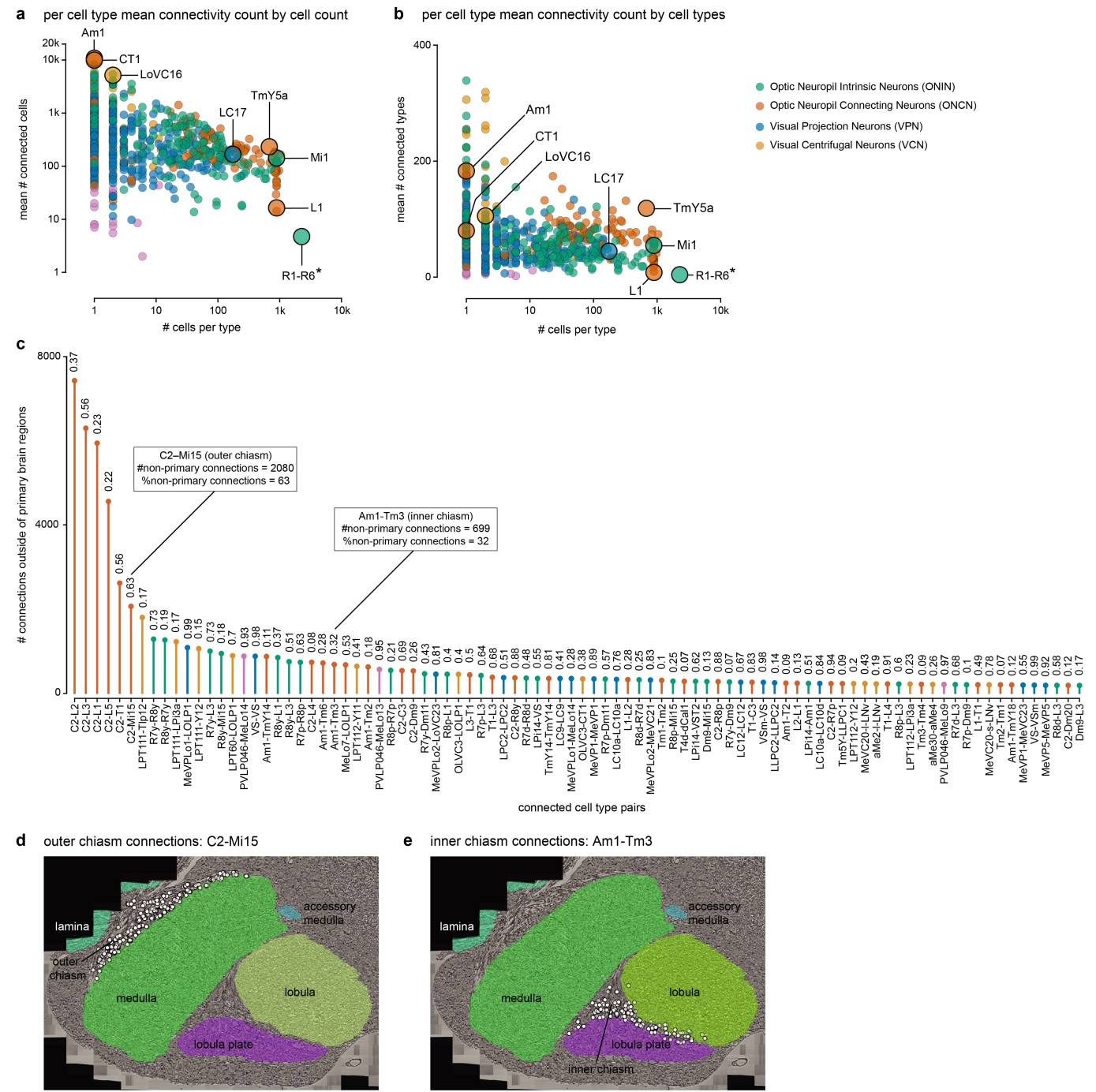

**Extended Data Fig. 2 | Connectivity Summary for the visual cell types.**
Related to Fig. 1. (a) The mean number of connected cells (in the optic lobe, counting all connections > 1) as a function of the number of cells per type for all cell types in the inventory. Several examples are highlighted, as in Fig. 1g. R1-6 are indicated with an asterisk since they are undercounted in the data set (see Methods). (b) Mean number of connected cell types (in the optic lobe, counting all connections > 1) as a function of number of cells per type. The same cells are highlighted as in (a). (c) The cell type pairs with the largest contributions to connections outside of the main optic lobe neuropils. The plot shows the number of synaptic connections outside the primary optic lobe neuropils and the fraction they contribute to all connections between the indicated cell type pairs. The connections are directional, and the pairs are ordered as A-B, where A is presynaptic to B. Many of these connections are in the outer or inner chiasm. Only pairs with > 5% of their connections outside of the neuropil boundaries are included. Additional details are in Supplementary Table 2. (d) Example of prominent connectivity in the outer chiasm: ≈60% of C2-Mi15 connections are within the outer chiasm, highlighted in (c). (e) ≈30% of Am1-Tm3 connections are within the inner chiasm, as highlighted in (c).

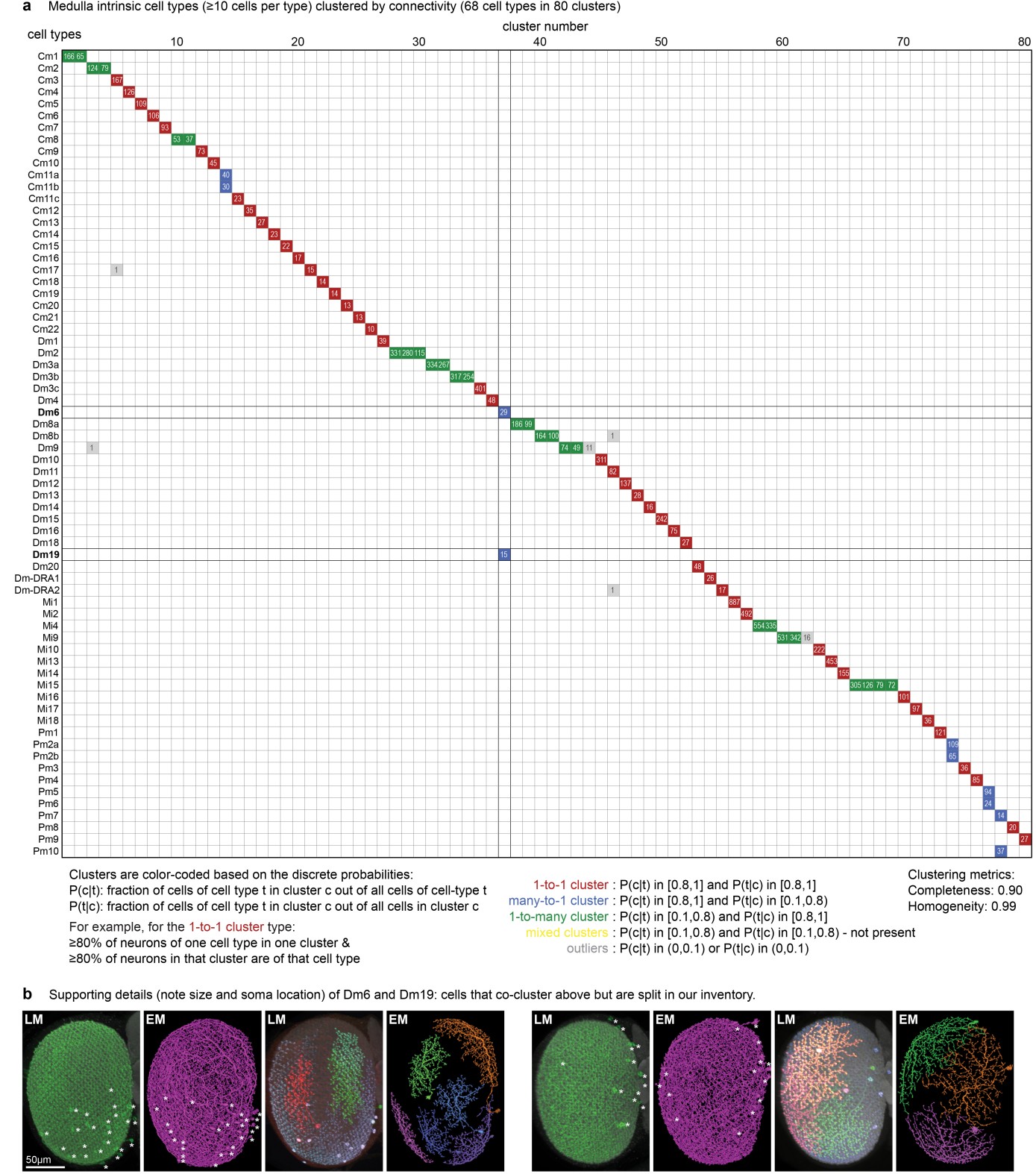

**a** Medulla intrinsic cell types (≥10 cells per type) clustered by connectivity (68 cell types in 80 clusters)

Clusters are color-coded based on the discrete probabilities:
P(c|t): fraction of cells of cell type t in cluster c out of all cells of cell-type t
P(t|c): fraction of cells of cell type t in cluster c out of all cells in cluster c

For example, for the 1-to-1 cluster type:
≥80% of neurons of one cell type in one cluster &
≥80% of neurons in that cluster are of that cell type

1-to-1 cluster : P(c|t) in [0.8,1] and P(t|c) in [0.8,1]
many-to-1 cluster : P(c|t) in [0.8,1] and P(t|c) in [0.1,0.8)
1-to-many cluster : P(c|t) in [0.1,0.8) and P(t|c) in [0.8,1]
mixed clusters : P(c|t) in [0.1,0.8) and P(t|c) in [0.1,0.8) - not present
outliers : P(c|t) in (0,0.1) or P(t|c) in (0,0.1)

Clustering metrics:
Completeness: 0.90
Homogeneity: 0.99

**b** Supporting details (note size and soma location) of Dm6 and Dm19: cells that co-cluster above but are split in our inventory.

Dm6 population Dm6 individual cells Dm19 population Dm19 individual cells

**Extended Data Fig. 3 | Sorting neurons with connectivity.** Related to Fig. 2. (a) Clustering based on connectivity for all medulla intrinsic cells with at least 10 cells per type. The 68 cell types are split into a preselected number of 80 clusters. Each cluster indicates the number of individual cells assigned to it. Cell types (Y-axis) are listed alphabetically, and clusters (X-axis) are further ordered by the number of cells per cluster. This sorting produces clusters with different cell type compositions (color-coded as indicated in the figure). We note that in all cases, 1-to-many clusters (green) for a given cell type are in a shared subtree of the dendrogram that does not contain other clusters.

Compare to morphology-only clustering for the same set of neurons in Extended Data Fig. 8. (b) EM to LM comparison of cell types Dm6 and Dm19. The LM images used split-GAL4 driver lines combined with population or stochastic labeling, see Methods. These cell types co-cluster in (a), but can be split by anatomy. For example, the cells have noticeably different sizes and distribution of cell bodies. Both features are visible in the EM and LM data (asterisks mark cell body locations). The two types can also be cleanly separated by further connectivity clustering (not shown), and the existence of a selective split-GAL4 driver for each indicates that they are genetically distinct.

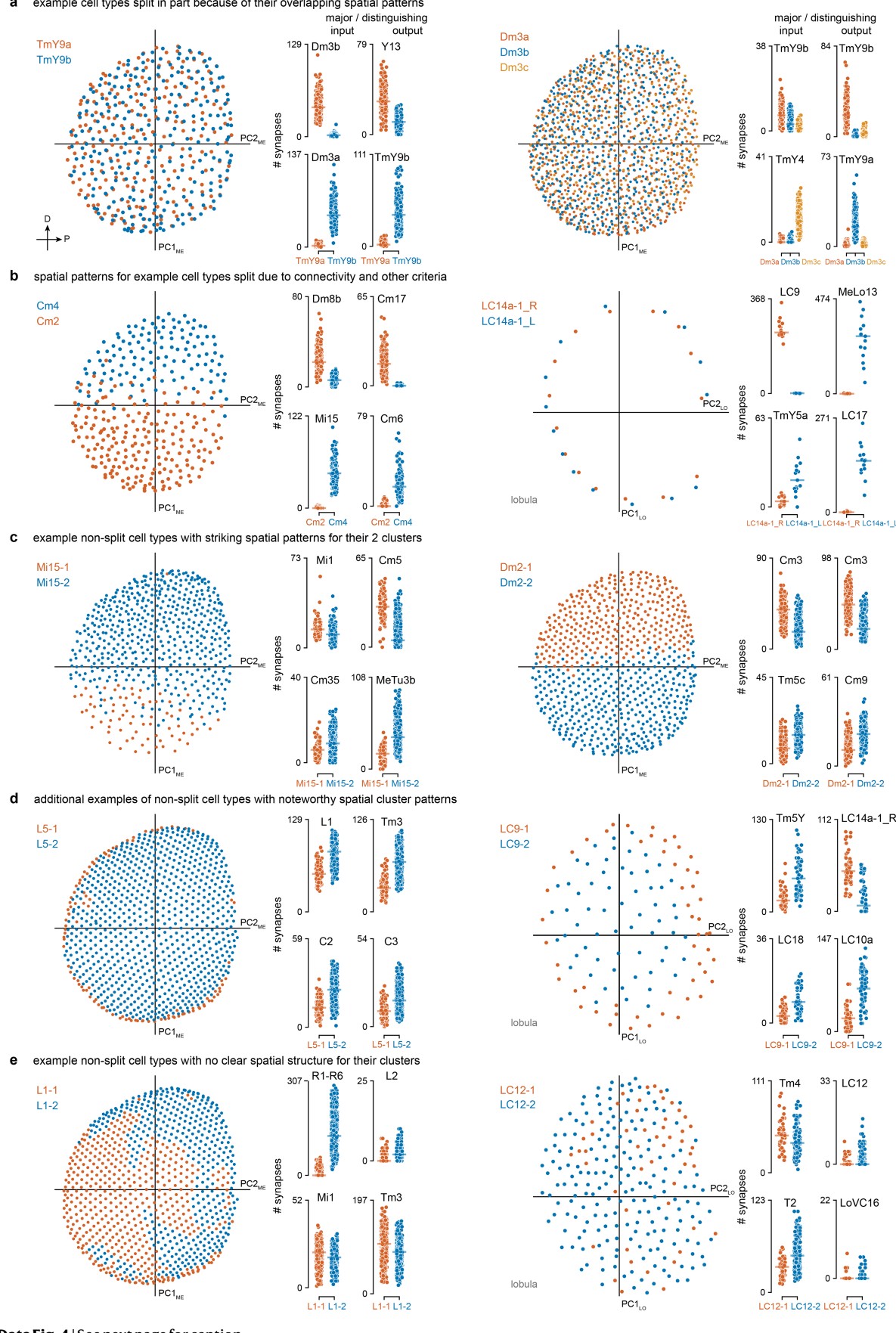

**Extended Data Fig. 4** | See next page for caption.

**Extended Data Fig. 4 | Combining connectivity clustering with spatial distributions to evaluate cell type merges/splits.** Related to Fig. 2. 10 groups of cell types evaluated for potential splitting into separate cell types; presentation follows the conventions of Fig. 2c,g,h, where selected distinguishing input and output connections are shown. On the right side of each panel, each point represents the sum of connections to/from the indicated cell type for a single neuron from each group, horizontal line indicates the median value. This set of examples covers the various cases we find across the dataset. Cells that are split into distinct types are indicated by their assigned names, but examples of non-split cells are indicated as −1 and −2 for the clusters of cells with the same type designation. (a) Examples of groups of similar cell types that show connectivity differences and overlapping spatial distributions (mosaics) and were split into different types. Splitting TmY9a/TmY9b and Dm3a/b/c is also supported by their different arbor orientations (Fig. 5). (b) The Cm2 and Cm4 cell types show overlapping distributions, consistent connectivity differences, and different neurotransmitter predictions. Left and right instances of cells with arbors in both optic lobes (here LC14a-1) often have distinct connectivity within the same optic lobe. Such neurons were treated as two types (e.g., LC14a−1_L and LC14a-1_R) in some analyses. (c) Examples of cell types with subclusters with distinct spatial distributions. Mi15 and Dm2 subclusters occupy different domains along the DV axis; note that the overall density of Mi15 cells also differs along this axis. Such divisions suggest regional differences within these cell populations, but the absence of strong, consistent connectivity differences prevented division into distinct types. (d) Other examples of cell types that were not split owing to the lack of strong connectivity differences between the subclusters that show striking spatial distribution patterns: L5 (cells at margin separate) and LC9 (center vs. perimeter). (e) Additional examples of cell types that were not split and show no obvious structure in the spatial distributions of subclusters: L1 and LC12.

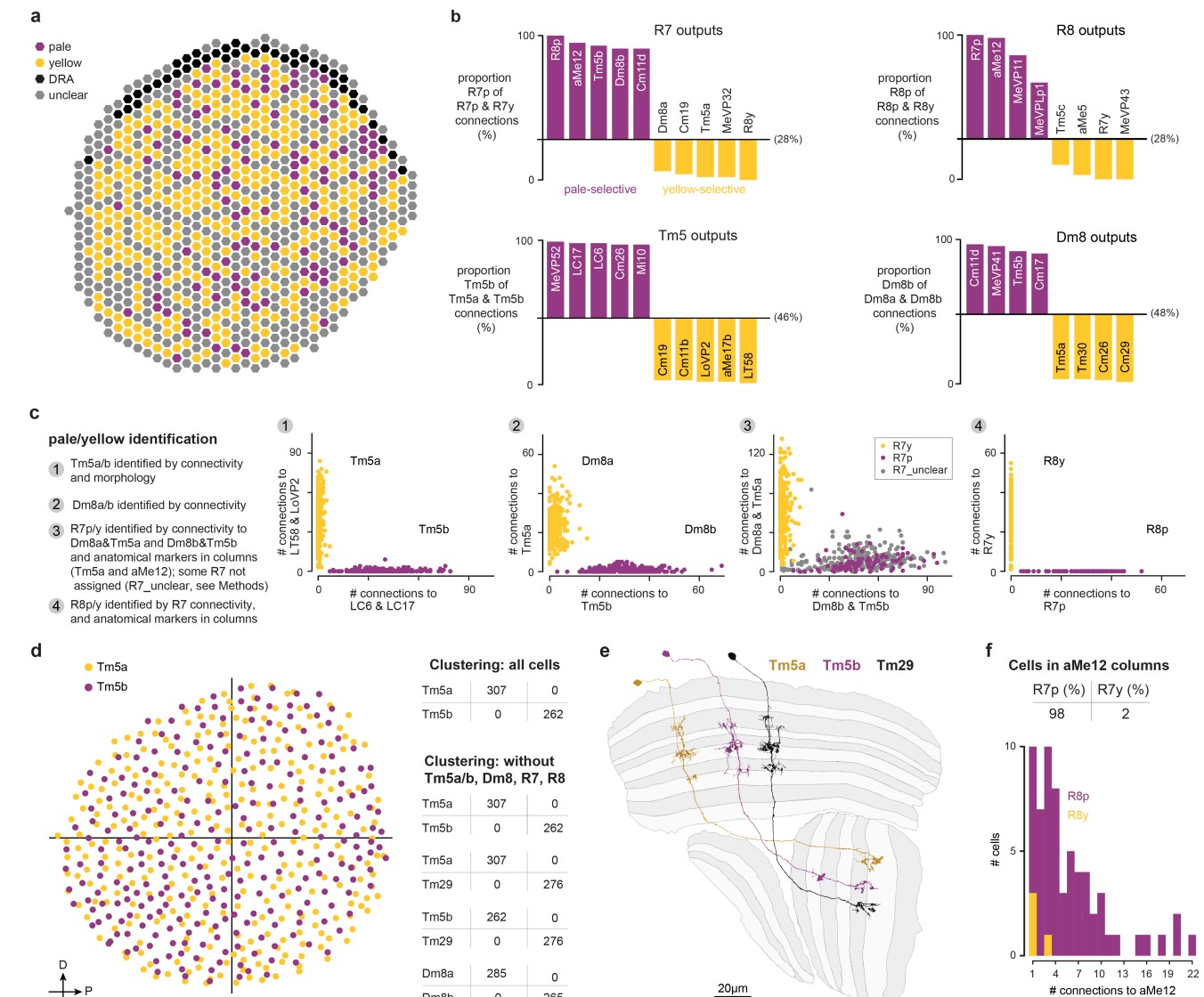

**Extended Data Fig. 5 | Functionally specialized inner photoreceptor types supply the visual system with color and polarized light information.** Related to Fig. 2. (a) Summary eye map of pale and yellow columns that supply color information and dorsal rim area (DRA) columns that supply polarized light information. Eye map uses the coordinate system based on medulla columns introduced in Fig. 3a. For details of column assignments, including columns with missing R7 or R8 photoreceptors (Extended Data Fig. 1g,h) see Methods: Assigning R7 and R8 photoreceptors and medulla columns to different types of ommatidia). (b) Plots of pale and yellow pathway connections. For each synaptic connection pair, bars indicate the fraction of pale-pathway connections, e.g. (# R7p to Tm5b connections) / (# R7 to Tm5b connections). The baseline is the fraction of connections expected from the fraction of pale-pathway cells of that type, e.g., # R7p cells / # R7 cells. (c) Overview of the pale/yellow identification process. Step 1: Identification of Tm5a and Tm5b cells based on connectivity clustering and cell morphology; summed connectivity with a subset of lobula cell types (LC6 & LC17, LT58 & LoVP2) is shown for individual Tm5a/b cells. Step 2: Identification of Dm8a and Dm8b with distinct connectivity to Tm5a, Tm5b,

and other cell types by connectivity clustering; connectivity with Tm5a and Tm5b shown. Step 3: R7 cells are classified as pale (R7p) or yellow (R7y) using connectivity with Dm8a/b and Tm5a/b and anatomical markers; cells that could not be confidently assigned are labelled as 'R7_unclear'. Step 4: R8 cells are classified as R8p or R8y based on connectivity with R7p/y or anatomical markers (in columns with missing R7 cells). (d) Left: Spatial map of Tm5a/b, separated as in Fig. 2, Extended Data Fig. 4. Right: Results of connectivity clustering Tm5a/b cells using all connections (top) and clustering each indicated pair of cells when connections with Tm5a/b, Dm8a/b, R7, and R8 are excluded (below). (e) Examples of Tm5a, Tm5b and Tm29 morphology. Many Tm5a cells have hook-shaped terminals in the lobula (yellow arrow) and slightly narrower medulla processes than Tm5b or Tm29 cell types; many Tm5b cells have a small process in the lobula layer 2 (purple arrow) missing in Tm5a and Tm29. (f) Colocalization (top) and connectivity (bottom) of inner photoreceptors with aMe12, a cell type previously identified as pale-specific[23]. Top: percentage of R7p and R7y in manually identified columns innervated by aMe12. Bottom: distribution of aMe12 synapses from R8p/y cells.

**a**

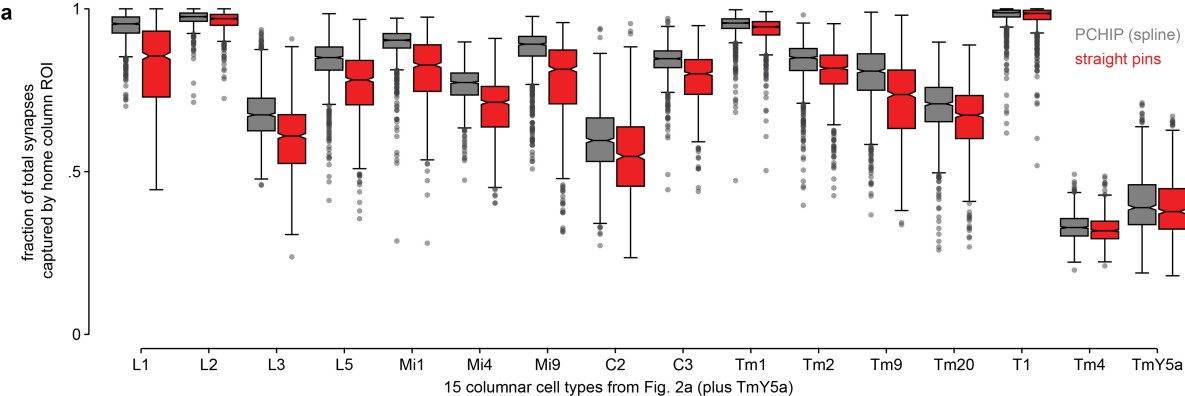

**b**  mapping sets of T4s to all Mi1-defined columns, extending medulla map to Lobula Plate

**c**  example Mi1 and its set of 4 assigned T4s

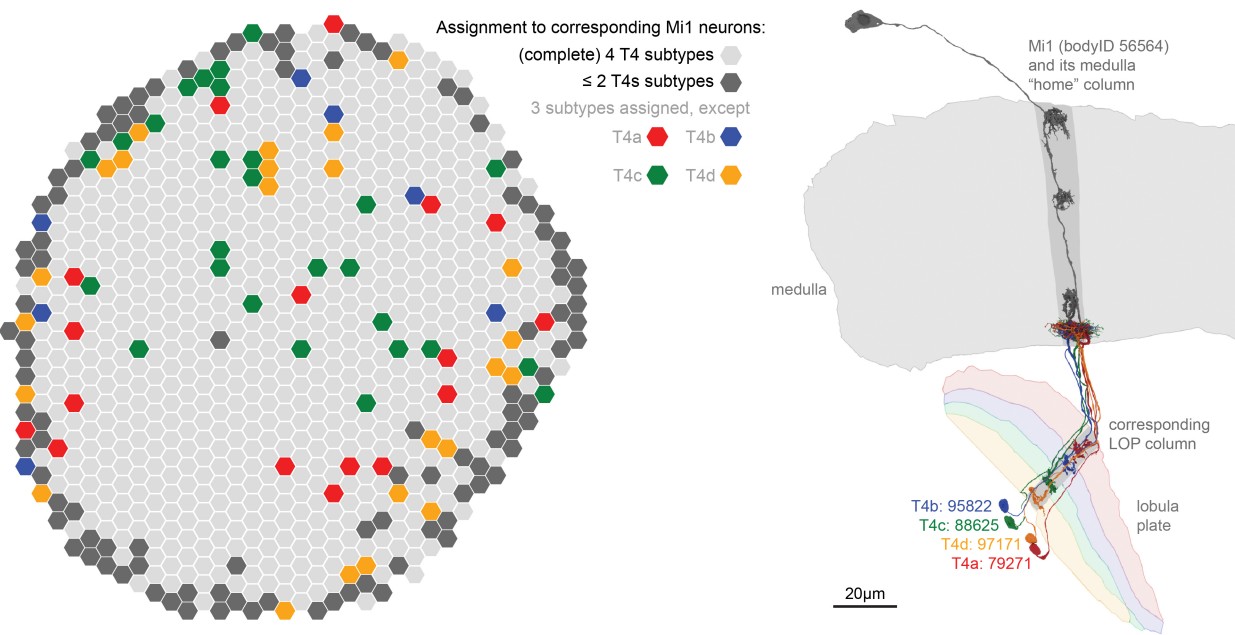

Assignment to corresponding Mi1 neurons:

(complete) 4 T4 subtypes ⬡

≤ 2 T4s subtypes ⬡

3 subtypes assigned, except

T4a ⬡   T4b ⬡

T4c ⬡   T4d ⬡

**d**  representative TmY5a neurons and their predominant columns in the ME, LO, and LOP (see also Fig. 1c)

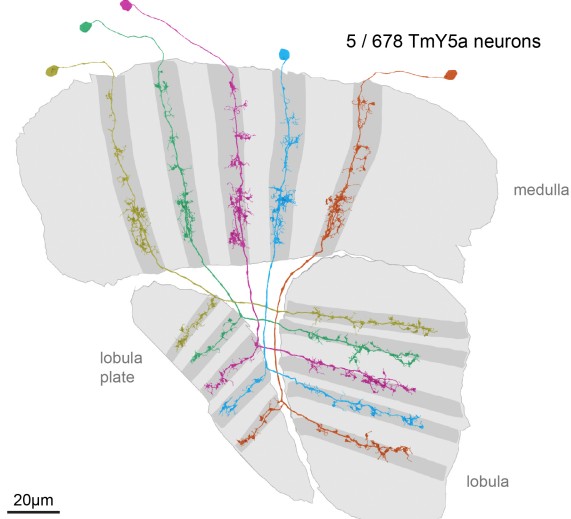

5 / 678 TmY5a neurons

**Extended Data Fig. 6** | See next page for caption.

**Extended Data Fig. 6 | Supporting data for the creation of columns and their extension to the lobula plate.** Related to Fig. 3. (a) Comparison of straight lines to spline-based method for defining medulla column centerlines. Plots shows the fraction of synapses in the home column (the column with the largest synapse count) of each neuron of a cell type. For each cell type used to construct the medulla columns (first 14), the fraction is higher for splines than for straight lines. For the other 2 cell types, the fractions are comparable. Boxes show first, second, and third quartiles; whiskers are drawn at 1.5 times the interquartile range below and above the first and third quartiles, respectively. (b) A lobula plate coordinate system was derived from the medulla coordinates by mapping neurons of the four T4 types to individual Mi1s. In most hexagons (712, in light gray), all 4 types could be matched to a single Mi1, but in 75, only 3 types could be matched (the missing neuron is indicated in color). In the 105 dark hexagons near the perimeter, only 2 or fewer T4s could be matched to an Mi1. Matching criteria include connectivity and distance (see Methods). (c) An example Mi1 and the set of 4 assigned T4 neurons. (d) TmY5a neurons are the most numerous TmY cells and are used to visualize the column assignments across neuropils. This example shows a few TmY5a cells with the corresponding home columns in the medulla, along with the lobula and lobula plate columns with the same hexagonal coordinate.

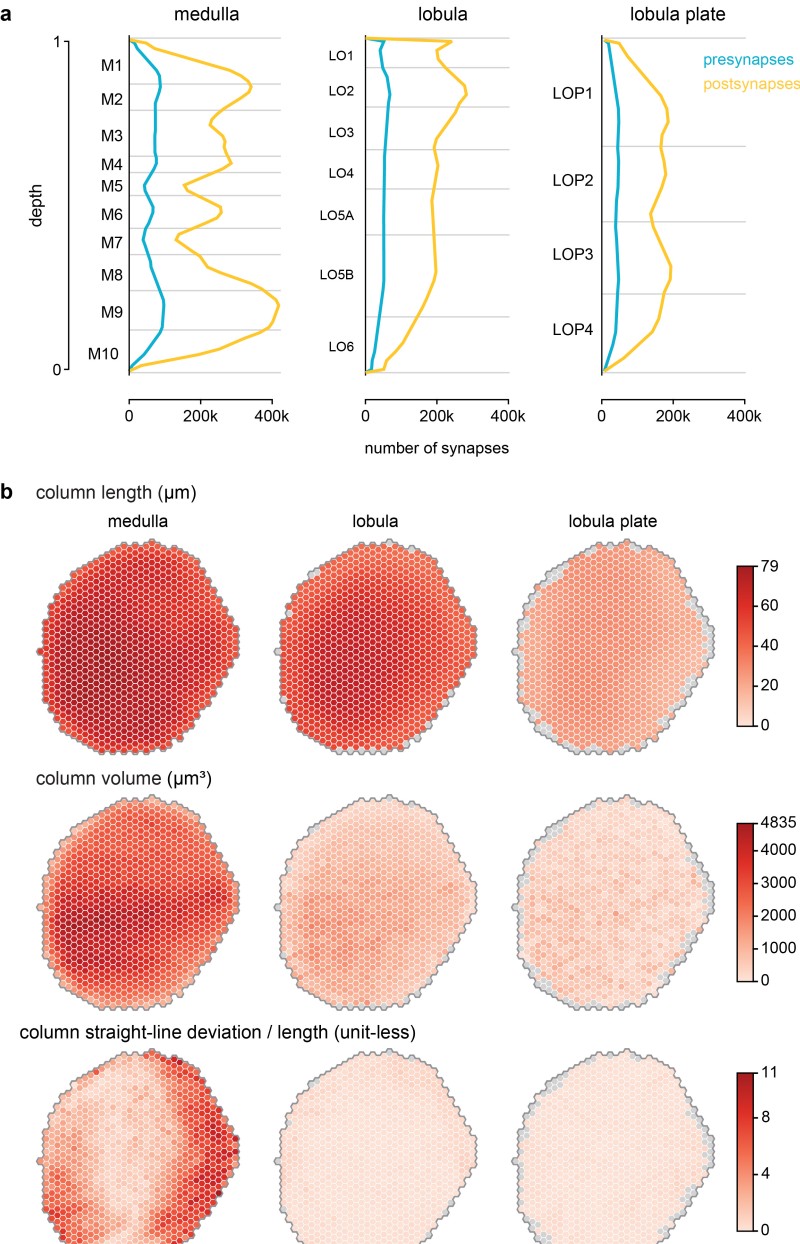

**Extended Data Fig. 7 | Summary of synapse distributions by columns and layers.** Related to Fig. 3. (a) The spatial distribution of presynapses (blue) and postsynapses (yellow) as a function of depth in the 3 main optic lobe neuropils. The ratio of pre-to-post is approximately 1:6 and this is conserved across most of the layers of the visual system. (b) Quantification of the dimensions of the columns in the main optic lobe neuropils. As expected, columns in the medulla are longest, the lobula columns somewhat shorter, and the lobula plate columns much shorter. Column volume is shown in the middle row. There is a noteworthy increase in volume near the equator of the eye and towards the front (left) that is especially prominent in the medulla. The bottom row shows the straightness of columns normalized by column length. Medulla columns, especially near the anterior and posterior margin are found to have the greatest curvature. While this reflects the nature of that neuropil, it is also in part due to the richer set of neurons used to define medulla columns. Gray columns around the periphery indicate columns that are not present in that brain neuropil.

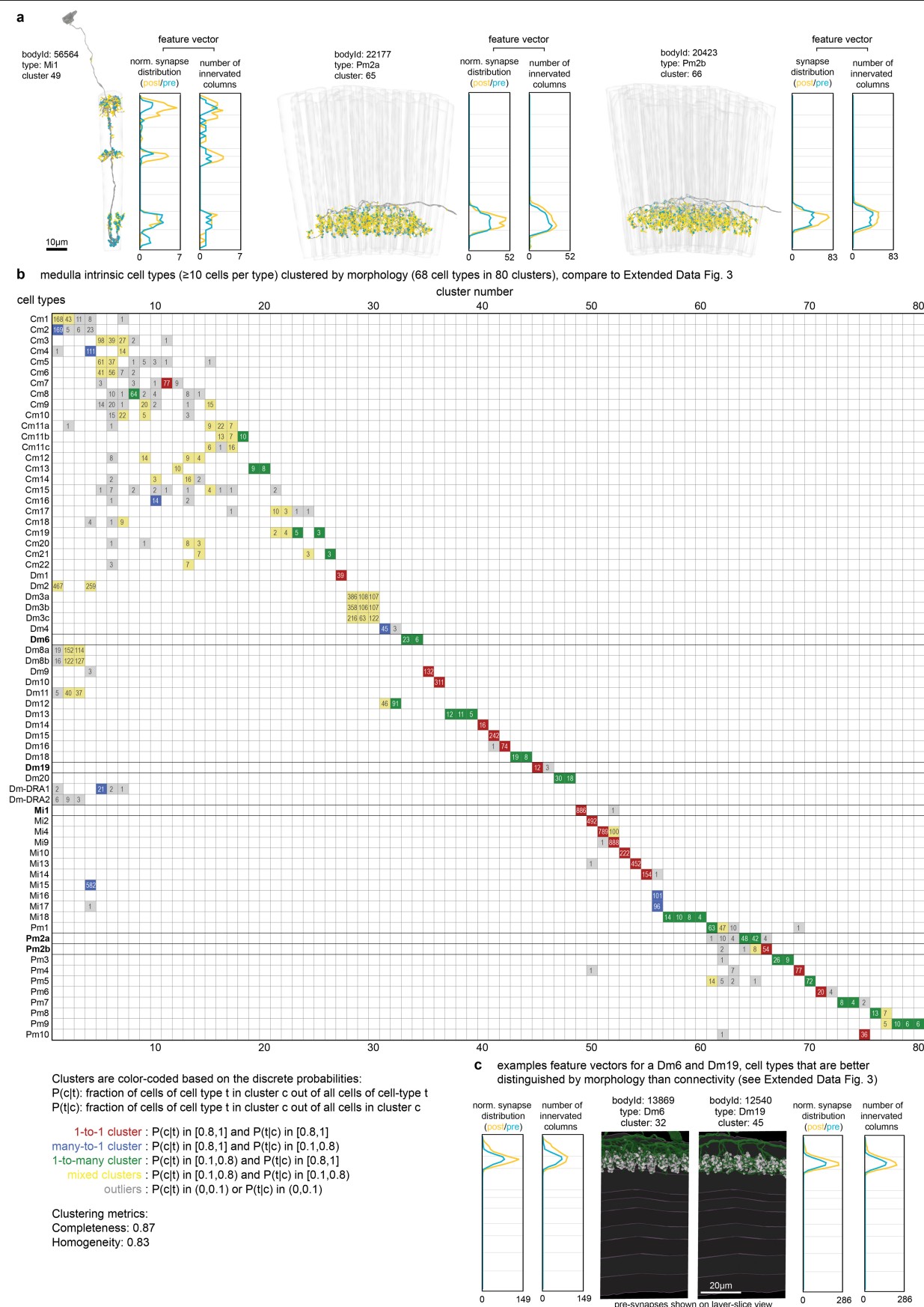

**Extended Data Fig. 8** | See next page for caption.

**Extended Data Fig. 8 | Clustering neurons using quantified anatomy.**
Related to Fig. 5. (a) Examples of 3 neurons, each of a different type, to illustrate the aspects of quantified morphology used to construct a feature vector for each cell. Each neuron is shown together with its primary columns and the layer distribution of its presynapses and postsynapses. The feature vector comprises scaled synapse distributions and the number of innervated columns per depth, separately for the presynapses and postsynapses (see Methods). The pre/post synapse distribution is normalized such that the sum equals the sum of pre/post number of innervated columns for that cell. It is noteworthy that Pm2a/b neurons can be more easily distinguished by the total number of innervated columns than by their volume (see Fig. 2f). (b) Confusion matrix of clustering the same 68 medulla intrinsic cell types as in Extended Data Fig. 3 where clustering was based on connectivity, but here based on the quantified morphological feature vectors. The data are sorted and color-coded as in Extended Data Fig. 3 to facilitate comparison. (c) As an example of morphologically similar neurons from different cell types, we highlight the feature vectors for one Dm6 and one Dm19, representing these cell types that are more separable by their quantified morphology (see (b)) than by their connectivity (Extended Data Fig. 3). The feature vectors show a noticeable difference in the innervation of layer M1, which we confirmed by visualizing the presynapses of both cells in a layer-slice view.

**Male *Drosophila* Visual System Connectome - Cell Type Explorer**

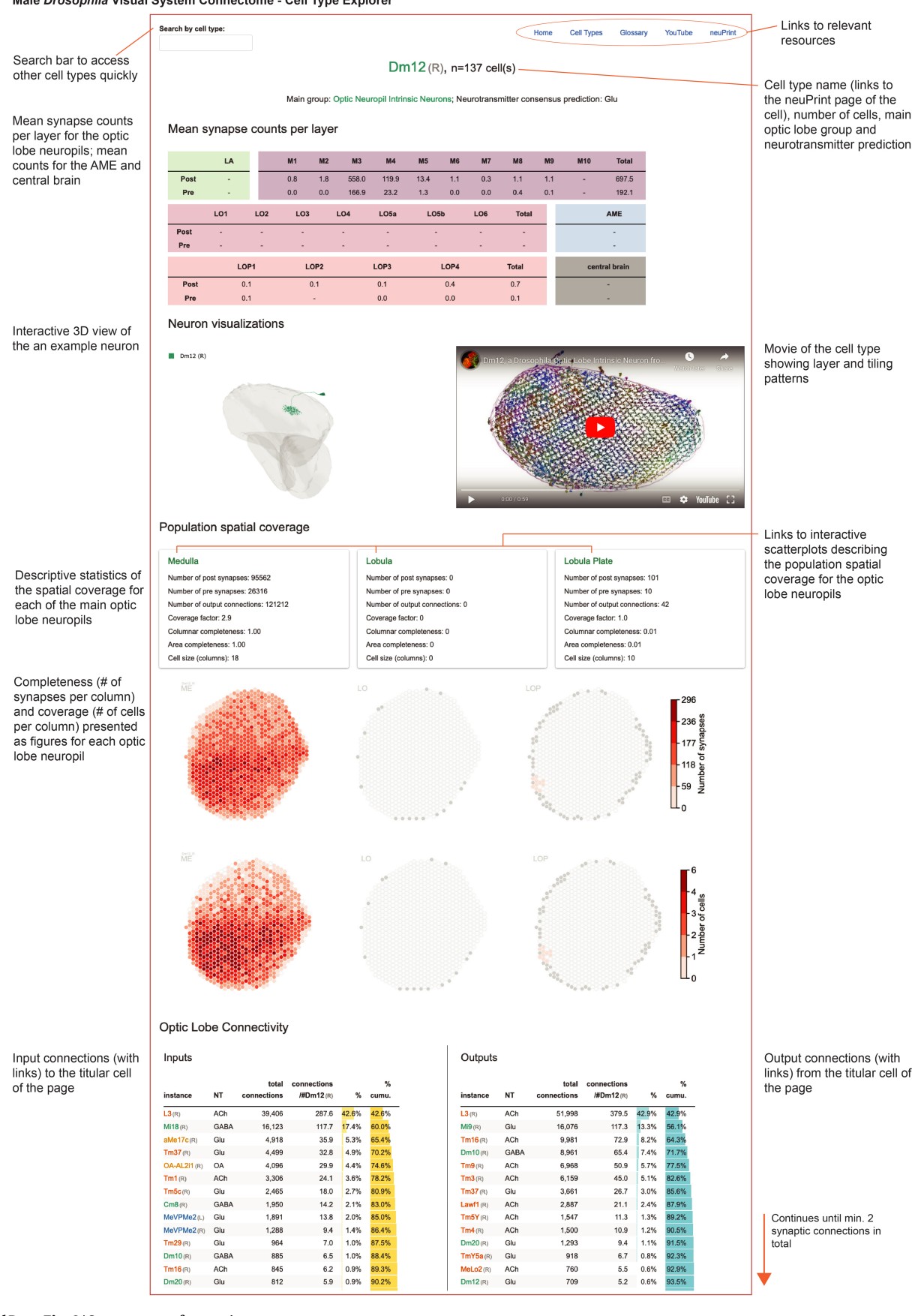

**Extended Data Fig. 9** | See next page for caption.

**Extended Data Fig. 9 | Interactive visual system Cell Type Explorer web resource.** Related to Fig. 5. The *Drosophila* Visual System Cell Type Explorer is a set of interactive webpages designed to offer more details about the retinotopy and connectivity for each cell type in the optic lobe. Each page features the name of the cell type at the top, which links to the cell type's neuPrint page, and includes information on its assigned neuron group and predicted neurotransmitter. Below this header are tables displaying mean presynapse and postsynapse counts across every optic lobe neuropil, with the main neuropils divided into layers. These tables include mean (by cell) synapse counts within the central brain as well. The next section presents a 3D interactive view of a representative cell alongside a linked video that illustrates the cell type's layer and tiling patterns, as well as the entire cell population. The 'Spatial coverage' section further describes the distribution of synapses and cell counts per column across the main optic lobe neuropils (introduced in Fig. 5d), including data on cell size and total synapse counts. The bottom of the page is devoted to the optic lobe connectivity table, arranged by the magnitude of total synaptic connections. This table color-codes each cell type according to its main group, listing on the left ('Inputs') those cell types that provide input to the profiled cell type, and on the right ('Outputs'), those cell types that receive input from it. The webpages are interconnected, allowing users to navigate between cell types via the Optic Lobe Connectivity table. Navigation is further facilitated by links at the page's top to the Home page, where users can search for cell types by name, an Index page listing all optic lobe cell types, and a Glossary page that clarifies the terminology used throughout the site. We also provide interactive versions of the scatter plots in Fig. 5e,f, and Extended Data Fig. 10, so users can discover cell types by their coverage properties.

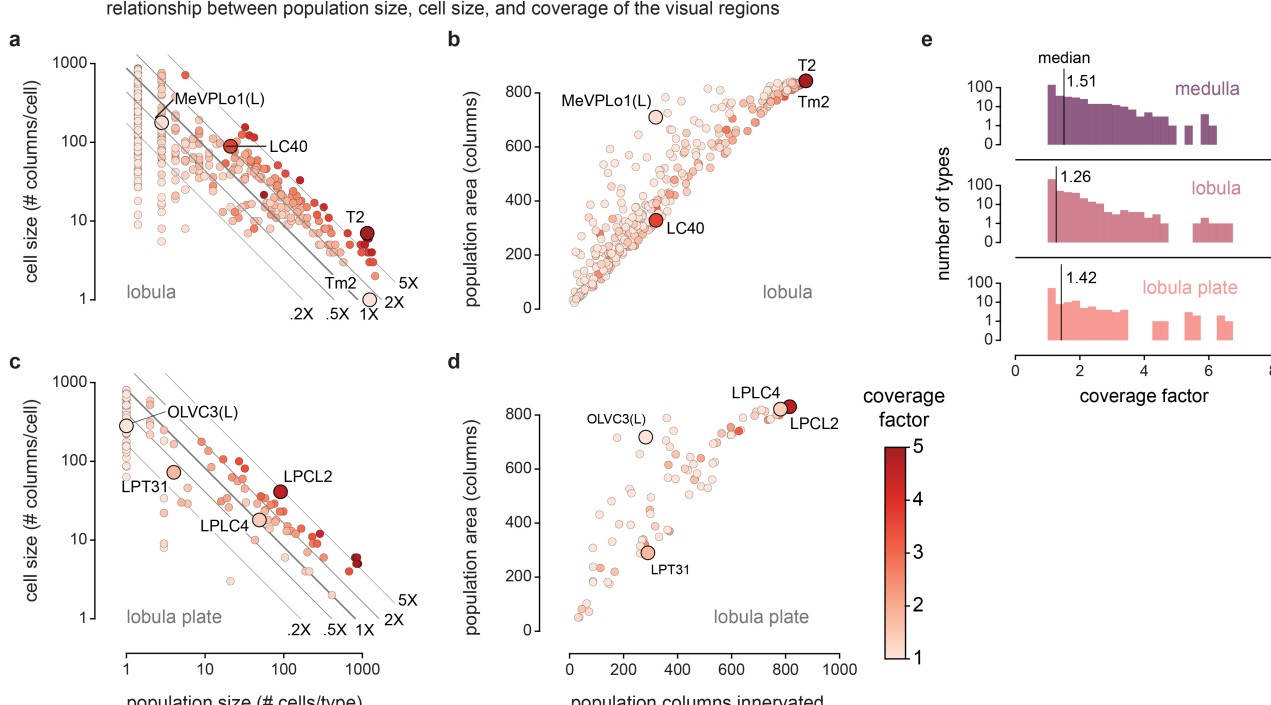

**Extended Data Fig. 10 | Coverage and density analysis for the cell types in the lobula and lobula plate.** Related to Fig. 5. (a) The relationship between population size (total number of cells) and cell size (the average number of columns innervated by a single cell) of cell types within the lobula, color-coded by coverage factor (the average number of cells that innervate a single column). The diagonal lines are guides to the ratio of global coverage: 1× for a given cell size, this is the 'optimal' population size to tile the columns of the neuropil, 2× and 5× more neurons and 0.5× and 0.2× fewer neurons than would be needed to fully cover the neuropil at this cell size. Selected types to highlight how the coverage factor reflects the tiling properties of the neurons are T2 (6.72), Tm2 (1.03). (b) Summary plots of the density with which cell types innervate the lobula, summarized by plotting the number of columns innervated against the convex area (in units of columns) covered by the total population. Only types with at least 5% of their total synapses and at least 50 synapses in total within the chosen brain neuropil were included in the plots. Selected types to highlight how the ratio between the columns innervated and area covered captures the density of innervation: MeVPLo1 (left instance, columns: 319, area: 710), LC40 (columns: 320, area: 329). (c) Same as (a) but for types assigned to the lobula plate. Selected types to highlight how the coverage factor reflects the tiling properties of the neurons are LPLC2 (4.63), LPLC4 (1.41). (d) As for (b) but for types assigned to the lobula plate. Selected types to highlight how the ratio between the columns innervated and area covered captures the density of innervation OLVC3 (left instance, columns: 283, area: 718), LPT31 (columns: 290, area: 290). (e) Distribution of coverage factor values in the three main optic lobe neuropils (top to bottom: medulla, lobula, lobula plate). Vertical black lines indicate the median coverage factor per neuropil.

**Dorsal Rim Area associated cell types in the medulla**

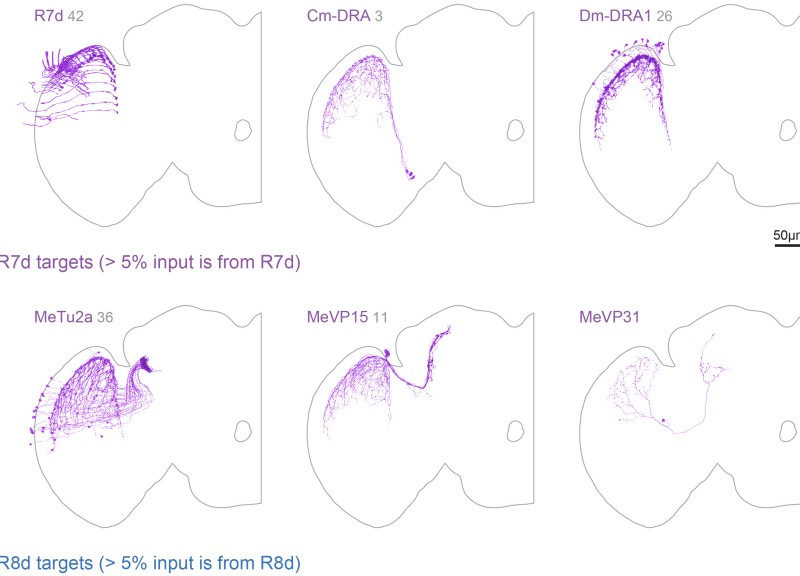

R7d 42 Cm-DRA 3 Dm-DRA1 26

50μm

**R7d targets (> 5% input is from R7d)**

MeTu2a 36 MeVP15 11 MeVP31

**R8d targets (> 5% input is from R8d)**

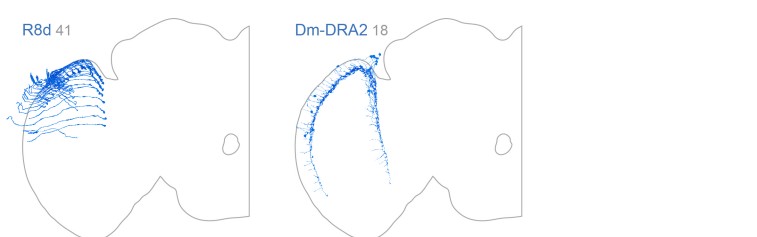

R8d 41 Dm-DRA2 18

Cell types that are spatially restricted to the dorsal medulla and receive >30%
of their input from the other types on this page

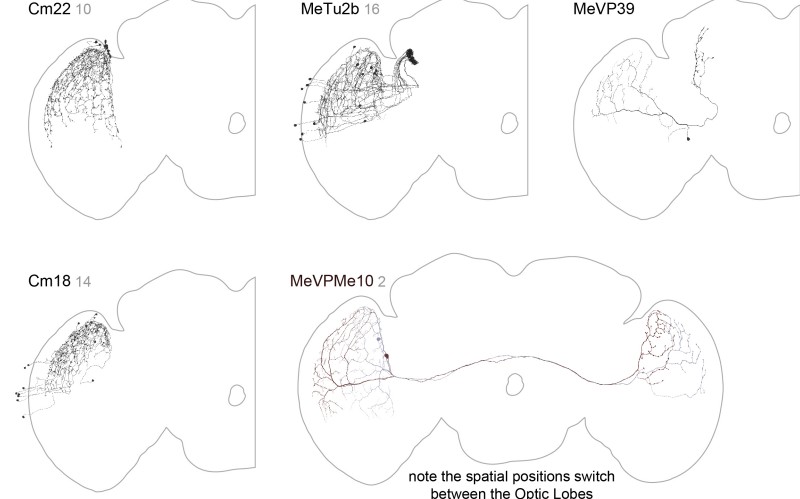

Cm22 10 MeTu2b 16 MeVP39

Cm18 14 MeVPMe10 2

note the spatial positions switch
between the Optic Lobes

**Extended Data Fig. 11 | Neurons associated with the Dorsal Rim Area.**
Related to Fig. 6. The Dorsal Rim Area (DRA) of the eye is a specialized zone of photoreceptors engaged in detecting polarized light. In the dorsal medulla, there are specialized cell types not found elsewhere in the medulla that process the output of the DRA photoreceptors. The primary cell types of the medulla area that correspond to the DRA of the eye are shown and organized into groups. The first two groups show R7d (magenta) and R8d (blue) photoreceptors and their main targets, which for R7d include both VPNs and medulla intrinsic cells.

The lower panels show additional cell types that are identified as other components of the medulla DRA network by their regional arborizations in the dorsal medulla and their connections with cells in the top panels and with each other. One of these cell types is a VPN that connects the DRA regions of the two optic lobes. Each panel shows all members of each cell type (for the right OL). The R7d and R8d photoreceptors are undercounted (see Extended Data Fig. 4 and Methods).

## Cell types with major innervation of the accessory medulla (AME)

HBeyelet plus targets; other cells with their main inputs/outputs in the AME

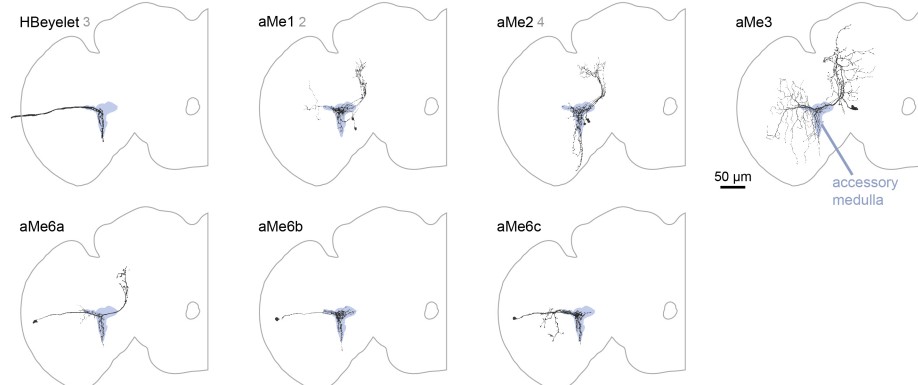

Cells with main Optic Lobe synapses in the AME and central synapses in PLP and/or dorsal brain;
many of these cells look similar and several are known clock neurons

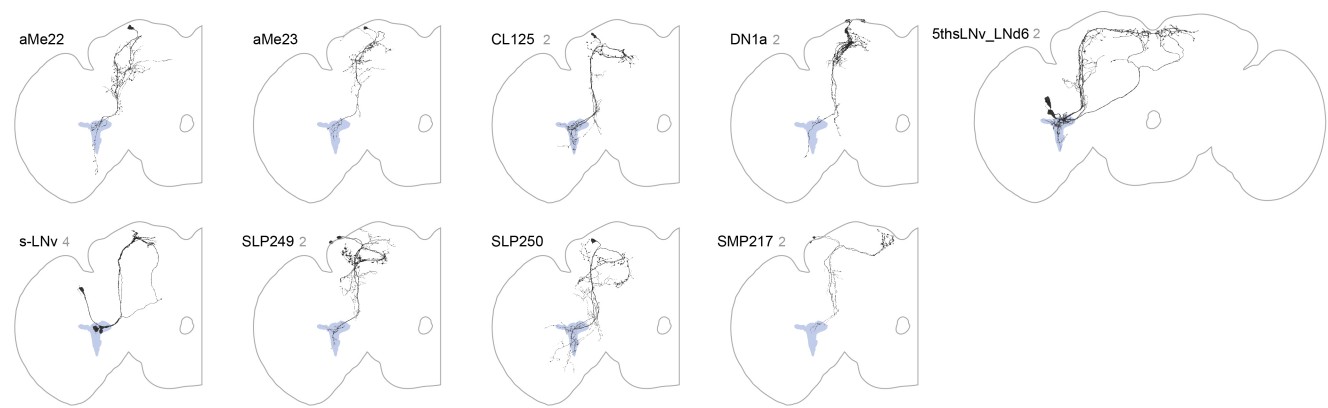

AME-associated Visual Projection Neurons: l-LNv (clock neurons) and aMe15 cells have major inputs in the AME;
other VPNs have major inputs/outputs in the AME but often large arbors in other optic lobe neuropils and the central brain

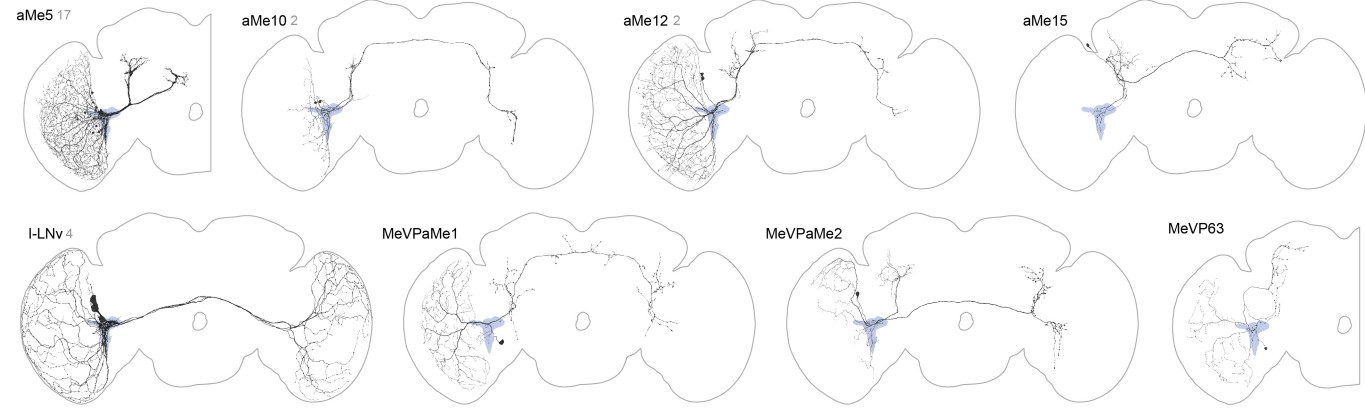

Visual Centrifugal Neurons with AME inputs/outputs

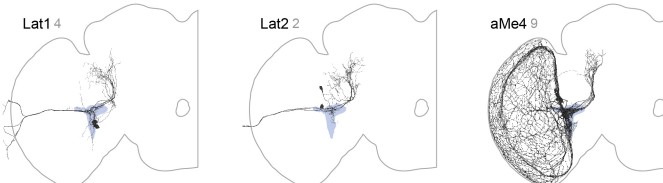

**Extended Data Fig. 12 | Selected neurons associated with the Accessory Medulla.** Related to Fig. 6. The Accessory Medulla (AME) is a small brain neuropil located at the anterior-medial edge of the medulla that is mainly known for its role in circadian regulation. It contains processes of several clock neuron types and a diverse group of VPN and VCN cells. This page does not include all AME-associated neurons but shows examples of cell types and cell type groups with processes in the aMe. For each cell type shown, all identified individual cells (for the right OL) are included.

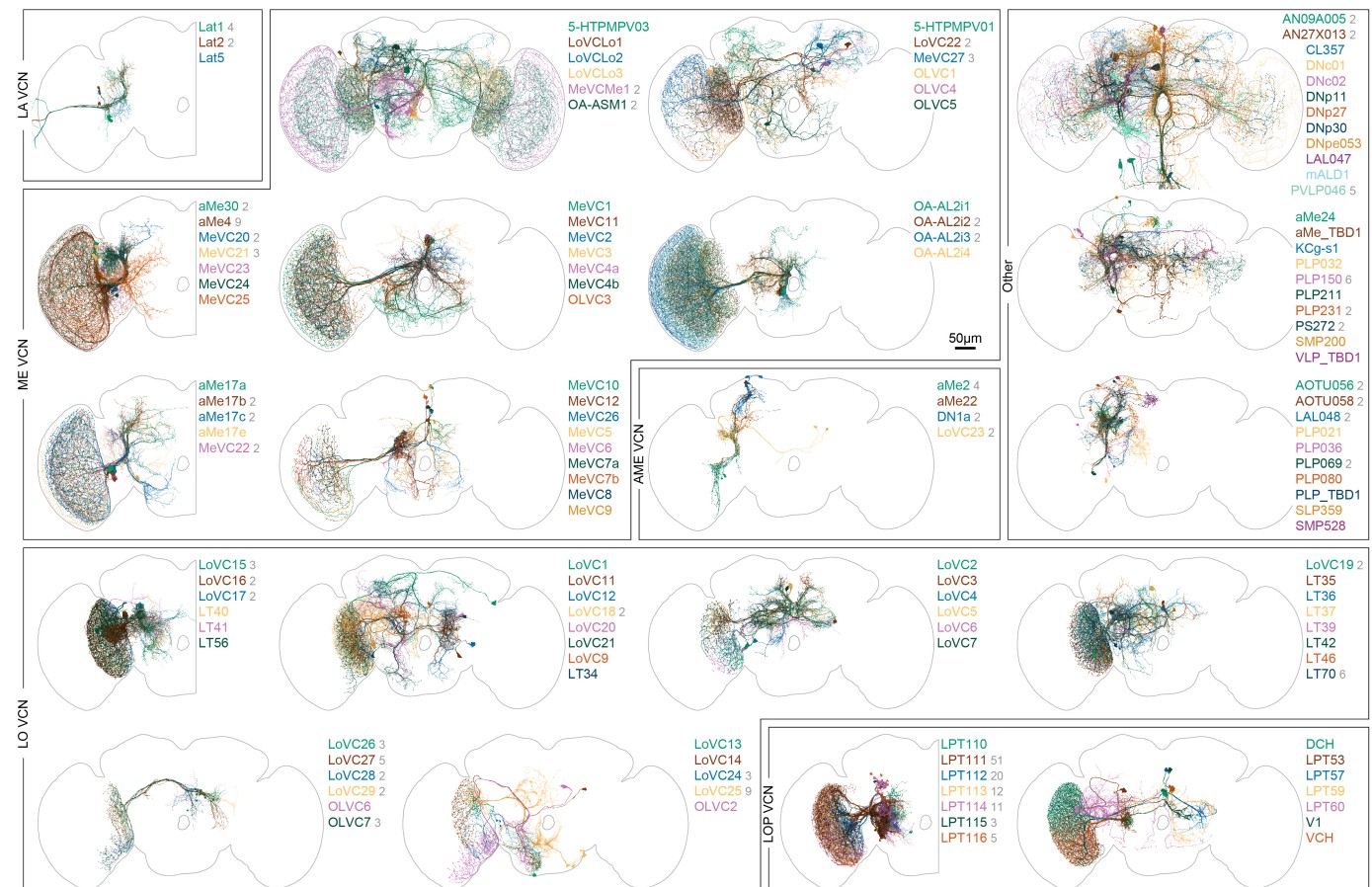

**Extended Data Fig. 13 | Visual Centrifugal Neurons.** Related to Fig. 6. Visual centrifugal neurons receive major input in the central brain and project back to the optic lobes. We cataloged 104 VCN types (270 cells combined for the right optic lobe) and show them all here in groups organized by their main target neuropils in the optic lobes and other anatomical features (such as ipsi-, contra- or bilateral projection patterns). The figure also includes neurons ('other') that have some optic lobe synapses in addition to central brain synapses but were not classified as VPN or VCN (see Methods). Each panel indicates the names of the cell types (color-matched to the rendered neurons) and the number of individual cells of each type (in gray); types without numbers are present once per brain hemisphere. Detailed morphology within the optic lobe neuropils can be found in the Cell Type Catalog (Supplementary Fig. 1).

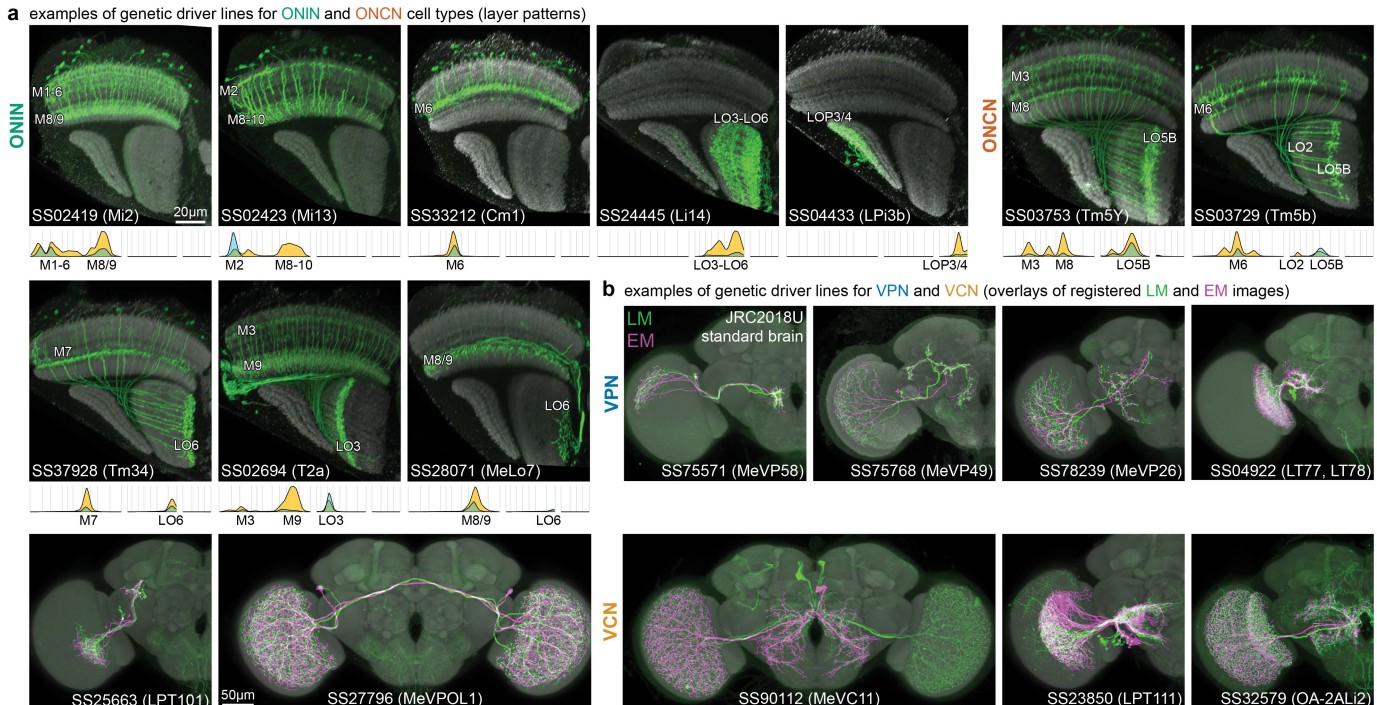

**Extended Data Fig. 14 | Additional examples of split-GAL4 lines matched to EM-defined cell types.** Related to Fig. 7. (a) Selected split-GAL4 lines driving expression in ONIN and ONCN types. Layer patterns of genetically labeled cell populations are shown with a neuropil marker (anti-Brp). Corresponding layers are indicated on both the LM images and the EM summary figures. (b) Selected split-GAL4 labeled VPN and VCN types. Images show overlays with registered EM skeletons. Detailed layer-specific patterns can be found in the Cell Type Catalog (Supplementary Fig. 1) and the list of split-GAL4 lines is in Supplementary Table 6. The EM/LM match shown for SS90112 is for MeVC11; this driver also has expression in MeVC2.

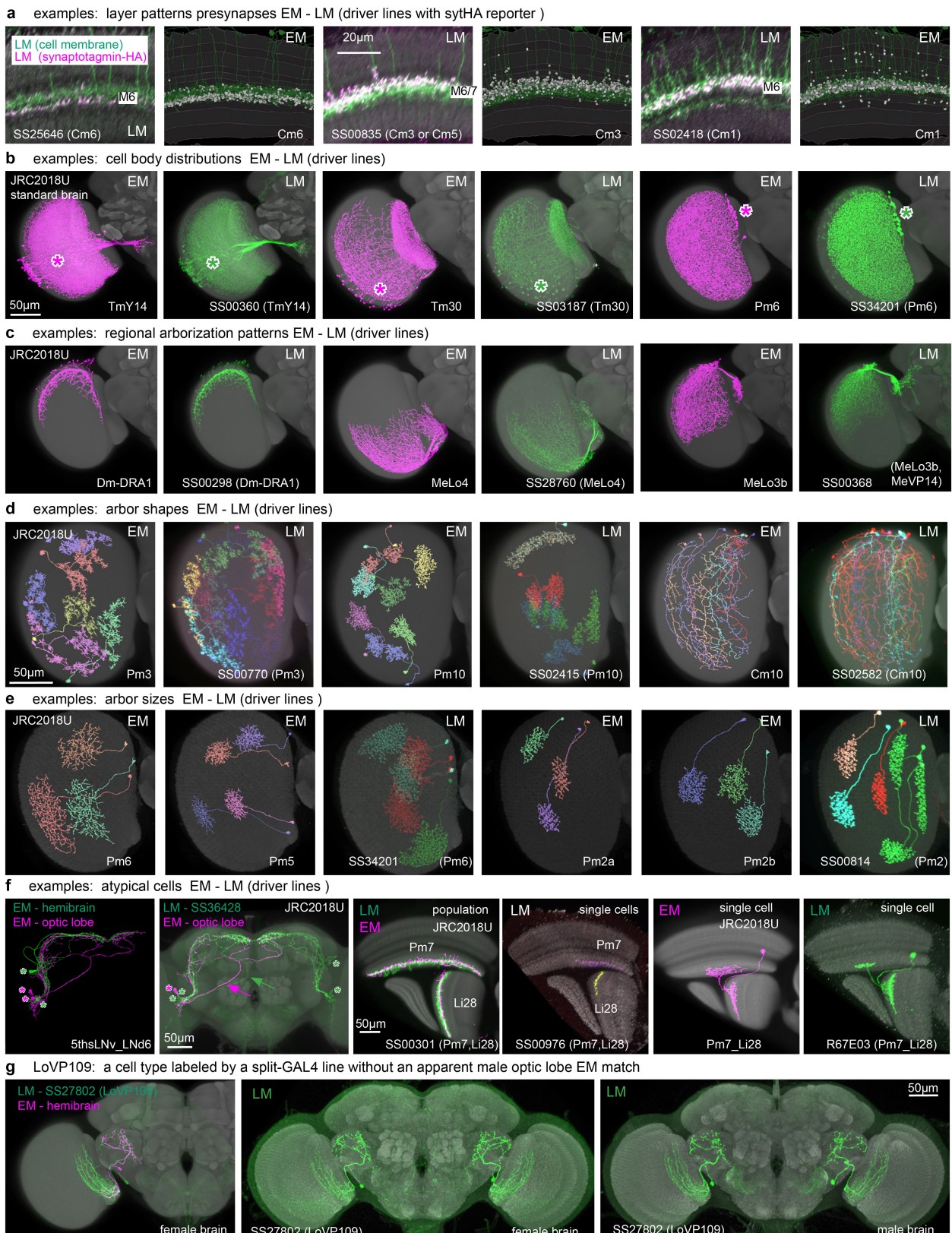

**Extended Data Fig. 15** | See next page for caption.

**Extended Data Fig. 15 | Supporting examples of anatomical features used for EM-LM matching.** Related to Fig. 7. Images in (b-e) and the left image in (g) show registered LM or EM images displayed on the JRC2018U template brain. LM images in (a-c) are based on full expression patterns; those in (d,e) show stochastic labeling. (a) Distribution of presynaptic sites in different medulla layers for three Cm cell types. LM images show projections through reoriented substacks (selected to show medulla layers) of the expression of a synaptic marker (syt-HA; magenta) and a membrane marker (green) driven by the indicated split-GAL4 lines. EM-based images show presynaptic sites (magenta) and EM meshes (green) of cells of the indicated types in a slice of the medulla selected to show layer patterns. (b) Cell body locations. While the soma location of individual cells is variable, general areas with cell bodies (indicated by asterisks) are similar within a cell type. Examples: TmY14 (cell bodies in a wedge-shaped subregion of the medulla cell body rind (MECBR), Tm30 (cell bodies in the ventral MECBR), and Pm6 (cell bodies in a cluster at the dorsal-medial edge of the medulla). (c) Regional arborization patterns. Examples: Dm-DRA1 (dorsal rim), MeLo4 (ventral medulla and lobula) and MeLo3b (dorsal medulla and subregion of dorsal lobula). The SS00368 driver also labels MeVP14 cells which overlap with MeLo3b in the medulla but extend into the central brain. (d) Arbor shape. Pm10 terminals have a more compact shape than those of Pm3 cells. Cm10 cells spread across nearly the full length of the medulla along the DV axis but are much narrower along the AP axis. (e) Arbor size indicates that SS34201 is expressed in Pm6 cells. SS00814 cells appears to be a better match to Pm2a than to Pm2b cells but the driver might also express in a combination of these cell

types. (f) Examples of atypical cells observed in the EM reconstructions and LM examples with similar morphology, indicating the EM morphology is unlikely to be a reconstruction error. (Left two panels) Two clock neurons (the 5th s-LNv and LNd6) that typically have different cell body locations (asterisks) and similar projection patterns, show similar cell body locations and, in one case, an unusual axonal path in the optic lobe dataset (cells in magenta, asterisks mark cell body locations). For comparison, hemibrain reconstructions of the same cell types are shown in green. Most available LM images show the typical morphology, but we found one brain in which the expression pattern of a split-GAL4 line in one hemisphere matches the cell shape and cell body distribution seen in the optic lobe dataset. (Right four panels) An unusual cell (annotated as Pm7_Li28 in the EM dataset) has LM counterparts. From left to right: Optic lobe layer pattern of a driver line labeling both Pm7 and Li28 overlaid with registered EM reconstructions of these cells. Stochastic LM labeling of a Pm7 and an Li28 cell. The combined Pm7_Li28 cell in the EM (displayed on the standard brain). An LM example of a similar cell is displayed in a similar view. (g) Overlay of a segmented LM image of LoVP109, labeled using split-GAL4 line SS27802, with a matching unnamed reconstruction (bodyId 1288888967) in the (female) hemibrain volume[8]. LoVP109 is also present in the (female) FAFB/FlyWire dataset (type LTe12[16], see Extended Data Fig. 16) but was not found in the (male) optic-lobe dataset. However, LM images of male and female brains (with SS27802 used to visualize LoVP109) indicate that this cell type is not female-specific (image on the right).

**a** summary of cell type matching from (male) Optic Lobe to (female) FlyWire; detailed in Supplementary Table 7

| matched as | ONIN # types # cells | ONCN # types # cells | VPN # types # cells | VCN # types # cells | Other # types # cells | Total # types # cells |
|---|---|---|---|---|---|---|
| 1-to-1 | 135 (90.6%) *14652 (94.2%)* | 89 (93.7%) *32212 (99.4%)* | 300 (85.3%) *3753 (84.6%)* | 92 (88.5%) *197 (75.2%)* | 28 (87.5%) *46 (86.8%)* | 644 (88.0%) *50860 (96.5%)* |
| many-to-1 | 8 (5.4%) *785 (5.0%)* | 3 (3.2%) *88 (0.3%)* | 38 (10.8%) *637 (14.4%)* | 12 (11.5%) *65 (24.8%)* | - | 61 (8.3%) *1575 (3.0%)* |
| 1-to-many | 1 (0.7%) *90 (0.6%)* | 1 (1.1%) *43 (0.1%)* | 4 (1.1%) *21 (0.5%)* | - | 2 (6.3%) *4 (7.5%)* | 8 (1.1%) *158 (0.3%)* |
| many-to-many | - | - | 6 (1.7%) *14 (0.3%)* | - | - | 6 (0.8%) *14 (0.03%)* |
| unmatched | 5 (3.4%) *23 (0.1%)* | 2 (2.1%) *51 (0.2%)* | 4 (1.1%) *10 (0.2%)* | - | 2 (6.3%) *3 (5.7%)* | 13 (1.8%) *87 (0.2%)* |

**b** examples of matched sets of Visual Projection Neurons (VPNs) from both datasets

**c** unmatched cell type

1-to-1

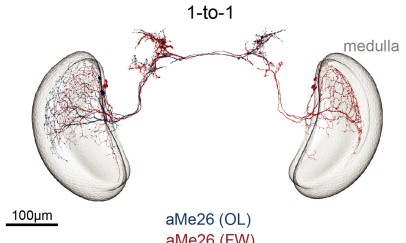

2-to-1

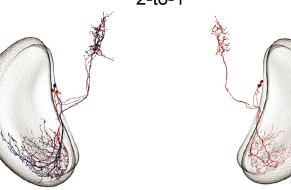

medulla

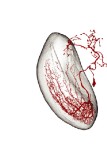

medulla

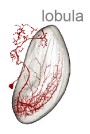

lobula

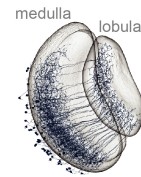

medulla    lobula

100μm

aMe26 (OL)
aMe26 (FW)

MeVP40+MeVP42 (OL)
MTe17 (FW)

LTe12 (FW)

Tm26 (OL)

**Extended Data Fig. 16 | Matching cell types between the male optic lobe and FlyWire datasets (related to Supplementary Table 7).** Related to Supplementary Table 7. (a) Summary table of cell type matching from the (male) Optic Lobe (OL) to the (female) FlyWire (FW) dataset. The full set of matches across cell types is detailed in Supplementary Table 7. This table shows the number of matched cell types and cells for each dataset, categorized by optic lobe cell type groups (Fig. 1c) and the level of matching: 1-to-1, many-to-1, 1-to-many, many-to-many, and unmatched. The counts here are referenced to the male optic lobe dataset and do not include LoVP109 which was not found in the OL dataset. Compared to the counts in Fig. 1e, the tabulated data here only include the 'right dominant' neurons and so are slightly smaller. (b) Examples of matched sets of neurons from both datasets. The left panel shows an example of a '1-to-1' match, with the neuron type aMe26 from both datasets. For most

cell types, the FlyWire annotations of Schlegel et al.[16] identify neurons on both sides of the brain. The right panel shows an example of a '2-to-1' match, with the cell types MeVP40 and MeVP42 from the OL dataset matched to the type MTe17 in FW. (c) Examples of candidate dimorphic neurons, each identified in one dataset but not the other. The left panel shows the neuron type LTe12 from the FW dataset (see Extended Data Fig. 15g for additional information about LTe12/LoVP109 data), while the right panel shows the cell type Tm26 from the OL dataset (only the right optic lobe is shown). Tm26 appears to be the cell type previously reported as male-specific[125], and an additional unmatched OL cell type (LoVP92, not shown, see Supplementary Table 7) resembles a different previously described male-specific cell type[126]. Images in b and c are shown at the same scale and perspective.

**Extended Data Table 1 | The counts of presynapses and the completeness of connections for the optic lobe brain neuropils and the layers of the medulla, lobula, and lobula plate**

| Neuropil | presynapses ($\times 10^3$) | completeness (percentage) | | |
|---|---|---|---|---|
| | | pre | post | connection |
| medulla (total) | 4146 | 96 | 57 | 55 |
| layer M1 | 504 | 93 | 60 | 57 |
| layer M2 | 411 | 97 | 64 | 62 |
| layer M3 | 624 | 94 | 50 | 48 |
| layer M4 | 229 | 97 | 54 | 53 |
| layer M5 | 206 | 92 | 52 | 48 |
| layer M6 | 385 | 96 | 52 | 50 |
| layer M7 | 235 | 98 | 47 | 46 |
| layer M8 | 480 | 98 | 56 | 55 |
| layer M9 | 738 | 98 | 62 | 61 |
| layer M10 | 334 | 98 | 62 | 61 |
| lobula (total) | 1943 | 98 | 51 | 50 |
| layer LO1 | 130 | 98 | 68 | 67 |
| layer LO2 | 288 | 98 | 59 | 58 |
| layer LO3 | 301 | 98 | 55 | 54 |
| layer LO4 | 251 | 97 | 50 | 49 |
| layer LO5A | 267 | 98 | 49 | 48 |
| layer LO5B | 477 | 98 | 47 | 46 |
| layer LO6 | 228 | 98 | 43 | 42 |
| lobula plate (total) | 869 | 98 | 61 | 60 |
| layer LOP1 | 259 | 98 | 59 | 58 |
| layer LOP2 | 230 | 98 | 61 | 59 |
| layer LOP3 | 181 | 98 | 63 | 62 |
| layer LOP4 | 199 | 98 | 61 | 60 |
| accessory medulla | 7 | 97 | 56 | 54 |
| lamina | 223 | 16 | 31 | 5 |

'Pre' completeness is the percentage of presynapses contained in 'traced' neurons. 'post' completeness is the percentage of postsynapses assigned to traced neurons. Connection completeness refers to the percentage of synapses for which both the presynapse and the postsynapse are in traced neurons. Neurons designated as 'status = Traced' in neuPrint have passed multiple rounds of quality checks, are believed to contain no false merges, and have all of their main branches within the volume reconstructed (whenever possible, corroborated by comparison to other neurons of the same type). The lamina posed substantial reconstruction challenges (see Methods).

# Reporting Summary

## Statistics

For all statistical analyses, confirm that the following items are present in the figure legend, table legend, main text, or Methods section.

| n/a | Confirmed | |
|---|---|---|
| ☒ | ☐ | The exact sample size (*n*) for each experimental group/condition, given as a discrete number and unit of measurement |
| ☒ | ☐ | A statement on whether measurements were taken from distinct samples or whether the same sample was measured repeatedly |
| ☐ | ☒ | The statistical test(s) used AND whether they are one- or two-sided *Only common tests should be described solely by name; describe more complex techniques in the Methods section.* |
| ☒ | ☐ | A description of all covariates tested |
| ☒ | ☐ | A description of any assumptions or corrections, such as tests of normality and adjustment for multiple comparisons |
| ☐ | ☒ | A full description of the statistical parameters including central tendency (e.g. means) or other basic estimates (e.g. regression coefficient) AND variation (e.g. standard deviation) or associated estimates of uncertainty (e.g. confidence intervals) |
| ☐ | ☒ | For null hypothesis testing, the test statistic (e.g. *F*, *t*, *r*) with confidence intervals, effect sizes, degrees of freedom and *P* value noted *Give P values as exact values whenever suitable.* |
| ☒ | ☐ | For Bayesian analysis, information on the choice of priors and Markov chain Monte Carlo settings |
| ☒ | ☐ | For hierarchical and complex designs, identification of the appropriate level for tests and full reporting of outcomes |
| ☒ | ☐ | Estimates of effect sizes (e.g. Cohen's *d*, Pearson's *r*), indicating how they were calculated |

*Our web collection on statistics for biologists contains articles on many of the points above.*

## Software and code

Policy information about availability of computer code

| | |
|---|---|
| Data collection | This project relied on code at many levels for data collection, reconstruction, curation, etc. All details are extensively documented in the Methods and the software tools used at each step are referenced. |
| Data analysis | The python code that replicates our analysis and data visualization is available via GitHub: https://github.com/reiserlab/male-drosophila-visual-system-connectome-Python-3.10 (or higher) is required with further dependencies detailed in GitHub repository. We make use of: python 3.12.5, navis 1.7, neuprint-python 0.4.26, snakemake 8.20.3, pymupdf 1.24.10, cloud-volume 10.4, google-cloud-storage 2.18.2 fastcluster 1.2.6, numpy 1.26, alphashape 1.3.1, kneed 0.8.5. The Code Availability section of the methods explains this repository. Extensive documentation is available in this repository to help readers replciate our figures. |

For manuscripts utilizing custom algorithms or software that are central to the research but not yet described in published literature, software must be made available to editors and reviewers. We strongly encourage code deposition in a community repository (e.g. GitHub). See the Nature Portfolio guidelines for submitting code & software for further information.

## Data

Policy information about availability of data

 All manuscripts must include a data availability statement. This statement should provide the following information, where applicable:

- Accession codes, unique identifiers, or web links for publicly available datasets
- A description of any restrictions on data availability
- For clinical datasets or third party data, please ensure that the statement adheres to our policy

All data is publicly available.

Data Availability statement:
The connectome data is directly accessible via the neuPrint database server: https://neuprint.janelia.org/?dataset=optic-lobe:v1.1

The Cell Type Explorer web resource is available at: https://reiserlab.github.io/male-drosophila-visual-system-connectome/index.html
and can also be downloaded as a zip file from DOI:10.5281/zenodo.10891950

The SWC skeleton is available through neuPrint+ (web interface) by clicking the bodyId in the skeleton viewer, which will provide a download button. For bulk downloads, the meshes of neurons, ROI boundaries and skeletons are provided as Google Cloud Storage buckets at gs://flyem-optic-lobe. In our shared code (see Code availability) we provide programmatic access to these files. For example, the skeletons and meshes of individual neurons are accessible through the OLNeuron class. The example Jupyter notebook: https://github.com/reiserlab/male-drosophila-visual-system-connectome-code/blob/main/src/python-bootcamp/access_skeleton_and_mesh.ipynb shows how to store the skeleton as a *.swc file and the mesh as a Wavefront *.obj file.

The LM-EM transformation vectors from our EM sample to the JRC2018M template brain are at https://figshare.com/s/e17528e5e2c44ba78b5d, also stored at gs://flyem-optic-lobe/transforms/MaleCNS_JRC2018M.h5. In that file, the "dfield" transformation vectors map points from EM space to LM template space, and "invdfield" vectors map points in the opposite direction.

Images of split-GAL4 driver lines are available at https://splitgal4.janelia.org/cgi-bin/splitgal4.cgi.

## Research involving human participants, their data, or biological material

Policy information about studies with human participants or human data. See also policy information about sex, gender (identity/presentation), and sexual orientation and race, ethnicity and racism.

| | |
|---|---|
| Reporting on sex and gender | n/a |
| Reporting on race, ethnicity, or other socially relevant groupings | n/a |
| Population characteristics | n/a |
| Recruitment | n/a |
| Ethics oversight | n/a |

Note that full information on the approval of the study protocol must also be provided in the manuscript.

# Field-specific reporting

Please select the one below that is the best fit for your research. If you are not sure, read the appropriate sections before making your selection.

☒ Life sciences      ☐ Behavioural & social sciences      ☐ Ecological, evolutionary & environmental sciences

For a reference copy of the document with all sections, see nature.com/documents/nr-reporting-summary-flat.pdf

# Life sciences study design

All studies must disclose on these points even when the disclosure is negative.

| | |
|---|---|
| Sample size | The EM dataset reported in this study represents the visual system of a single male fly.  Over 50000 individual neurons were reconstructed. Light microscopy (LM) images of genetic driver lines include at least two (typically more) images per driver line. |
| Data exclusions | No data were excluded. |
| Replication | The EM dataset is currently the only reconstruction of the visual system of a male Drosophila melanogaster and therefore cannot be independently replicated at this time.  However, many of the observed cell morphologies are supported by LM data and, for some cell types, |

prior EM data.  LM images of expression patterns of driver lines were examined for at least two flies. Since the original submission we have compared all the cell types to the FlyWire dataset of a female brain and have matched 98% of the cell types between the data sets.

| Randomization | N/A |
| Blinding | N/A |

# Reporting for specific materials, systems and methods

We require information from authors about some types of materials, experimental systems and methods used in many studies. Here, indicate whether each material, system or method listed is relevant to your study. If you are not sure if a list item applies to your research, read the appropriate section before selecting a response.

## Materials & experimental systems

| n/a | Involved in the study |
|---|---|
| ☐ | ☒ Antibodies |
| ☒ | ☐ Eukaryotic cell lines |
| ☒ | ☐ Palaeontology and archaeology |
| ☐ | ☒ Animals and other organisms |
| ☒ | ☐ Clinical data |
| ☒ | ☐ Dual use research of concern |
| ☒ | ☐ Plants |

## Methods

| n/a | Involved in the study |
|---|---|
| ☒ | ☐ ChIP-seq |
| ☒ | ☐ Flow cytometry |
| ☒ | ☐ MRI-based neuroimaging |

## Antibodies

| Antibodies used | We used antibodies against epitope tags (HA, V5 and FLAG) and GFP to detect transgenically expressed reporter constructs and a widely used monoclonal antibody against Brp (clone: mAb Nc82) as a general neuropile label.  We use well-established protocols that have been used in many studies, with additional details described in our previous studies: Wu et. Al. 2016, Nern et al 2015, Meissner et al 2023, Meissner et al 2024, and availabe: https://www.janelia.org/project-team/flylight/protocols |
| Validation | All antibodies used in this study have been extensively used and validated previously (https://www.janelia.org/project-team/flylight/protocols) |

## Animals and other research organisms

Policy information about studies involving animals; ARRIVE guidelines recommended for reporting animal research, and Sex and Gender in Research

| Laboratory animals | All experiments used laboratory strains of Drosophila melanogaster. This study reports a collection of transgenic fly lines (split-AGL4 lines), genotypes of which are listed in Supplementary Table 6.  Images are of 1-8 day old flies. |
| Wild animals | No wild animals were used. |
| Reporting on sex | The EM dataset is of a single male fly.  For light microscopy images, fly sex is included in the image metadata (available with the images at https://splitgal4.janelia.org/cgi-bin/splitgal4.cgi ). |
| Field-collected samples | No field collected samples were used. |
| Ethics oversight | No ethics oversight required. |

Note that full information on the approval of the study protocol must also be provided in the manuscript.

## Plants

| Seed stocks | n/a |
| Novel plant genotypes | n/a |
| Authentication | n/a |

