## [Peer Review file · Nature]

Connectome-driven neural inventory of a complete visual system

Corresponding Author: Dr Michael Reiser

Version 0:

Reviewer comments:

Referee #1

(Remarks to the Author)

This is a truly outstanding paper. In this work, a large team of researchers present a completely new connectome dataset with an unprecedented, thorough and careful curation of neuronal connectivity in the *Drosophila* visual system. It is the first complete, isotropic volume of a male *Drosophila* optic lobe with all neurons annotated and assigned to cell types (including visual projection neurons). The manuscript combines a tremendous amount of work from sample preparation to EM imaging, segmentation, proofreading, neurotransmitter and synapse prediction, neuron and neuropil annotation and connectivity and morphology analysis. In addition, a very large collection of genetic driver lines (577 split Gal4s) complements the EM data and provides essential access points for experiments in vivo.

I am aware of (but not involved in) recent new analyses of existing, non-isotropic connectome data; compared to the existing data, the new connectome presented here is a quantum leap. The data, insights and tools presented in this paper will motivate and drive experiments in fields beyond circuit neuroscience in flies. Specifically, I see important applications in computational circuit analysis and a basis to integrate cell biological and gene expression data in a functional context. From my viewpoint, I even see new inspirations and entry points to study circuit development by better understanding stereotypy of the outcome in terms of neuronal morphologies and synaptic specificity.

In sum, I strongly recommend consideration of this work as an article in Nature. I only have relatively moderate and minor suggestions to improve the paper prior to publication.

- (1) The authors state that not all inner photoreceptors could be proofread as the segmentation quality in some region (posterior side) is insufficient - most likely due to strong fixation. Can the authors exclude that there is a "spatial quality gradient" in the data due to uneven fixation of the sample? Or with other words: is this a problem specific to the inner photoreceptors or to a specific region in the medulla? How could one differentiate between column-to-column differences in connectivity due to intrinsic variability or variable data quality?
- (2) The authors say little about the amount of proofreading that was required for this dataset. I am aware from a set of current preprints that proofreading efforts of the so-called FAFB dataset raised concerns about variable proofreading quality due to different individual proofreaders. What measures were taken to ensure an even proofreading quality throughout the dataset? Again, is there a way to differentiate between column-to-column differences due to intrinsic variability or variable proofreading quality?
- (3) A recent preprint ("Neuronal "parts list" and wiring diagram for a visual system") attempts to curate a similar (yet incomplete) inventory although in a different dataset with a different approach. From an outside perspective, it would certainly be helpful if the authors of both manuscripts could employ a common neuron type nomenclature for at least major neuron types and groups of neuron types. Some examples: Cm vs. Sm, Tm5a,b,c vs Tm5a,b,c,d,e,f. Obviously the authors cannot be held responsible for the nomenclature of others - but I think they should comment on the issue and do everything possible to facilitate cross-comparisons. Maybe they can further comment on how feasible it might be to actually reach unified nomenclature?
- (4) The authors introduce Optic Lobe Intrinsic Neurons (OLINs) and Optic Lobe Connecting Neurons (OLCNs). This naming scheme is a bit ambiguous: aren't Optic Lobe Connecting Neurons also intrinsic to the optic lobe? They connect optic neuropils and not optic lobes (which MeVPM neurons do I guess). The authors themselves state in line 102-103 that "the lamina, medulla, accessory medulla, lobula, and lobula plate, [...] <form> the structure called the optic lobe". Would Optic Neuropil Intrinsic Neurons (ONINs) and Optic Neuropils Connecting Neurons (ONCNs) be suitable alternatives?
- (5) The connectivity jitter plots (e.g. in Fig. 2h but especially in extended figure 3) are unable to show the data distribution. Can the authors provide histograms or perhaps adjust the jitter and/or transparency?

Other minor suggestions:

- In Supplementary Fig. 1: for some neuron types the top 5 output (e.g LoVP74, (LT51 only one top 5 input) is missing. For others the input seems missing e.g. MeVPM8 also the density plot shows at least some input.
- Fig. 3a Too small to see missing cells
- Fig. 5b While these celltype summary plots are extremely helpful and informative it would help to understand the circuit/pathway (from lamina monopolar cell to LC neurons) with a circuit diagram
- Fig. 5d to what is the missing column label referring to?
- Extended Data Figure 4c 3: R7p label is missing
- Line 229-230 how does this extend to the lobula? Isn't T4 only connecting medulla with lobula plate?
- Line 300-301 Is it that each synapse is predicted to be 50/50 ACh/Dop or is the average over the entire neuron 50/50 ACh/Dop with some synapses clearly being predicted to be ACh or Dop?
- Line 350-353 The "The Drosophila Visual System Cell Type Explorer" is a great webpage! Here are some nice-to-have features:
 - Information about nt prediction
 - Interactive 3D viewer of all neurons of the neuron type

(Remarks on code availability)

Referee #2

(Remarks to the Author)

The paper presents a monumental effort of electron microscopy (EM) reconstruction and cataloging of the neurons encompassing the fly optic lobe. To say that this is heroic would be euphemistic. The level of detail and the extreme care taken throughout the process of alignment, segmentation, synapse identification, proofreading and annotation is truly remarkable. The outcome of this endeavor is highly successful with a comprehensive list of cells mapped onto the hexagonal grid composing the fly eye and with a coordinate system maintained across each optic lobe neuropil. This provides an unprecedented opportunity to match the structural organization of the fly visual system to the point-by-point view of the visual world from photoreceptors all the way to projections into the central brain. The predictions that can be drawn from this quantitative connectome, combined with the set of genetic fly lines that target single cell types, unleash the full potential for a comprehensive understanding of fly vision. In short, this is a generational resource to gain insight, develop hypotheses, and obtain the tools necessary to investigate every basic visual computation in a very powerful model organism. In the framework of the three levels of analysis proposed by David Marr, this work is arguably the best description to date of how the implementational level (i.e., the physical structure) informs the algorithmic level (i.e., the rules) that in turn explains the computational one (i.e., the goal). In this sense, the work constitutes a profoundly transformative achievement for the visual neuroscience community.

The paper is entirely descriptive, clearly explaining the classification approach by walking the reader through the rationale for branchpoint decisions. I particularly appreciated the cell type quantitative summary (Fig. 5b and Supp. Fig. 1) in which the authors have done a great job capturing and synthesizing the essential properties that characterize different cell types (neurotransmitter, morphology, connectivity, and size) while also leaving lots of room for further quantitative analysis and experimental analysis. Moreover, the synapse-by-depth metric that sorts neurons into their respective types nearly as effectively as the connectivity-based clustering is impressively compact and clever. This allows comparisons between EM and light microscopy (LM) data. The Cell Type Explorer web application is an extraordinarily useful resource to easily access connectivity and spatial distribution of synapses and cells in the hexagonal coordinate system. Fly researchers will strongly benefit from this open-source platform that speeds up the emergence of valuable insights.

This study forms the highest benchmark standard for the preparation and dissemination of forthcoming EM connectomic data sets.

I have no scientific criticisms; the work represents the first complete connectome of male fly visual system (and more). This is a milestone for the neuroscience community that, together with the first whole-brain connectome of a female fly (FAFB, FlyWire), will help bridge the gap between neural structure and function by guiding strong experimental predictions to test.

Minor:

L93: I would give a bit more context here by adding information reported in "EM volume imaging" of the Methods section "... imaged using seven customized FIB-SEM systems in parallel over a period of almost a year"

L127-128: "motion vision pathway" in this case the authors are referring to "direction motion vision pathway"

L294: for consistency use fan-out with or without hyphen. Moreover, for the broader audience, it might be helpful to add a short sentence about the authors' interpretation/speculation of the different fan-out among neurotransmitters

L336: on LC11, the authors might also cite Keleş and Frye, 2017 (Current Biology)

L767: TM5a

Methods:

“EM volume synapse identification”, Line3: dot missing after “detection” and maybe a citation?

“LM-EM volume correspondence”, Line3: “for for”

Add a space before “Pipeline for rendering neurons”

Fig. 3a: the picture of the fly head on the right might be confusing because it looks like a picture from outside (i.e., left eye), I suggest to use only a silhouette of the head without depth information (view from inside) or to flip the image along the vertical axis so that the fly is looking to the right and the eye corresponds to the right one from an outside view

Caption of Fig. 3c: marker is between different quotation marks font

Add a short description (software, package, etc.) for the ANCOVA run in Fig. 4i

(Remarks on code availability)

The data from the paper have been made openly and easily accessible to everyone without coding experience through web interfaces (neuPrint and Cell Type Explorer). The data can also be accessed by using a well-established API for which the authors have created a thorough guide to get started in Python and replicate the analysis.

Referee #3

(Remarks to the Author)

This is an impressive manuscript. Aljoscha Nern and colleagues follow in the footsteps of Ramón y Cajal and finish what the latter started approximately 140 years ago: a catalogue of all cell types in the visual system of an insect. To understand a nervous system, one needs to know its components; this manuscript contains an (almost) complete inventory. In generating this dataset, the authors developed and used new software tools to analyze large amounts of electron microscopic images; they introduced a new nomenclature of cell types in the *Drosophila* optic lobe, and provide a genetic address to many of the catalogued neurons. The authors managed to condense an enormous amount of data into a surprisingly palatable format that was a joy to read. So far, most connectomic reconstructions were tainted by the lingering awareness that they were incomplete. This one is also incomplete, but it is probably as close to a complete connectome as it gets. As far as I can judge, the manuscript is of high quality and provides a rich resource that makes neuroscientists working on other model organisms green with envy.

Below please find several suggestions for improvement:

1) My principal reservation is that the work remains purely descriptive. Previously published connectomes by the same authors (e.g., Takemura et al. 2013)—incomplete as they may be—always offered some concrete insight into the function of the circuit. In the concluding remarks of the present manuscript, the authors also foresee “countless experimental roadmaps for detailed investigation of brain circuit function”, yet, they fail to provide a single example of a functional prediction that emerges from this ultrastructural analysis. Recent work (which the authors are careful to acknowledge) demonstrates how predictions emerge from a comparable connectome (Seung 2023). I encourage the authors to give us a glimpse of what we can expect to learn from efforts like this. This would provide the icing on the cake.

2) The title of the manuscript contains a claim to completeness. To preempt disappointment, I recommend to disclose early in main text that the volume contains only part of the lamina. At present, the authors discuss this minor and almost unavoidable blemish only in the Methods section. The claim to establish “the most comprehensive survey of any insect optic lobe” (lines 84 and 85), in my opinion, cries out for a comparison with the only other comprehensive electron microscopic survey of the *Drosophila* visual system (Matsliak et al. 2023). The cursory mention of this highly relevant work by the FlyWire consortium strikes me as unfair. Differences between the two connectomes—one from a female and one from a male fly—should be fleshed out. For example, can LoVP109 be found in the (openly accessible) connectome of the female optic lobe?

3) Despite the overwhelming amount of data and technical detail, I am convinced that some of the figures could be combined and presented more concisely. Figures 2, 3 and 4, for example, all describe the process by which neurons were assigned to cell types. I suggest to combine Figures 2 and 3 and move some of the space-consuming panels (e.g., Fig. 3a,d) to the Extended Data. The same applies to Figures 6 and 7. I appreciate that these figures sum up a large body of work, but they provide little useful information to the non-expert reader; and I expect that even a *Drosophila* neurobiologist cannot extract much information from these images and might prefer to consult the supplementary material or the web resource instead. Alternatively, the authors could display one exemplary rendering alongside the light micrograph of the corresponding split-GAL4 expression pattern from Fig. 8 and move the remaining panels to the Extended Data.

The following further issues deserve some thought, analyses, or editorial changes:

The paper is well written. On occasion, however, the authors seem to place a strong focus on their own past achievements. I suspect that they might look back at this in a few years' time and cringe at some of the more self-aggrandizing passages. It is

laudable that many of the authors were involved in generating vast libraries of genetic driver lines over the past 15 years that advanced the field (lines 425–427), but this is not central to this manuscript. Given the wealth of possible points to discuss, there seems to be an undue emphasis on history.

The authors acknowledge that the low number of neurons releasing certain neurotransmitters impedes the classification of, for example, octopaminergic, dopaminergic, and serotonergic synapses. Can they exclude that the overwhelming number of cholinergic synapses in the training dataset causes cholinergic misclassifications? Have they tried to train the network with equal (albeit low) numbers of synapses of each type?

Computational neuroscientists would profit from easy access to skeletons and meshes of neurons (for example, in the standardized SWC format). I tried, but failed to download the segmentations and ROIs through the respective Google Storage buckets (flyemoptic-lobe/v1.0/segmentation and flyem-optic-lobe/rois/). I would welcome an extension of the Cell Type Explorer that includes the option to download meshes.

The statement that “the prediction accuracy is further improved when the single-synapse predictions are aggregated across the cells of each type” should be clarified. Were single-synapse predictions first aggregated across synapses of one cell and then across cells of one type?

The text jumps between the present and the past tense and contains an inconsistent mix of American and British English; this should be ironed out. Some parts of the Methods section are written more eloquently, but less precisely, than others. The text fluctuates between wordy digressions (e.g., “while neurons have complex, extended 3D shapes, it is remarkable that 2D visual representations of neurons have typically served as extremely rich descriptions of cell types across generations of neuroscientists, from hand drawings of Golgi-stained neurons to computer-generated ray traces of reconstructed EM volumes”) and chains of sentences that start with “we took”, “we applied”, “we performed”. I suggest to restrict the (already extensive) Methods section to the pertinent bits, with readability in mind.

The authors provide ample references to their own work (~27 out of 58), but occasionally forget to acknowledge that of others. Half a paragraph (lines 50–59) is devoted to the motion vision pathway without a reference to recent (connectome-guided) advances (e.g., Ammer et al. 2023; Braun et al. 2023; Cornean et al. 2024; Groschner et al. 2022; Sanfilippo et al. 2024).

In lines 63–65 the authors state that “there is little mystery about the role of visual brain areas—their neurons and circuits must be involved in seeing”; this is a truism, but hardly a profound insight.

The first mention of “automatic segmentation” in line 95, should be followed by a reference to the corresponding part of the Methods section.

Lines 108–109: “Optic Lobe Intrinsic Neurons (OLINs) have synapses confined to a single region.” Region is ill-defined, I suggest to replace it with neuropil.

Lines 156–157: “Nearly all neurons in the dataset were reviewed multiple times by several team members as part of this process.” The juxtaposition of “nearly”, “multiple” and “several” in one sentence makes it appear sketchy. Perhaps the authors could state this more precisely?

Lines 173–174: I suggest to spell “cell typing” as “cell-typing” and highlight that the connectivity-defined clusters were of approximately equal cell counts—an important feature.

In lines 176–179, the authors describe that putative misclassifications of cell types were “resolved using additional criteria, such as cell morphology or, when available, genetic markers.” Why didn’t the authors use a combination of connectivity and morphology-based clustering for all neurons? I appreciate that the collection of cell-type-specific driver lines can provide important hints for cell typing, but I struggle to understand how genetic markers, whose expression patterns were visualized in separate LM experiments, can be used to resolve discrepancies in EM data. Formally, genetic markers should only inform EM cell classification if they were used to drive the expression of markers detectable in the same sample under the electron microscope, such as EMcapsulins (Sigmund et al. 2023).

The main figures are not referenced in sequence: Fig. 5 is cited before Fig. 3.

Lines 232–233: “They [volumes] can be visualized alongside the gray-scale EM data (Fig. 3d)...” Judging by Fig. 3d, I would describe this as “superimposed on” rather than “visualized alongside”.

The capitalization in line 249 seems inconsistent. Why is only “Fluorescence” capitalized?

In line 257, the neural network mentioned in passing, which was used to assign neurotransmitter likelihoods, should be accompanied by a reference to the Methods section or to the relevant literature.

The acronym EASI-FISH (line 262) should be spelled out. Instead of listing at length the methods that could be used to assign neurotransmitters to neurons, I suggest that the authors limit their description to the methods they used, but state them explicitly: The phrase “primarily using EASI-FISH” could simply be replaced by “using FISH and EASI-FISH”.

In lines 269–270, the authors state that “the consistency or accuracy of predictions is comparable between the types in the training dataset and cell types not used in training”. This statement should 1) be supported by a reference to the corresponding data/figure, and 2) be specified: Was the consistency or the accuracy comparable? If both were comparable, I suggest to replace “or” with “and”.

In line 294, the authors mention a “significant and potentially interesting difference across the neurotransmitters” with regard to the neurons' fan-out properties. The (statistical) significance should be supported by a statistical test, as in Fig. 4i.

Line 300: “Mi15s”, could be lab slang for Mi15 neurons, or it could specify a subclass of Mi15 neurons (like TmY9b). This should be clarified.

Mi15 neurons express both cholinergic and dopaminergic marker genes (Davis et al. 2020). Are the two neurotransmitters co-packaged (and co-released) at the same synapse or does one Mi15 cell house separate cholinergic and dopaminergic presynapses? A discrepancy between the confidence of single synapse predictions and cell predictions in Mi15 neurons might provide a hint.

It was not clear to me, which data the authors refer to in their descriptions in lines 340–343.

Line 451: “We use brain registration to juxtapose EM-reconstructed neurons in the same reference as registered LM data”—the term “brain registration” could do with a more detailed explanation.

Lines 471–472: “we asked whether specific connectivity patterns [...] are most prevalent”. I suggest to replace “whether” with “which” or reword the sentence.

While I am happy to read that “in the coming months, proofreading of the rest of the central brain, and eventually the ventral nerve cord, of this connected volume [?] will be completed”, I think that the text should describe the present work, rather than ambitious promises for the future.

In Fig. 2d, the abscissa is titled “# cells per connectivity-defined cluster”. In case the hue of the red boxes corresponds to the cell numbers, I believe that this should be the title of a (missing) colour map; the axis label should be “cluster #”.

The yellow colour in Fig. 2i is difficult to see.

The colour bars in Fig. 4b,e lack labels.

Line 612: Why is “Neurotransmitter” capitalized?

Line 625: Fanout/fan-out is spelled inconsistently throughout the text and figure legends.

Line 697: “...identified by the algorithm.” It is not clear, which algorithm the authors refer to.

The term “bodylds” in Extended Data Figs. 2 and 7 should be spelled “body IDs” or “bodyIDs”, as in the Supplementary Tables.

The fixation and staining of the sample in the second paragraph of the Methods section should be described in more detail, including concentrations and durations.

Compared to the ~97% of presynapses that could be assigned to named neurons, the percentage of assigned post-synapses (55%) appears low. Why?

Reference 42 (of the main text) has, in the meantime, been published and should be cited accordingly.

Methods reference 76 concerning the Trimesh Python package lacks information: “Dawson-Haggerty, M. Trimesh. (2024).”

References:

- Ammer G, Serbe-Kamp E, Mauss AS, Richter FG, Fendl S, Borst A. 2023. Multilevel visual motion opponency in *Drosophila*. *Nat. Neurosci.* 26(11):1894–1905
- Braun A, Borst A, Meier M. 2023. Disynaptic inhibition shapes tuning of OFF-motion detectors in *Drosophila*. *Curr. Biol.* 33(11):2260-2269.e4
- Cornean J, Molina-Obando S, Gür B, Bast A, Ramos-Traslosheros G, et al. 2024. Heterogeneity of synaptic connectivity in the fly visual system. *Nat. Commun.* 15(1):1570
- Davis FP, Nern A, Picard S, Reiser MB, Rubin GM, et al. 2020. A genetic, genomic, and computational resource for exploring neural circuit function. *eLife.* 9:e50901
- Groschner LN, Malis JG, Zuidinga B, Borst A. 2022. A biophysical account of multiplication by a single neuron. *Nature.* 603(7899):119–23
- Matsliah A, Yu S, Kruk K, Bland D, Burke A, et al. 2023. Neuronal “parts list” and wiring diagram for a visual system. 2023.10.12.562119 Preprint at <https://doi.org/10.1101/2023.10.12.562119> (2023).

Sanfilippo P, Kim AJ, Bhukel A, Yoo J, Mirshahidi PS, et al. 2024. Mapping of multiple neurotransmitter receptor subtypes and distinct protein complexes to the connectome. *Neuron*. 112(6):942-958.e13
Seung HS. 2023. Insights into vision from interpretation of a neuronal wiring diagram. 2023.11.15.567126
Preprint at <https://doi.org/10.1101/2023.11.15.567126> (2023).
Sigmund F, Berezin O, Beliakova S, Magerl B, Drawitsch M, et al. 2023. Genetically encoded barcodes for correlative volume electron microscopy. *Nat. Biotechnol.* 41(12):1734–45
Takemura S, Bharioke A, Lu Z, Nern A, Vitaladevuni S, et al. 2013. A visual motion detection circuit suggested by *Drosophila* connectomics. *Nature*. 500(7461):175–81

(Remarks on code availability)

I tested only a sample of the many Jupyter Notebook files. Virtually all important files were well annotated and included README documents. The code provides a valuable resource.

Version 1:

Reviewer comments:

Referee #1

(Remarks to the Author)

The authors provide an extensive revision and have addressed all points raised in my review. In particular, the authors provide additional analysis (Extended Data Figure 1) that demonstrates that the data quality inconsistency of PRs is restricted to this cell type and not to a region, and the matching of cell types in this manuscript to the recent flywire optic lobe dataset. The remarkable consistency is a powerful confirmation of the cell type assignment approach. I congratulate the authors to this important, impactful paper.

Prior to publication, there is only one section where the revised version might still need some simple clarification: The method for assigning DRA / p / y types to columns underwent a considerable workover. In addition to not only using connectivity information as in the previous version the authors now also use morphological features to do the assignment. This allows the assignment to also extend into the region of the medulla where photoreceptor quality is not optimal.

However, surprisingly, this approach drastically changed the overall number of pale and yellow columns and subsequently the ratio of pale / yellow columns. I am not sure about the morphological features used here and would love to see some clarification in the final manuscript on the following questions:

- Are there clear, quantifiable criteria for a Dm8 home column which can be used for the assignment?
- Do Dm8b neurons also have home columns which can be used to assign the pale type to columns with potentially missing aMe12 branches?
- The authors argue that a substantial number of R7_unclear neurons maybe R7y even though the connectivity suggests that these neurons are R7p (preference connection to Tm5b & Dm8b). Doesn't this question the connectivity argument (R7y > Dm8a & Tm5a and R7p > Dm8b & Tm5b) that the authors used and that is reported in the literature?

Incidentally, the optic-lobe:1.0 version is no longer available from neuprint at the time of writing this review. It would be very welcome if the older version were still available from neuprint.

I support to enhance the transparency of the peer review process, as suggested in the reviewer guidelines, by signing this review: Robin Hiesinger

(Remarks on code availability)

Referee #2

(Remarks to the Author)

The authors have adequately addressed my suggestions and I commend the authors for making substantive changes to the manuscript, which I feel is significantly improved. I have no further comments and I strongly recommend the paper for publication in *Nature*.

(Remarks on code availability)

Although I am not very familiar with Python, the code includes sufficient instructions. However, I encountered multiple issues with dependencies installation and couldn't obtain all the necessary libraries to run the src/completeness/connection-completeness_named.py script. Additionally, I was unable to locate the Jupyter notebook page https://github.com/reiserlab/male-drosophila-visual-system-connectome-code/blob/main/src/pythonbootcamp/access_skeleton_and_mesh.ipynb for storing skeletons as meshes.

Referee #3

(Remarks to the Author)

The authors have addressed all of my concerns. This paper sets a new standard for connectomes to come. Congratulations!

Four minor points should be rectified before publication:

1. The updated Fig. 4e contains neurotransmitter predictions for 78 cell types, but the corresponding caption still refers to 79 cell types.
2. The link to the example Jupyter notebook for accessing skeletons and meshes (https://github.com/reiserlab/male-drosophila-visual-system-connectome-code/blob/main/src/pythonbootcamp/access_skeleton_and_mesh.ipynb) does not work and the promised button to access SWC skeletons via the neuPrint+ webpage is either difficult to find or not yet implemented. This should be fixed ahead of publication.
3. The collection of FlyWire papers has been published in the meantime and the corresponding references should be updated accordingly.
4. The example chosen by the authors to demonstrate the value of connectomics in the introduction is, in my opinion, a particularly unfortunate one. The connectome by Takemura et al. (2013)—the most extensive of its kind at the time—proposed a circuit mechanism to compute the direction of visual motion. The reconstruction, and the suggested mechanism, lacked crucial cellular detail, but inspired a functional follow-up based on false premise (Behnia et al., 2014). The authors cite both papers but omit to mention recent work that seems to have clarified one mechanism of direction selectivity (Groschner et al., 2022). If the authors decide to keep this example, but find it impractical to pick and choose from the many relevant papers, perhaps they should instead refer to one of the recent reviews on the topic.

(Remarks on code availability)

I downloaded and tested samples of the extensive collection of code, all of which contained sufficient explanation in README files to reproduce the results. As mentioned above, the file 'access_skeleton_and_mesh.ipynb' is missing from the repository.

Dear Reviewers:

Our team deeply appreciated these detailed reviews, the supportive comments, and the many helpful suggestions. We are truly humbled by the high regard our work has received. We have updated the manuscript, methods, and figures to reflect all of this feedback and to include some minor updates to the data set (now called optic-lobe:v1.1). The major changes to the manuscript include:

- A new figure summarizing the consistency of the proofreading across the data set (new **Extended Data Figure 1**) and new analyses showing that data quality issues that affect a subset of photoreceptors do not extend to other cells of the optic lobe.
- A systematic comparison of all cell types in our optic lobe data set (male brain, FIB-SEM) to the cell types in FAFB-FlyWire (female brain, TEM). This comparison is detailed in the **Methods** and summarized in new **Extended Data Fig. 14** and new **Supplementary Table 7**.
- Substantially updated **Extended Data Fig. 4** and the associated **Methods** section to include an additional method and further analyses for classifying pale and yellow columns and their R7/R8 photoreceptors.
- Many updates to the **Methods** to include all requested additional information and thorough editing to make the style more consistent.
- Many manuscript figures have been updated to reflect the v1.1 database release which includes some additional proofreading and further curation of the cell types.
- Provide additional evidence that one cell type, LoVP109, is simply missing from our data set but is not a sexually dimorphic neuron (**Extended Data Fig. 13g**).
- Many additional figure updates were requested by the reviewers—8/9 primary figures and 12/14 Extended Data figures have been updated since our original submission.

Please find our point-by-point response to all comments below. Figure references are all to the revised manuscript, where the Extended Data figure numbers have been incremented by one from the original submission.

Yours sincerely,

Aljoscha Nern, Stuart Berg, Gerry Rubin, and Michael Reiser

Reviewer #1

This is a truly outstanding paper. In this work, a large team of researchers present a completely new connectome dataset with an unprecedented, thorough and careful curation of neuronal connectivity in the *Drosophila* visual system. It is the first complete, isotropic volume of a male *Drosophila* optic lobe with all neurons annotated and assigned to cell types (including visual projection neurons). The manuscript combines a tremendous amount of work from sample preparation to EM imaging, segmentation, proofreading, neurotransmitter and synapse prediction, neuron and neuropil annotation and connectivity and morphology analysis. In addition, a very large collection of genetic driver lines (577 split Gal4s) complements the EM data and provides essential access points for experiments *in vivo*.

I am aware of (but not involved in) recent new analyses of existing, non-isotropic connectome data; compared to the existing data, the new connectome presented here is a quantum leap. The data, insights and tools presented in this paper will motivate and drive experiments in fields beyond circuit neuroscience in flies. Specifically, I see important applications in computational circuit analysis and a basis to integrate cell biological and gene expression data in a functional context. From my viewpoint, I even see new inspirations and entry points to study circuit development by better understanding stereotypy of the outcome in terms of neuronal morphologies and synaptic specificity.

In sum, I strongly recommend consideration of this work as an article in *Nature*. I only have relatively moderate and minor suggestions to improve the paper prior to publication.

We sincerely thank the reviewer for this thorough evaluation and the generous assessment of our work.

- (1) The authors state that not all inner photoreceptors could be proofread as the segmentation quality in some region (posterior side) is insufficient - most likely due to strong fixation. Can the authors exclude that there is a “spatial quality gradient” in the data due to uneven fixation of the sample? Or with other words: is this a problem specific to the inner photoreceptors or to a specific region in the medulla? How could one differentiate between column-to-column differences in connectivity due to intrinsic variability or variable data quality?

Based on our deep familiarity with the data set and extensive efforts to improve the photoreceptors specifically, we found no evidence of a “spatial quality gradient” affecting the connectome beyond a subset of the R7/R8 photoreceptors. However, we did not explain this properly in the submitted manuscript, and we appreciate the request to clarify. We now provide new analysis in the new **Extended Data Fig. 1** that excludes the possibility of a “spatial quality gradient.” **Extended Data Fig. 1g,h** shows the mean output connections by cell type in two groups of columns, partitions based on the number of R7 output connections. As these groups essentially lie along the equality line, except for three cell types, we can see that the

reconstruction challenges with R7 and R8 cells do not spill over to any neurons except the Dm9 cells with which they share extensive physical contact. We have also extended our method for identifying pale and yellow photoreceptors to now make the assignments on medulla columns rather than individual R7/R8 cells (described in **Methods: Assigning R7 and R8 photoreceptors and medulla columns to different types of ommatidia**, and shown in **Extended Data Fig. 5a**). This method further mitigates the data limitations of the missing photoreceptor cells, especially since their most prominent downstream targets—Tm5a/b and Dm8a/b—are reliably sorted across the medulla (**Extended Data Fig. 5b-e**).

The second analysis, in **Extended Data Fig. 1e,f** shows that the “postsynapse completeness fraction” the fraction of downstream connections that are captured by “traced” neurons, i.e., included in the connectome, are uniformly distributed across the medulla, lobula and lobula plate, with no gradient in the ratio of orphaned to captured synapses. This demonstrates that the completeness of the data set is approximately uniform across columns; that is, the probability of any synapse being connected between two reconstructed and named neurons is nearly uniform across these retinotopic neuropils, ruling out variable data quality as a major factor in our connectome.

- (2) The authors say little about the amount of proofreading that was required for this dataset. I am aware from a set of current preprints that proofreading efforts of the so-called FAFB dataset raised concerns about variable proofreading quality due to different individual proofreaders. What measures were taken to ensure an even proofreading quality throughout the dataset? Again, is there a way to differentiate between column-to-column differences due to intrinsic variability or variable proofreading quality?

We thank the reviewer for the opportunity to explain the effort and care that we have put into assembling this dataset. The FlyEM team at Janelia has learned many lessons from prior connectome projects, and we realized that much of our process is not well described in previous publications. We now provide an extensive explanation of how we managed the proofreading process in **Methods: EM volume proofreading**. One of the key details is that our process is staged and tasks are randomly and automatically assigned to proofreaders. This means that every neuron is “touched” by many people at different stages and is subject to several rounds of review. In addition, we allocate approximately 10% of total proofreading time toward quality control tasks to measure the overall data consistency and the performance of each proofreader (remarkably, the agreement rate between proofreaders on, e.g., “merge” vs. “don’t merge” decisions is over 99%). Along with this expanded **Methods** section, we have now added additional metrics summarizing the proofreading and quality control (**Extended Data Fig. 1**), including a pie chart summarizing our estimated total effort (~10 person-years). We encourage the reviewer to review this new methods section and the accompanying figure. While it is

impossible to make a ‘perfect’ connectome, we are quite proud of the final product and the considerable effort towards quality control to minimize errors.

- (3) A recent preprint (“Neuronal “parts list” and wiring diagram for a visual system”) attempts to curate a similar (yet incomplete) inventory although in a different dataset with a different approach. From an outside perspective, it would certainly be helpful if the authors of both manuscripts could employ a common neuron type nomenclature for at least major neuron types and groups of neuron types. Some examples: Cm vs. Sm, Tm5a,b,c vs Tm5a,b,c,d,e,f. Obviously the authors cannot be held responsible for the nomenclature of others - but I think they should comment on the issue and do everything possible to facilitate cross-comparisons. Maybe they can further comment on how feasible it might be to actually reach unified nomenclature?

We appreciate this suggestion and understand the frustration of parallel nomenclatures. We were inspired by this request to “facilitate cross-comparisons” and a related one from Reviewer #3 to “flesh out differences” between the two data sets. In the last few months, we have carried out a comprehensive cell type matching between all the optic lobe cell types in our data set with all the cell types in FlyWire-FAFB across the two available optic lobe cell type efforts in that dataset (documented in Matsliah, et al. 2023 and Schlegel et al. 2023). We worked on this matching together with members of Greg Jefferis’s group, who are deeply familiar with FlyWire and are now included as authors on our revised manuscript (documented in **Methods: Matching cell types between the male optic lobe and FlyWire datasets**). The outcome of this matching is a satisfyingly high concordance between the data sets, as we now note in the manuscript: “We matched 98% of the cell types in our inventory (accounting for 99.8% of cells) to cell types in the female FlyWire-FAFB dataset^{20–22,30} (Extended Data Fig. 14a,b; Supplementary Table 7), providing strong evidence for the completeness of our inventory and the absence of significant cell-type-level sexual dimorphism. Cell types that could not be matched between the datasets are candidates for sexually dimorphic neurons.”

Supplementary Table 7 provides a list of all the matches at the cell-type level and will serve as an important resource for scientists in the field who wish to compare neurons in both datasets. We produced and now share links to neuroglancer views that make these comparisons directly accessible. We have prioritized this matching process over generating a standardized nomenclature since we believe the scientific value of matching neurons exceeds that of reconciling names, which can now be easily found in our provided table. The diverging nomenclature is an unintended consequence of the independent nature of our two efforts, but because of this independence, the high degree of confidently matched cell types between the data sets is genuinely remarkable. We agree that generating consensus cell-type names is an important goal, but it has been impractical to do so at this time for these very new data sets. After sharing our cell-type matching with the Princeton group, we have had productive discussions with them,

and we envision working with the Virtual Fly Brain team to establish a consensus nomenclature soon.

- (4) The authors introduce Optic Lobe Intrinsic Neurons (OLINs) and Optic Lobe Connecting Neurons (OLCNs). This naming scheme is a bit ambiguous: aren't Optic Lobe Connecting Neurons also intrinsic to the optic lobe? They connect optic neuropils and not optic lobes (which MeVPM neurons do I guess). The authors themselves state in line 102-103 that “the lamina, medulla, accessory medulla, lobula, and lobula plate, [...] <form> the structure called the optic lobe”. Would Optic Neuropil Intrinsic Neurons (ONINs) and Optic Neuropils Connecting Neurons (ONCNs) be suitable alternatives?

We appreciate this thoughtful suggestion. Our original intention was to avoid the term “neuropil” as it comes across as jargon mainly used by invertebrate neurobiologists, but upon reflection, this term is clearer and unavoidable for describing the fly brain. Therefore, we have adopted this suggestion and renamed the cell type groups. We also replaced most uses of the term “region” with “neuropil,” resulting in over 200 changes across the manuscript and methods.

- (5) The connectivity jitter plots (e.g. in Fig. 2h but especially in extended figure 3) are unable to show the data distribution. Can the authors provide histograms or perhaps adjust the jitter and/or transparency?

We apologize that these figures did not clearly show the data distribution. The plots in **Fig. 2c,h** and **Extended Data Fig. 4** have been updated with much larger spread of the points and modified transparency.

Other minor suggestions:

- In Supplementary Fig. 1: for some neuron types the top 5 output (e.g. LoVP74, LT51 only one top 5 input) is missing. For others the input seems missing e.g. MeVPM8 also the density plot shows at least some input.

We thank the reviewer for pointing out this potential source of confusion. Many cell types feature a long tail of very weak connections, whose significance is unclear. Therefore, we apply a very conservative threshold to discard extremely weak connections. We only include connections of strength >1 within the optic lobe volume (averaged across all cells of the type, per row) in the summary figure. For most cells this is hardly relevant, but for outputs of VPns or inputs of VCNs these marginal connections could impact the top 5. This threshold accounts for the missing connected cell types. For example, MeVPM8 (L) only has 4 input connections, all of strength 1 in the data set. We have updated the cover page of **Supplementary Fig. 1** and the

corresponding details in Methods: **Summary of connectivity and size by depth** to explain this threshold.

- Fig. 3a Too small to see missing cells

We apologize for this inconvenience. We have not succeeded in producing an alternative visualization that can show information about 15 cell types over 892 columns. We can assure the reviewer that in our .pdf files, these data are easily resolvable when zoomed in, and we will endeavor to retain the highest quality figure in the production version of the manuscript files. Additionally, the complete list of cell types for all columns, with their corresponding unique identifiers, is provided as **Supplementary Table 3**, and this table is referenced in the figure legend. Therefore, we expect that any interested readers will be able to find this information either by zooming in to the figure or by consulting the table (and our shared analysis code).

- Fig. 5b While these celltype summary plots are extremely helpful and informative it would help to understand the circuit/pathway (from lamina monopolar cell to LC neurons) with a circuit diagram

Thank you for this suggestion. We have added a circuit diagram as new panel **Fig. 5a**, that highlights the main connections of this set of 13 cell types from the medulla inputs (L2 and L3) and their photoreceptor inputs in the lamina, to the outputs (LC11 and LC15).

- Fig. 5d to what is the missing column label referring to?

This label is more helpful in **Fig. 3e**, where it refers to the columns on the edge of the eye-map, those found in the medulla, but missing from the lobula or lobula plate. We agree that this label is confusing in **Fig. 5d** and have removed it.

- Extended Data Figure 4c 3: R7p label is missing

This figure (now **Extended Data Fig. 5**) has been substantially updated to include a new analysis of pale and yellow column assignments and is labeled accordingly. A new, corresponding section (**Methods: Assigning R7 and R8 photoreceptors and medulla columns to different types of ommatidia**) documents this new analysis.

- Line 229-230 how does this extend to the lobula? Isn't T4 only connecting medulla with lobula plate?

We apologize for the confusion. The lobula coordinate system was built directly from the columnar cell types (Tm1, Tm2, Tm4, Tm9, Tm20) that extend from the medulla to the lobula,

and the lobula plate columns were built from sets of T4 cells. We have added an additional sentence to clarify this section: “Lobula plate columns were built by assigning sets of T4 neurons to each Mi1 in the medulla (Extended Data Fig. 6, Supplementary Table 4). Together, these sets of corresponding neurons were used to extend the same coordinate system to all three major neuropils.”

- Line 300-301 Is it that each synapse is predicted to be 50/50 ACh/Dop or is the average over the entire neuron 50/50 Ach/Dop with some synapses clearly being predicted to be Ach or Dop?

We have examined this carefully (also see figure in response to Reviewer #3’s question) and can confirm that the neurotransmitter prediction method, as currently implemented and trained, is not suitable for detecting co-transmission at Mi15 synapses. There is only a weak prediction for Dop at Mi15 synapses. We have updated this section to clarify that we are referring to the RNAseq data and to comment on how we do not see good evidence for co-transmission based on the predictions. We attribute this to the method (which was established to predict single transmitters at each synapse) and the training data, where we had deliberately avoided Mi15 and R8, the two cell types for which we have the best independent evidence of co-transmission.

- Line 350-353 The “The Drosophila Visual System Cell Type Explorer” is a great webpage!

Thank you very much. We built it to be a useful companion to neuPrint and plan to update it in the future as additional analyses and connectivity information become available.

Here are some nice-to-have features:

- Information about nt prediction

The ‘consensus’ NT predictions are listed at the top of each cell type’s webpage and in the connectivity tables below, the consensus NT for each connected cell type is also listed. We think that the consensus NT prediction is the most useful information for most users, and further details about synapse- and cell-level predictions are available from neuPrint.

- Interactive 3D viewer of all neurons of the neuron type

We were not able to include all neurons in the current style of 3D views since these rely on data that is embedded in each webpage, leading to very large file sizes and slow loading times. All neurons are viewable in the linked neuPrint pages, but to better accommodate this request, we are working on a set of interactive neuroglancer views for each cell type, that we will integrate with the web pages ASAP.

Reviewer #2

The paper presents a monumental effort of electron microscopy (EM) reconstruction and cataloging of the neurons encompassing the fly optic lobe. To say that this is heroic would be euphemistic. The level of detail and the extreme care taken throughout the process of alignment, segmentation, synapse identification, proofreading and annotation is truly remarkable. The outcome of this endeavor is highly successful with a comprehensive list of cells mapped onto the hexagonal grid composing the fly eye and with a coordinate system maintained across each optic lobe neuropil. This provides an unprecedented opportunity to match the structural organization of the fly visual system to the point-by-point view of the visual world from photoreceptors all the way to projections into the central brain. The predictions that can be drawn from this quantitative connectome, combined with the set of genetic fly lines that target single cell types, unleash the full potential for a comprehensive understanding of fly vision. In short, this is a generational resource to gain insight, develop hypotheses, and obtain the tools necessary to investigate every basic visual computation in a very powerful model organism. In the framework of the three levels of analysis proposed by David Marr, this work is arguably the best description to date of how the implementational level (i.e., the physical structure) informs the algorithmic level (i.e., the rules) that in turn explains the computational one (i.e., the goal). In this sense, the work constitutes a profoundly transformative achievement for the visual neuroscience community.

The paper is entirely descriptive, clearly explaining the classification approach by walking the reader through the rationale for branchpoint decisions. I particularly appreciated the cell type quantitative summary (Fig. 5b and Supp. Fig. 1) in which the authors have done a great job capturing and synthesizing the essential properties that characterize different cell types (neurotransmitter, morphology, connectivity, and size) while also leaving lots of room for further quantitative analysis and experimental analysis. Moreover, the synapse-by-depth metric that sorts neurons into their respective types nearly as effectively as the connectivity-based clustering is impressively compact and clever. This allows comparisons between EM and light microscopy (LM) data. The Cell Type Explorer web application is an extraordinarily useful resource to easily access connectivity and spatial distribution of synapses and cells in the hexagonal coordinate system. Fly researchers will strongly benefit from this open-source platform that speeds up the emergence of valuable insights.

This study forms the highest benchmark standard for the preparation and dissemination of forthcoming EM connectomic data sets.

I have no scientific criticisms; the work represents the first complete connectome of male fly visual system (and more). This is a milestone for the neuroscience community that, together with the first whole-brain connectome of a female fly (FAFB, FlyWire), will help bridge the gap between neural structure and function by guiding strong experimental predictions to test.

We thank the reviewer for this kind and thoughtful feedback and are honored by the many superlative remarks.

Minor:

L93: I would give a bit more context here by adding information reported in “EM volume imaging” of the Methods section “... imaged using seven customized FIB-SEM systems in parallel over a period of almost a year”

We have added the requested additional information.

L127-128: “motion vision pathway” in this case the authors are referring to “direction motion vision pathway”

Updated. Now referred to as the “directionally selective motion vision pathway”

L294: for consistency use fan-out with or without hyphen. Moreover, for the broader audience, it might be helpful to add a short sentence about the authors’ interpretation/speculation of the different fan-out among neurotransmitters

Fan-out is now spelled with a hyphen throughout. We have added a brief sentence speculating on two possible interpretations of this difference: “This difference may reflect a more targeted role for inhibition in visual circuits or higher efficacy of GABAergic synapses.”

L336: on LC11, the authors might also cite Keleş and Frye, 2017 (Current Biology)

We have added this reference for its careful characterization of LC11 responses.

L767: TM5a

Corrected to Tm5a.

Methods:

“EM volume synapse identification”, Line3: dot missing after “detection” and maybe a citation?
Citation and period added.

“LM-EM volume correspondence”, Line3: “for for”
Corrected.

Add a space before “Pipeline for rendering neurons”
Added an additional line before this section’s title.

Fig. 3a: the picture of the fly head on the right might be confusing because it looks like a picture

from outside (i.e., left eye), I suggest to use only a silhouette of the head without depth information (view from inside) or to flip the image along the vertical axis so that the fly is looking to the right and the eye corresponds to the right one from an outside view

We have adjusted the cartoon in **Fig. 3a** as suggested. The fly head is now shown as an outlined silhouette, and we have adjusted the shading to give a better impression that the eye is curving away from the viewer.

Caption of Fig. 3c: marker is between different quotation marks font
Corrected.

Add a short description (software, package, etc.) for the ANCOVA run in Fig. 4i
This information has been added to **Methods: Neurotransmitter prediction.**

Reviewer #3

This is an impressive manuscript. Aljoscha Nern and colleagues follow in the footsteps of Ramón y Cajal and finish what the latter started approximately 140 years ago: a catalogue of all cell types in the visual system of an insect. To understand a nervous system, one needs to know its components; this manuscript contains an (almost) complete inventory. In generating this dataset, the authors developed and used new software tools to analyze large amounts of electron microscopic images; they introduced a new nomenclature of cell types in the *Drosophila* optic lobe, and provide a genetic address to many of the catalogued neurons. The authors managed to condense an enormous amount of data into a surprisingly palatable format that was a joy to read. So far, most connectomic reconstructions were tainted by the lingering awareness that they were incomplete. This one is also incomplete, but it is probably as close to a complete connectome as it gets. As far as I can judge, the manuscript is of high quality and provides a rich resource that makes neuroscientists working on other model organisms green with envy.

We thank the reviewer for the detailed and thoughtful evaluation and appreciate this positive assessment of our work.

Below please find several suggestions for improvement:

1) My principal reservation is that the work remains purely descriptive. Previously published connectomes by the same authors (e.g., Takemura et al. 2013)—incomplete as they may be—always offered some concrete insight into the function of the circuit. In the concluding remarks of the present manuscript, the authors also foresee “countless experimental roadmaps for detailed investigation of brain circuit function”, yet, they fail to provide a single example of a functional prediction that emerges from this ultrastructural analysis. Recent work (which the authors are careful to acknowledge) demonstrates how predictions emerge from a comparable connectome (Seung 2023). I encourage the authors to give us a glimpse of what we can expect to learn from efforts like this. This would provide the icing on the cake.

We share the reviewer's motivation to distill functional insights from connectomic data, and indeed, this is a large thrust of the manuscript's **Introduction**. Since prior analyses of connectomes already generated many hypotheses, surely a much more complete data set would enable even more discoveries. As a practical matter, we cannot properly present novel connectome analyses in this already dense manuscript, and plan to do so in several future publications. Whereas other teams may have prioritized data analysis to make some new predictions, our group made the deliberate decision to establish new methods for the ‘descriptive’ aspects of the work. We built new tools and used them to quantify the properties of all the optic lobe neurons. Nevertheless, we are only too happy to summarize many exciting directions for analysis, which we've added to the revised **Concluding Remarks**:

Pathway analysis provides a direct source of functional hypotheses. For instance, neurons downstream of R7 are strong candidates for mediating visual responses to UV illumination⁶⁵, while those downstream of L3 are likely influenced by absolute luminance⁶⁶, and neurons downstream of T4 and T5 almost certainly receive directionally selective input⁸. Extending the spatial analysis of neuronal inputs—an approach that was essential for uncovering T4’s role in motion detection⁷—to larger cells within deeper circuits will be key for advancing mechanistic understanding of visual transformations, such as those implemented by the LC neurons⁴⁹. Additionally, identifying small-scale connectivity patterns, known as circuit motifs, has already yielded important insights into the gating of visual information⁵⁸ and the extensive role of normalization by interneurons in the optic lobes⁶⁰. This complete connectome, paired with our new tools for incorporating visual-spatial coordinates, enables a detailed evaluation of connectivity patterns, which may reveal heterogeneity⁶⁷, subtype-specificity as in the targets of pale and yellow photoreceptors²⁹ (Extended Data Fig. 5), or spatial variations (Extended Data Fig. 3). A particularly exciting avenue for discovery will come from spatially mapping molecular information—such as the identity of neurotransmitter receptors—onto connectomes^{43,68}.

2) The title of the manuscript contains a claim to completeness. To preempt disappointment, I recommend to disclose early in main text that the volume contains only part of the lamina. At present, the authors discuss this minor and almost unavoidable blemish only in the Methods section.

We have added this disclosure to section **Neurons of the *Drosophila* visual system**: “Our dataset contains all optic lobe neuropils in their entirety, except for the most peripheral neuropil, the lamina.”

The claim to establish “the most comprehensive survey of any insect optic lobe” (lines 84 and 85), in my opinion, cries out for a comparison with the only other comprehensive electron microscopic survey of the *Drosophila* visual system (Matsliah et al. 2023). The cursory mention of this highly relevant work by the FlyWire consortium strikes me as unfair. Differences between the two connectomes—one from a female and one from a male fly—should be fleshed out. For example, can LoVP109 be found in the (openly accessible) connectome of the female optic lobe?

We always intended to compare the two data sets, but at the first stage, it was essential for these two data sets to be independently cell typed. We were inspired by this request to “flesh out differences” and a related one from Reviewer #1 to “facilitate cross-comparisons” between the two data sets. As described in our response to Reviewer #1, we have carried out a comprehensive cell type matching between all the optic lobe cell types in our data set with all the cell types in FlyWire-FAFB (documented in **Methods: Matching cell types between the male optic lobe**

and FlyWire datasets, Extended Data Fig. 14, Supplementary Table 7). With regard to sexually dimorphic cell types, and specifically the rather unusual case of LoVP109, we now provide additional evidence in **Extended Data Fig. 13g**, and summarize in the main text: “We matched 98% of the cell types in our inventory (accounting for 99.8% of cells) to cell types in the female FlyWire-FAFB dataset^{20–22,30} (Extended Data Fig. 14a,b; Supplementary Table 7), providing strong evidence for the completeness of our inventory and the absence of significant cell-type-level sexual dimorphism. Cell types that could not be matched between the datasets are candidates for sexually dimorphic neurons. Tm26 shown in Extended Data Fig. 14c is one such example, while the second illustrated example appears to be a rare case of a missing cell type—LoVP109/LTe12 (FlyWire), is not found in the right optic lobe, but LM analysis suggests it is not a dimorphic neuron (Extended Data Fig. 13g).”

The revised manuscript now includes at least 11 references to work carried out in FAFB-FlyWire: references #20, 21, 22, 29, 30, 33, 37, 40, 49, 55, 61.

We note that it is a bit premature to interpret this comparison as a definitive analysis of sexually dimorphic optic lobe cell types. Sex-specific connectivity differences may be found between ‘identical’ cell types, and there is a large effort to exhaustively compare the male CNS to FAFB-FlyWire once the CNS is further proofread.

3) Despite the overwhelming amount of data and technical detail, I am convinced that some of the figures could be combined and presented more concisely. Figures 2, 3 and 4, for example, all describe the process by which neurons were assigned to cell types. I suggest to combine Figures 2 and 3 and move some of the space-consuming panels (e.g., Fig. 3a,d) to the Extended Data. The same applies to Figures 6 and 7. I appreciate that these figures sum up a large body of work, but they provide little useful information to the non-expert reader; and I expect that even a *Drosophila* neurobiologist cannot extract much information from these images and might prefer to consult the supplementary material or the web resource instead. Alternatively, the authors could display one exemplary rendering alongside the light micrograph of the corresponding split-GAL4 expression pattern from Fig. 8 and move the remaining panels to the Extended Data.

We thank the reviewer for the helpful suggestion, and truly appreciated their previous comment “The authors managed to condense an enormous amount of data into a surprisingly palatable format that was a joy to read.” We believe that our carefully constructed figure layout contributed to the surprising palatability of the format: the early figures (1-5) introduce and summarize the data set, then present different key aspects of the methods: cell typing (**Fig. 2**), the data-driven coordinate system (**Fig. 3**), the neurotransmitter predictions and accompanying ground truth data (**Fig. 4**), and the quantitative description of all cell types (**Fig. 5**). **Figures 6-7** provide a quick visual impression of the complete cell type diversity of all the neurons connecting the optic lobes and the central brain. Such a collection of cell types has never been

documented before, and these figures serve as an overview for which our supporting materials provide extensive details for those interested. The split-GAL4 lines of **Figure 8** are an essential document of how LM data informed the cell typing and vice-versa. In short, we strongly prefer to keep the figures formatted as they are and will leave final guidance about the number and length of figures to the discretion of the journal's editors.

The following further issues deserve some thought, analyses, or editorial changes:

The paper is well written. On occasion, however, the authors seem to place a strong focus on their own past achievements. I suspect that they might look back at this in a few years' time and cringe at some of the more self-aggrandizing passages. It is applaudable that many of the authors were involved in generating vast libraries of genetic driver lines over the past 15 years that advanced the field (lines 425–427), but this is not central to this manuscript. Given the wealth of possible points to discuss, there seems to be an undue emphasis on history.

We respectfully disagree. The effort to assemble split-GAL4 driver lines was essentially one long-running project of the manuscript's first author Aljoscha Nern, that deeply informed our cell typing and is the relevant scientific background for the collection of split-GAL4 lines we document and share in this manuscript. Rather than self-aggrandizement, we view this as a style of writing in which the scientists are more visible, and their contributions are part of the story. The correspondence of LM anatomy and GAL4 lines to the optic lobe cell typing is central to the manuscript, and we believe it sets our work apart from other fly connectome efforts, or any compendium of neuroanatomical information in just about any system. Nevertheless, we have toned down the “history lesson” somewhat.

The authors acknowledge that the low number of neurons releasing certain neurotransmitters impedes the classification of, for example, octopaminergic, dopaminergic, and serotonergic synapses. Can they exclude that the overwhelming number of cholinergic synapses in the training dataset causes cholinergic misclassifications? Have they tried to train the network with equal (albeit low) numbers of synapses of each type?

Yes, we can exclude the possibility of “cholinergic misclassification” as a major problem in our data through multiple lines of evidence:

1. The accuracy of the entire method was assessed 2 ways, by comparing held-out training data (**Fig. 4b**) and the new experimental data (**Fig. 4e**). Both “confusion matrices” show per-synapse prediction on the order of 90%. Furthermore, the less numerous transmitters are not predicted as ACh with greater frequency. For example, 10% of 5HT synapses are confused for Dop synapses, OA synapses are more likely to be confused for 5HT or Dop than ACh, etc.

2. The training set was balanced, not precisely as suggested, but to preserve the frequency of each transmitter type in the training (70%), validation (10%), and testing (20%) sets, as described in **Methods: Neurotransmitter prediction**.
3. The remarkable agreement with the neurotransmitter predictions from FAFB-FlyWire, as we describe in **Methods: Matching cell types between the male optic lobe and FlyWire datasets**. “The comparison of predicted neurotransmitters for the matched cell types provides yet another independent evaluation of both the putative matches and the neurotransmitter predictions (in two independent EM volumes, with independent training data, but using the same method of Eckstein *et al.*⁶³). In general, the concordance is very high...there are 544 cell types whose predictions can be evaluated, of which 503 agree (92.5%). Of the 41 mismatched predictions, 8 are OL predictions for histamine, which was not included in the FlyWire training or predictions. Of the remaining 33 mismatched predictions, 26 are OL predictions for glutamate and FW predictions for GABA (interestingly, there are no mismatched predictions for the opposite assignments).” Incidentally, we have independent ground truth data for several of these mismatches, and in all cases the OL predictions agree.

Computational neuroscientists would profit from easy access to skeletons and meshes of neurons (for example, in the standardized SWC format). I tried, but failed to download the segmentations and ROIs through the respective Google Storage buckets (flyemoptic-lobe/v1.0/segmentation and flyem-optic-lobe/rois/). I would welcome an extension of the Cell Type Explorer that includes the option to download meshes.

We apologize for this difficulty downloading the segmentations from the Google Storage buckets. We now provide additional information and methods, including example code for accessing the skeletons and meshes in **Methods: Data availability**.

The statement that “the prediction accuracy is further improved when the single-synapse predictions are aggregated across the cells of each type” should be clarified. Were single-synapse predictions first aggregated across synapses of one cell and then across cells of one type?

We have clarified this sentence to explain that the high-level aggregation we present is performed across all synapses of all neurons of each type. In **Methods: Neurotransmitter prediction** we further explain that these predictions are also aggregated for individual neurons and also across all synapses of all neurons of a type, with both versions stored in the database. The reason we aggregate all synapses across all cells is that we use a conservative threshold for number of synapses – many individual cells don’t have many synapses leading to less accurate predictions, but at the population level there are often enough synapses for higher-confidence predictions.

The text jumps between the present and the past tense and contains an inconsistent mix of American and British English; this should be ironed out. Some parts of the Methods section are written more eloquently, but less precisely, than others. The text fluctuates between wordy digressions (e.g., “while neurons have complex, extended 3D shapes, it is remarkable that 2D visual representations of neurons have typically served as extremely rich descriptions of cell types across generations of neuroscientists, from hand drawings of Golgi-stained neurons to computer-generated ray traces of reconstructed EM volumes”) and chains of sentences that start with “we took”, “we applied”, “we performed”. I suggest to restrict the (already extensive) Methods section to the pertinent bits, with readability in mind.

Our methods sections document the work of dozens of people with unique expertise and writing styles, reflecting the diversity of the team and the technical areas represented. Therefore, some differences between sections are unavoidable, but we have endeavored to minimize these. We have made several passes through the methods document and edited for readability and consistency. We have searched for and removed any British spellings. We used the state-of-the-art grammar checker Grammarly to review the entire document, carefully considered each of its suggestions, and accepted >200 of them. As the **Methods** sections are an online supplement and electronic documents are easily searchable, we see the length of our **Methods** document as a strength of the work and not a liability. We strongly prefer to err on the side of including too much information as opposed to omitting details.

The authors provide ample references to their own work (~27 out of 58), but occasionally forget to acknowledge that of others. Half a paragraph (lines 50–59) is devoted to the motion vision pathway without a reference to recent (connectome-guided) advances (e.g., Ammer et al. 2023; Braun et al. 2023; Cornean et al. 2024; Groschner et al. 2022; Sanfilippo et al. 2024).

We assure the reviewer that we did not intentionally slight people in the field, but two factors contribute to this bias in citations:

- 1) We have a very large team comprised of people from many different labs, including many people who have worked in the field for decades, often collaborating inside and outside of Janelia.
- 2) Given the journal’s limitations on citations, we deliberately de-emphasized functional studies and prioritized citing anatomical discoveries and/or the relevant methods (which strongly reinforces the bias towards Janelia authors). For example, in the section mentioned above, in those four sentences about motion vision, we only cite 2 references (in total), and both are to the foundational EM studies. When writing this section, it seemed unfair and impractical to “pick and choose” from the dozens of relevant papers (including >10 authored by our co-authors).

We have taken the reviewer's note to heart and have since added 12 additional references to the manuscript, that we hope the journal's editor will let us keep, and most of the additional citations are to work from other groups (including most of those suggested here).

In lines 63–65 the authors state that “there is little mystery about the role of visual brain areas—their neurons and circuits must be involved in seeing”; this is a truism, but hardly a profound insight.

This sentence sets up the following one, which explains how visual neurons should be analyzed. (“This core insight guides the analysis of the neurons engaged in vision: these neurons should sample and represent information from across the field of view, this information should then be transformed by a series of steps that extract increasingly selective signals, and finally, these signals should be conveyed to higher brain areas.”) Therefore, we respectfully disagree. These sentences are profound in that these deep structure-function connections cannot be made in most brain areas and may not be obvious to the general reader.

The first mention of “automatic segmentation” in line 95, should be followed by a reference to the corresponding part of the Methods section.

At this point in the text, we added a reference to the **Methods** section as an umbrella reference for the seven sections that describe the generation of the connectome.

Lines 108–109: “Optic Lobe Intrinsic Neurons (OLINs) have synapses confined to a single region.” Region is ill-defined, I suggest to replace it with neuropil.

We appreciate this thoughtful suggestion. Our original intention was to minimize the use of the term “neuropil” as it comes across as jargon mainly used by invertebrate neurobiologists. Upon reflection, we agree that “neuropil” is less ambiguous and preferable for describing the fly brain. Therefore, we have adopted this suggestion throughout and replaced many uses of the term “region” with “neuropil.” Also, following the suggestion of Reviewer #1, we have renamed the cell type groups as Optic Neuropil Intrinsic Neurons (ONINs) and Optic Neuropil Connecting Neurons (ONCNs).

Lines 156–157: “Nearly all neurons in the dataset were reviewed multiple times by several team members as part of this process.” The juxtaposition of “nearly”, “multiple” and “several” in one sentence makes it appear sketchy. Perhaps the authors could state this more precisely?

This sentence has been updated to be more concise and accurate: “All neurons in the dataset were reviewed by several team members.” More useful is that we've thoroughly updated the description of our proofreading and quality control in **Methods: EM volume proofreading**.

Lines 173–174: I suggest to spell “cell typing” as “cell-typing” and highlight that the connectivity-defined clusters were of approximately equal cell counts—an important feature.

We included a note about the clusters’ approximately equal cell counts in the figure legend. We reviewed all cases of “cell typing” in the manuscript and found three instances where it is used as a compound adjective followed by a noun, which we hyphenated. Other uses are unchanged, following our best understanding of the standard grammar rules for hyphenation.

In lines 176–179, the authors describe that putative misclassifications of cell types were “resolved using additional criteria, such as cell morphology or, when available, genetic markers.” Why didn’t the authors use a combination of connectivity and morphology-based clustering for all neurons? I appreciate that the collection of cell-type-specific driver lines can provide important hints for cell typing, but I struggle to understand how genetic markers, whose expression patterns were visualized in separate LM experiments, can be used to resolve discrepancies in EM data. Formally, genetic markers should only inform EM cell classification if they were used to drive the expression of markers detectable in the same sample under the electron microscope, such as EMcapsulins (Sigmund et al. 2023).

We regret using the term “genetic markers” so casually in this context, when we simply meant that expression patterns from the collection of driver lines can provide additional information for cell typing. We did not use genetic markers to assign individual cells, as in the example reference, rather we use them in the form of split-GAL4 lines to support decisions for splitting cell types. We updated this section to reflect the intended usage, which the next sentence continues with the illustrative example of Dm6/Dm19 (**Extended Data Fig. 3**). As the reviewer suggests, we did indeed use a combination of morphology and connectivity for typing all neurons (as explained in **Fig. 2, Extended Data Fig. 2-3, and Methods: EM volume proofreading, Overview of cell typing, and connectivity-based cell clustering**).

The main figures are not referenced in sequence: Fig. 5 is cited before Fig. 3.
Updated so that main figures are cited sequentially.

Lines 232–233: “They [volumes] can be visualized alongside the gray-scale EM data (Fig. 3d)...” Judging by Fig. 3d, I would describe this as “superimposed on” rather than “visualized alongside”.

Clarified to “visualized together with the gray-scale EM data”.

The capitalization in line 249 seems inconsistent. Why is only “Fluorescence” capitalized?
Updated to Fluorescence in Situ Hybridization (FISH)

In line 257, the neural network mentioned in passing, which was used to assign neurotransmitter likelihoods, should be accompanied by a reference to the Methods section or to the relevant literature.

Reference added to the method of Eckstein, et al. (2024).

The acronym EASI-FISH (line 262) should be spelled out. Instead of listing at length the methods that could be used to assign neurotransmitters to neurons, I suggest that the authors limit their description to the methods they used, but state them explicitly: The phrase “primarily using EASI-FISH” could simply be replaced by “using FISH and EASI-FISH”.

The acronym is now spelled out and this phrase has been updated as suggested. We retained the reference to an additional method, RNAseq, which provides ground truth neurotransmitter identity data for most of the cell types in the training set (as listed in **Supplementary Table 5**).

In lines 269–270, the authors state that “the consistency or accuracy of predictions is comparable between the types in the training dataset and cell types not used in training”. This statement should 1) be supported by a reference to the corresponding data/figure, and 2) be specified: Was the consistency or the accuracy comparable? If both were comparable, I suggest to replace “or” with “and”.

We have replaced “or” with “and” as suggested and included a reference to **Fig. 4c**. This statement is intended as a narrow summary of **Fig. 4c** and we believe that the preceding sentences provide sufficient context to interpret this conclusion: “Three of the 16 cell types (indicated by asterisks) were included in the training data, while the neurotransmitters expressed by six of these types are covered in our new, independent experimental data (labels on gray in Fig. 4c, experimental data in Fig. 4d). The fraction of the top transmitter prediction is high; 14/16 TmY cell types have >85% of synapses classified as one neurotransmitter.”

In line 294, the authors mention a “significant and potentially interesting difference across the neurotransmitters” with regard to the neurons' fan-out properties. The (statistical) significance should be supported by a statistical test, as in Fig. 4i.

We now provide the supporting result of the statistical test at the end of this sentence.

Line 300: “Mi15s”, could be lab slang for Mi15 neurons, or it could specify a subclass of Mi15 neurons (like TmY9b). This should be clarified.

Thank you. Updated to “Mi15 neurons”.

Mi15 neurons express both cholinergic and dopaminergic marker genes (Davis et al. 2020). Are the two neurotransmitters co-packaged (and co-released) at the same synapse or does one Mi15 cell house separate cholinergic and dopaminergic presynapses? A discrepancy between the

confidence of single synapse predictions and cell predictions in Mi15 neurons might provide a hint.

This is a very interesting question! Unfortunately, the current neurotransmitter prediction methods are not suited to directly address co-transmission since each network is trained as a classifier to predict single transmitters (and the few known cases of co-transmission were deliberately excluded from the training data). We looked for such a hint as the reviewer suggested in the two clearest cases of co-transmission (Mi15 and R8), and can confirm that a very, very weak ‘signal’ of the second transmitter can be found in the predictions of individual synapses:

These histograms show the network’s predictions across all the presynapses of these three cell types for the indicated neurotransmitters. These values can be interpreted approximately, but not exactly as “probability” (for further details see Eckstein, et al. 2024). Dm2 is included as a comparison as it is closely associated with Mi15. As can be seen Dm2 is predicted even more clearly as Ach, but the tail of higher probability Dop predictions for Mi15 and ACh predictions for R8 are far too small to use as the basis for any predictions. Nevertheless, this small hint suggests that a different network architecture and additional training data may improve this method as a detector of co-transmission. We have updated the text to clarify that the claim about co-transmission comes from molecular profiling, and the current method does not reliably detect two neurotransmitters for these cell types.

It was not clear to me, which data the authors refer to in their descriptions in lines 340–343.

Clarified: “The classic layer boundaries are shown as a reference (vertical lines in Fig. 5b), but the synaptic distribution and column innervation are presented with higher depth resolution since it is clear these established neuroanatomic features do not fully capture the organization.”

Line 451: “We use brain registration to juxtapose EM-reconstructed neurons in the same reference as registered LM data”—the term “brain registration” could do with a more detailed explanation.

We removed this unnecessary term, and now more plainly explain that: “We transformed the EM-reconstructed neurons into the same reference brain volume (Bogovic, et al. 2020) as registered LM data, allowing us to compare single neuron morphology and population patterns” And added the relevant citation to Bogovic, et al 2020.

Lines 471–472: “we asked whether specific connectivity patterns [...] are most prevalent”. I suggest to replace “whether” with “which” or reword the sentence.
Updated to “which”

While I am happy to read that “in the coming months, proofreading of the rest of the central brain, and eventually the ventral nerve cord, of this connected volume [?] will be completed”, I think that the text should describe the present work, rather than ambitious promises for the future.

We have removed the ambitious promises for future deliverables and have simplified the statement.

In Fig. 2d, the abscissa is titled “# cells per connectivity-defined cluster”. In case the hue of the red boxes corresponds to the cell numbers, I believe that this should be the title of a (missing) colour map; the axis label should be “cluster #”.

We have updated **Fig. 2d** to include a color map (with the title: “fraction of cells in cluster”), replaced the abscissa label with “cluster #” and retained the previous label as an annotation for the numbers of cells in each cluster.

The yellow colour in Fig. 2i is difficult to see.
Updated to a darker shade of yellow and thickened the line.

The colour bars in Fig. 4b,e lack labels.
Updated to include labels (“fraction”) for these bars.

Line 612: Why is “Neurotransmitter” capitalized?
Capitalization removed.

Line 625: Fanout/fan-out is spelled inconsistently throughout the text and figure legends.
The more common hyphenated version is now used throughout.

Line 697: “...identified by the algorithm.” It is not clear, which algorithm the authors refer to.
Clarified and now references the relevant **Methods** section.

The term “bodyIds” in Extended Data Figs. 2 and 7 should be spelled “body IDs” or “bodyIDs”, as in the Supplementary Tables.

Thank you for pointing out these inconsistencies. “bodyId” is the name of this property (the unique identifier for each neuron) in the neuPrint database, so we now define this clearly in **Methods: Connectome data access overview**. As this term is very specific to our database, we have removed several unnecessary uses of the term (such as in these two Extended Data Figures), and those that remain all use “bodyId” as the only spelling.

The fixation and staining of the sample in the second paragraph of the Methods section should be described in more detail, including concentrations and durations.

We have substantially updated **Methods: EM sample preparation** to include more details of all steps, including concentrations, durations, and temperatures.

Compared to the ~97% of presynapses that could be assigned to named neurons, the percentage of assigned post-synapses (55%) appears low. Why?

This is a good question! This asymmetry comes from a combination of biological and technical factors. The physical structure of *Drosophila* synapses, where a single, large presynaptic area of one neuron typically contacts multiple postsynaptic sites of often multiple different neurons, leads to a higher fraction of presynapses being assigned to neurons vs. postsynapses at initial stages of segmentation and proofreading. As proofreading progresses, many postsynapses are assigned, but it is not reasonable (or even possible) to connect all the very small fragments housing small numbers of postsynapses. Previous analyses comparing different *Janelia* FlyEM datasets (FIB-SEM medulla vs. TEM medulla, hemibrain regions, mushroom body study, etc.) have shown that very few new connections (between cells) are found at higher completion percentages, a paradigmatic case of diminishing returns. By comparison to other data sets, our completion percentage is quite high, as explained in the **Methods EM volume proofreading**: “An important metric for evaluating the “completeness” of a connectome, or a region of a connectome, is the percentage of all synapses where both the pre- and post-synaptic partners belong to identified neurons. We provide this completion percentage for the optic lobe neuropils in Extended Data Table 1, and these summary metrics show this new connectome to be based on one of the highest-quality reconstructions to date. The completion rate is high, and relatively uniform across neuropils. Across the whole dataset, the connection completeness is 52.7% (54.2% when the partial lamina, discussed below, is excluded). This is considerably higher than the hemibrain, where the corresponding metric is 37.5%¹.” We have now added a brief summary of the root cause of this asymmetry in completion metrics after these metrics are discussed in **Methods: Summarized inventory of visual neurons and connectivity**.

Reference 42 (of the main text) has, in the meantime, been published and should be cited accordingly.

Updated.

Methods reference 76 concerning the Trimesh Python package lacks information: “Dawson-Haggerty, M. Trimesh. (2024).”

Updated.

References:

- Ammer G, Serbe-Kamp E, Mauss AS, Richter FG, Fendl S, Borst A. 2023. Multilevel visual motion opponency in *Drosophila*. *Nat. Neurosci.* 26(11):1894–1905
- Braun A, Borst A, Meier M. 2023. Disynaptic inhibition shapes tuning of OFF-motion detectors in *Drosophila*. *Curr. Biol.* 33(11):2260-2269.e4
- Cornean J, Molina-Obando S, Gür B, Bast A, Ramos-Traslosheros G, et al. 2024. Heterogeneity of synaptic connectivity in the fly visual system. *Nat. Commun.* 15(1):1570
- Davis FP, Nern A, Picard S, Reiser MB, Rubin GM, et al. 2020. A genetic, genomic, and computational resource for exploring neural circuit function. *eLife.* 9:e50901
- Groschner LN, Malis JG, Zuidinga B, Borst A. 2022. A biophysical account of multiplication by a single neuron. *Nature.* 603(7899):119–23
- Matsliah A, Yu S, Kruk K, Bland D, Burke A, et al. 2023. Neuronal “parts list” and wiring diagram for a visual system. 2023.10.12.562119 Preprint at <https://doi.org/10.1101/2023.10.12.562119> (2023).
- Sanfilippo P, Kim AJ, Bhukel A, Yoo J, Mirshahidi PS, et al. 2024. Mapping of multiple neurotransmitter receptor subtypes and distinct protein complexes to the connectome. *Neuron.* 112(6):942-958.e13
- Seung HS. 2023. Insights into vision from interpretation of a neuronal wiring diagram. 2023.11.15.567126 Preprint at <https://doi.org/10.1101/2023.11.15.567126> (2023).
- Sigmund F, Berezin O, Beliakova S, Magerl B, Drawitsch M, et al. 2023. Genetically encoded barcodes for correlative volume electron microscopy. *Nat. Biotechnol.* 41(12):1734–45
- Takemura S, Bharioke A, Lu Z, Nern A, Vitaladevuni S, et al. 2013. A visual motion detection circuit suggested by *Drosophila* connectomics. *Nature.* 500(7461):175–81

Dear David Rowland:

We are pleased to provide the final documents modified in accordance with your guidance and the reviewers' final requests. All changes are detailed in this cover letter. We have updated the manuscript, methods, and figures to reflect all of this feedback and to include some minor updates to cell typing. The major changes to the manuscript include:

- We have reduced the number of primary figures following your guidance on 11/22/24. The submitted manuscript has 7 primary figures and 16 Extended Data Figures.
- The manuscript's text has been reduced to ~5700 words.
- The manuscript now has 60 references.
- The text, methods, and reporting summary have been updated based on the specific requests of the editors.

Please find point-by-point response to the flagged issues and the reviewer's additional comments.

I wish to participate in transparent peer review.

Yours sincerely,

Michael Reiser (on behalf of all the co-authors)

Flagged Issues

Please also note the following issues flagged for your paper:

1. Flagging that the manuscript is not in .docx format. Currently it is in PDF format.

The manuscript is now provided in .docx format.

2. The number of main text references should be 60 in total or less - currently there are 70.

The updated manuscript now has 60 references.

3. Flagging that methods references are not continuously numbered.

The methods references are now continuously numbered.

4. Please remove the main figures from the article file and re-supply them individually in an acceptable format such as EPS, AI, PS, PDF, PPT, PSD or XLS (for graphs) with editable vector files.

Files are now provided as PDF files.

5. Please remove the Extended data figures from the article file and re-supply them individually in EPS, JPEG or TIF format.

Extended Data Figure files are now provided as individual JPG files.

6. Please ensure that the text size in all figures is at least 5 pt Arial. Flagging that there are display items in the SI

The display item(s) referred to are probably Supplementary Fig. 1, which is a large PDF summary of the connectivity and morphology of all the cell types. We arrived at this compromise in consultation with David Rowland as a way to include substantial data that serves as the central reference to our data set, but is too large to place elsewhere.

7. Please provide a supplementary information guide.

Now provided as SI_Guide.docx

8. Flagging that there are more than 10 extended data figures. Currently there are 14.

We have direct confirmation from Editor David Rowland that additional Extended Data Figures will be allowed for this manuscript.

9. Flagging that there are potential third party rights issues in the figures - please check sources or if permissions are needed for the organs, cells and schematic illustrations in the figures.

We have checked and do not believe there are any 3rd part rights issues. All figure panels are produced by the co-authors of the manuscript and presented here for the first time.

10. Please provide a competing interest statement in the main text of the manuscript.
Now provided.

11. Flagging that there are potential third party rights issues in the figures and the TPR table has not been provided on eJP.

As explained in response to flag # 9 above, we do not believe there are any 3rd party rights issues and therefore we don't need a TPR table.

Issues relating to reporting and reproducibility

1. The statement on competing interests is missing from the manuscript although, in the EPC, authors declare no competing financial or non-financial interests exist.

Statement now included in the text of the manuscript.

2. Please note that the authors have made the 'Connectome data' available in a webpage (<https://neuprint.janelia.org/?dataset=optic-lobe:v1.1>). Further, login credentials are required for accessing the data. We are unsure if the data is publicly available, therefore, we have not made any requests to the authors in this regard. Kindly look into this.

The data are publicly available and login credentials are freely granted.

3. Information on code availability has been provided under the data availability section of the manuscript. A request has been made in the reporting summary, for provision of a separate 'Code availability' statement in the manuscript.

The manuscript provides a separate Code availability section, and the corresponding information is indicated separately in the reporting summary.

4. Please note that the manuscript reports the use of an invertebrate i.e. *D. melanogaster*.

Note this in the reporting summary and methods.

5. Please note that the manuscript does not have any additional Supplementary figures to evaluate. Kindly look into this if it requires any further attention.

This is correct, the manuscript only has a single, large PDF file as Supplementary Figure 1, with no additional supplementary Figures.

Guideline notes:

TRANSPARENT PEER REVIEW

I wish to participate in transparent peer review.

ORCID:

The online system allowed me to provide the ORCID for Michael Reiser (0000-0002-4108-4517) as the corresponding author, but we have ORCID information for all authors that we would like to supply but this was not possible in the online system.

The other two corresponding authors are:
Stuart Berg (ORCID 0000-0002-0766-0488)
Gerald M Rubin (ORCID 0000-0001-8762-8703)

STATISTICS:

Confirmed to be sound and conforming to guidelines.

REPRODUCIBILITY: To ensure that the quality and transparency of methods and statistical reporting (as discussed here) are sound before the paper is published, we have reviewed your Reporting summary and Editorial policy checklist editorially. I have attached two documents: one listing specific issues related to your manuscript and one containing an annotated version of the Reporting summary. Please ensure that, as well as the more general points below, the points highlighted in the attached documents are addressed in full, both on these forms and within the manuscript. Both forms should be uploaded as a “Related Manuscript” file type. The Reporting summary will be published with your paper.

TITLE:

Title is 62 characters with spaces.

SUMMARY PARAGRAPH:

Our summary paragraph matches the suggested format and is 234 words in length.

All other aspects of the submitted files are in compliance with the requests, so are omitted here for brevity.

Referee #1

The authors provide an extensive revision and have addressed all points raised in my review. In particular, the authors provide additional analysis (Extended Data Figure 1) that demonstrates that the data quality inconsistency of PRs is restricted to this cell type and not to a region, and the matching of cell types in this manuscript to the recent flywire optic lobe dataset. The remarkable consistency is a powerful confirmation of the cell type assignment approach. I congratulate the authors to this important, impactful paper.

We sincerely thank the reviewer for this thorough evaluation and the generous assessment of our work.

Prior to publication, there is only one section where the revised version might still need some simple clarification: The method for assigning DRA / p / y types to columns underwent a considerable workover. In addition to not only using connectivity information as in the previous version the authors now also use morphological features to do the assignment. This allows the assignment to also extend into the region of the medulla where photoreceptor quality is not optimal.

However, surprisingly, this approach drastically changed the overall number of pale and yellow columns and subsequently the ratio of pale / yellow columns. I am not sure about the morphological features used here and would love to see some clarification in the final manuscript on the following questions:

- Are there clear, quantifiable criteria for a Dm8 home column which can be used for the assignment?
- Do Dm8b neurons also have home columns which can be used to assign the pale type to columns with potentially missing aMe12 branches?
- The authors argue that a substantial number of R7_unclear neurons maybe R7y even though the connectivity suggests that these neurons are R7p (preference connection to Tm5b & Dm8b). Doesn't this question the connectivity argument ($R7y > Dm8a \& Tm5a$ and $R7p > Dm8b \& Tm5b$) that the authors used and that is reported in the literature?

Incidentally, the optic-lobe:1.0 version is no longer available from neuprint at the time of writing this review. It would be very welcome if the older version were still available from neuprint.

This was a momentary error when these data were not available, but from now on, we consider the final, release version of the data set to be 1.1. Version 1.0 is best thought of as a pre-release candidate. Version 1.0 (in the form of v1.0.1) will continue to be available together with version 1.1. Version 1.0.1 corrects a minor metadata issue that resulted in incorrect results for some queries for a small number of cells (under 1%) but is otherwise identical to version 1.0, i.e. there are no changes to the segmentation, synapse predictions or cell type annotations.

I support to enhance the transparency of the peer review process, as suggested in the reviewer guidelines, by signing this review: Robin Hiesinger

We thank the reviewer for these comments, which we address below and in our revision to the corresponding Methods section.

The reviewer is correct that the pale/yellow ratio in our new annotations is considerably changed, which is almost entirely the result of changing the assignments of a significant number of former 'R7p/ R8p' labels to 'R7_unclear/R8_unclear.' We would not describe this change in the p/y

ratio as surprising, rather, it is an expected result of being more cautious about assigning some of the cells previously classified as ‘pale,’ which was a decision we reached after considerable review of these assignments. As we note, the reclassification into ‘_unclear’ subtypes was conservative. The subset of photoreceptors for which our primary morphological yellow and pale markers (Tm5a and aMe12 vertical branches) are informative is almost identical between the previous and current subtyping.

That the reconstructed R7 photoreceptors can be divided into multiple groups based on connectivity is an empirical result that is not dependent on assumptions from the literature. What appears unclear, at least for some cells, is how these groups map onto the pale and yellow subtypes defined by Rhodopsin expression, and there are issues that prevent us from producing a complete assignment. Our results suggest that there may be two different types of columns with yellow (as defined by Rhodopsin expression in photoreceptors) photoreceptor input, based on synaptic connectivity within each column. But it is difficult to distinguish a subset of yellow columns from pale columns based on all available information. Preferring a conservative approach, we therefore labeled such columns and photoreceptors as ‘unclear.’

We primarily relied on Tm5a and aMe12 branches as morphological markers, and we also examined Dm8 cells (specifically their vertical branches or ‘home columns’), but these cells (and the ‘home column’ concept in general) only played a minor supporting role in our annotations. The reason for this is that the exact correspondence between the yellow- or pale-specific yDm8 and pDm8 subtypes reported in prior light microscopy studies and our connectivity-defined Dm8a and Dm8b has not yet been established (this will likely require new experiments). To clarify our approach and the thought process behind it, we have expanded and revised the Methods section about pale/yellow assignments. The major additions and changes are as copied below:

“As Rhodopsin expression cannot be directly detected in EM image (without additional genetic labels that were not used in this or prior EM studies), assigning candidate pale and yellow photoreceptor subtypes to EM reconstructions relies on morphological features (the presence of arbors of specific cell types in a column) that were identified by prior light microscopy studies. As described further below, we used both such anatomical markers and synaptic connectivity to assign candidate subtypes to the reconstructed R7 and R8 photoreceptors in this dataset. We used synaptic connectivity to distinguish different groups of photoreceptors and anatomical markers established in prior studies (primarily Tm5a and aMe12 branches) to assign subtypes to these groups. The morphological markers also allowed us to distinguish photoreceptor groups without relying on synaptic connections and to assign some columns without reconstructed R7 or R8 (Extended Data Fig. 1g,h) cells as yellow or pale. However, as further discussed below, it is unclear whether the available information is sufficient to predict subtypes for all R7 and R8 and

we accordingly typed a substantial subset of cells (and columns) as ‘R7_unclear’ or ‘R8_unclear’. “

“The identification of these ‘home columns’ was primarily based on visual inspection of Dm8 cells (with candidate cells chosen by connectivity). We note that for small vertical branches it can be somewhat ambiguous whether these are home column arbors. We did not attempt to introduce a more quantitative definition of a Dm8 ‘home column’ since EM-resolution anatomical ‘ground truth’ data (which would be necessary to test the utility of such a definition for identifying pale and yellow columns) are currently not available.”

“We found that nearly all columns with a candidate R7y cell suggested by synaptic connectivity included a Tm5a arbor but lacked vertical aMe12 branches, in near perfect agreement with expectations for R7y neurons (reported to be associated with Tm5a but not aMe12 vertical branches). Accordingly, we typed this subset of R7 cells as R7y cells. We found that these R7y cells were typically also associated with vertical branches of one Dm8 type (Dm8a), identifying a potential additional marker for these columns. While most R7y were annotated based on both morphology or connectivity, only one type of evidence was used for a small subset of these assignments.”

“Our results suggest the possibility that yellow R7 (and R8) cells (as defined by Rhodopsin expression), may terminate in different types of medulla columns: Columns with Tm5a and Dm8a arbors (here identified as yellow columns) and other columns more similar to those with pale photoreceptor input. Further testing this hypothesis will likely require new experimental data. Although light microscopy analyses of Dm8 cells, which are potential additional markers for pale and yellow columns, have revealed two Dm8 subtypes, one (yDm8) associated with yellow and one (pDm8) with pale columns, the exact correspondence between yDm8 and pDm8 and the connectivity defined Dm8a and Dm8b is unclear. For example, we find similar numbers of Dm8a and Dm8b cells whereas yDm8 have been reported to outnumber pDm8. “

Referee #2

The authors have adequately addressed my suggestions and I commend the authors for making substantive changes to the manuscript, which I feel is significantly improved. I have no further comments and I strongly recommend the paper for publication in Nature.

We thank the reviewer for this strong support of our work.

Referee #2 (Remarks on code availability):

Although I am not very familiar with Python, the code includes sufficient instructions. However, I encountered multiple issues with dependencies installation and couldn't obtain all the necessary libraries to run the `src/completeness/connection-completeness_named.py` script. Additionally, I was unable to locate the Jupyter notebook page https://github.com/reiserlab/male-drosophila-visual-system-connectome-code/blob/main/src/pythonbootcamp/access_skeleton_and_mesh.ipynb for storing skeletons as meshes.

The missing files are now available in our updated repository. To address the challenges of dependency installation across the major platforms (Linux, Windows, and MacOS), we have moved to Pixi, a cross-platform package management tool, and simplified the installation instructions.

Referee #3

The authors have addressed all of my concerns. This paper sets a new standard for connectomes to come. Congratulations!

Four minor points should be rectified before publication:

1. The updated Fig. 4e contains neurotransmitter predictions for 78 cell types, but the corresponding caption still refers to 79 cell types.

This was an error that we have corrected by updating the figure and the corresponding text. 78 is the correct number of cell types (with accompanying explanation in the methods section

Training and evaluation data for transmitter predictions (related to Fig. 4).

2. The link to the example Jupyter notebook for accessing skeletons and meshes (https://github.com/reiserlab/male-drosophila-visual-system-connectome-code/blob/main/src/pythonbootcamp/access_skeleton_and_mesh.ipynb) does not work and the promised button to access SWC skeletons via the neuPrint+ webpage is either difficult to find or not yet implemented. This should be fixed ahead of publication.

The missing files are now available in our updated repository.

3. The collection of FlyWire papers has been published in the meantime and the corresponding references should be updated accordingly.

References to all (known) published papers have been updated.

4. The example chosen by the authors to demonstrate the value of connectomics in the introduction is, in my opinion, a particularly unfortunate one. The connectome by Takemura et al. (2013)—the most extensive of its kind at the time—proposed a circuit mechanism to compute

the direction of visual motion. The reconstruction, and the suggested mechanism, lacked crucial cellular detail, but inspired a functional follow-up based on false premise (Behnia et al., 2014). The authors cite both papers but omit to mention recent work that seems to have clarified one mechanism of direction selectivity (Groschner et al., 2022). If the authors decide to keep this example, but find it impractical to pick and choose from the many relevant papers, perhaps they should instead refer to one of the recent reviews on the topic.

We thank the reviewer for this suggestion, and now only cite the nice review article by Borst and Groschner (2023).

Referee #3 (Remarks on code availability):

I downloaded and tested samples of the extensive collection of code, all of which contained sufficient explanation in README files to reproduce the results. As mentioned above, the file 'access_skeleton_and_mesh.ipynb' is missing from the repository.

The missing files are now available in our updated repository.